# Real-world stress resilience is associated with the responsivity of the locus coeruleus

Marcus Grueschow [1✉], Nico Stenz[2,3], Hanna Thörn[2,3,4], Ulrike Ehlert [4], Jan Breckwoldt[5], Monika Brodmann Maeder[6], Aristomenis K. Exadaktylos[6], Roland Bingisser[7], Christian C. Ruff [1,8] & Birgit Kleim[2,3,8✉]

Individuals may show different responses to stressful events. Here, we investigate the neurobiological basis of stress resilience, by showing that neural responsivity of the noradrenergic locus coeruleus (LC-NE) and associated pupil responses are related to the subsequent change in measures of anxiety and depression in response to prolonged real-life stress. We acquired fMRI and pupillometry data during an emotional-conflict task in medical residents before they underwent stressful emergency-room internships known to be a risk factor for anxiety and depression. The LC-NE conflict response and its functional coupling with the amygdala was associated with stress-related symptom changes in response to the internship. A similar relationship was found for pupil-dilation, a potential marker of LC-NE firing. Our results provide insights into the noradrenergic basis of conflict generation, adaptation and stress resilience.

[1] Zurich Center for Neuroeconomics (ZNE), Department of Economics, University of Zurich, Zurich, Switzerland. [2] Division of Experimental Psychopathology and Psychotherapy, Dept of Psychology, University of Zurich, Zurich, Switzerland. [3] Department of Psychiatry, Psychotherapy and Psychosomatics, University of Zurich, Zurich, Switzerland. [4] Division of Clinical Psychology and Psychotherapy, Dept of Psychology, University of Zurich, Zurich, Switzerland. [5] Medical School, Deanery, University of Zurich, Zurich, Switzerland. [6] Accident and Emergency Department, Inselspital Bern, Bern, Switzerland. [7] Department of Emergency Medicine, University Hospital Basel, Basel, Switzerland. [8] These authors jointly supervised this work: Christian C. Ruff, Birgit Kleim. ✉email: marcus.grueschow@econ.uzh.ch; birgit.kleim@uzh.ch

Mental disorders are a major source of cost and societal burden worldwide, and the prevalence of such disorders is on the rise[1,2]. A decisive contributing factor is the increased level of acute stress inherent to society[3,4], particularly in the work place[5]. Even though exposure to prolonged stress or potentially traumatic events generally increases vulnerability to psychopathology, individuals vary considerably in how they respond to such stressors[4]. The majority of individuals exhibit resilient responding without any psychological problems or only minimal, transitory reductions in everyday functioning. Others, however, exhibit substantial stress-related psychopathology comprising anxiety and depression symptoms[4,6–9]. Despite great interest in advancing our mechanistic understanding of stress-related psychopathology[10], the biological basis for human stress resilience and its heterogeneity remains largely unknown. Here we highlight a neurobiological mechanism associated with stress resilience in humans and may therefore help to improve prediction and treatment of stress-related psychopathology.

Decades of invasive animal neurophysiology have associated the vulnerability to prolonged stress with a hyper-responsivity of the locus coeruleus, the noradrenergic (LC-NE) arousal system[11–15]. The LC-NE is a small pontine nucleus that sends numerous projections throughout the entire central nervous system[16,17] and plays a vital role in the central stress circuitry[18–22]. The LC-NE is ideally suited for upregulating various physiological processes that mobilize energy and promote autonomic adaptation in response to stress[23–27]. Critically, sustained stress responses associated with LC-NE hyper-responsivity have been shown to contribute to chronic anxiety and depression, fear, posttraumatic stress disorder (PTSD), increased risk of hypertension, and cardiovascular disease[28–35]. Moreover, recent reports suggest that the functional coupling of noradrenergic projections between the locus coeruleus and amygdala—a region associated with the amount of emotional intensity, fear and threat perception in humans[36–39]—promote stress-induced anxiety-like behavior in mice[30,40]. In humans, however, a lot less is known about the relationship between LC-NE hyper-responsivity and stress vulnerability. Even though heightened LC-NE activity has been observed in patients with anxiety, depression, and PTSD[41–46], the majority of human studies has for ethical reasons only been able to investigate brain activity accompanying stress-related symptoms after severe stressors had been encountered[46–48]. Such studies essentially measure a system already perturbed by stress; they thereby can neither identify the brain mechanisms that predispose an individual to be vulnerable to stress nor help us predict who will succumb to stress and who will be resilient. The current gold standard for psychopathology predictions consists of standardized verbal surveys, but these can be unreliable due to self-report bias and other factors[49]. Evidently, there is a clear need for truly prospective studies linking individual characteristics of the human LC-NE system to the severity of psychopathology subsequently induced by real-life stressors[10,11,50–52].

Here, we measured individual LC-NE responsivity in a laboratory task and use this to predict the degree by which participants are affected by future stress in the real world. We addressed this question by acquiring behavioral measures, pupil-dilation (an external marker associated with noradrenergic LC-NE firing[53] and cholinergic activity[54]), and fMRI data of LC-NE responses as well as functional coupling between LC-NE and amygdala[30,40]. This allowed us to identify which of these markers is associated with changes in stress-related symptomatology, thereby helping us to understand the neural mechanisms of stress-related psychopathology and inform future intervention and treatment methods[10,11,50–52].

To assess the responsivity of the LC-NE system inside the MR scanner, we employed the emotional-Stroop task[36,55,56], a well-established laboratory measure of affective conflict[57,58]. Affective conflict in this task serves as a model for conflicts that people have to resolve in emotionally charged situations that are often perceived as real-life stressors[58,59]. Considerable evidence suggests that such conflict-related signals engage the arousal system in monkeys and humans. For instance, conflict signals in the macaque brain, induced via task-congruent and -incongruent stimuli of monkey faces, predicted subsequent changes in pupil size and reduced behavioral distractor interference[60,61]. These data are consistent with the hypothesis that pupil-linked arousal mechanisms regulate conflict adjustments in non-human primates[60,61]. Moreover, several prior human functional imaging reports show reliable involvement of LC-NE system during conflict resolution involving stroop-tasks[62–64] as well as during tasks requiring the resolution of unexpected uncertainty[65]. We chose the conflict task over another standard task reported to activate the arousal system, the odd-ball task[66], because a previous report indicated no correlation between human LC-odd-ball-responses and any additional physiological measure such as pupil-dilation, skin-conductance or heart-rate[46]. Critically, this study also did not find any association between the odd-ball induced LC-NE response and the severity of anxiety or depression symptoms, which are the focus of the present work.

In the emotional-Stroop task we employed, participants categorized faces according to their emotional expression (happy vs. fearful), while at the same time ignoring overlaid emotionally congruent (C) or incongruent (I) words ("HAPPY", "FEAR", Fig. 1a). In this task, conflict arises from an emotional incompatibility between task-relevant and task-irrelevant stimulus dimensions. The resolution of conflict incurs processing costs, including an upregulation of task-relevant information[67], which have been associated with increased arousal and noradrenalin release thought to involve the LC-NE[68–72]. Behaviorally, the conflict is typically observed as higher reaction times (RT) for incongruent than congruent trials[36,55,73] and as congruency-sequence effects[73,74] (Fig. 1b): responses in conflict-inducing incongruent trials are faster when the previous trial was also incongruent (II), compared to when the previous trial was congruent (CI), reflecting time-consuming noradrenergic upregulation processes necessary when conflict is encountered after no-conflict trials[68,69]. These upregulation processes have lasting effects and therefore carry over to the subsequent incongruent stimulus on II trials[36,55,73,75]. We thus contrasted CI > II trials (which are identical in terms of presented stimuli and response requirements) to isolate neural processes involved in potentially noradrenergic[69–72] upregulation of cognitive control. This contrast essentially provides us with a measure of how much an individual brain is taxed by upregulation to resolve emotional conflict. We indexed the effects of this contrast on basic LC-NE activation, the downstream consequence of functional coupling between LC-NE and amygdala, and the peripheral LC-NE-related pupil dilation[70–72]. Using these measures, we could thus test whether higher responsivity of human LC-NE before the onset of a real-world stressor may predict the degree to which an individual will be affected by this stressor.

To accomplish this, we used a prospective design in a sample of medical students prior to their first medical internship. Medical students constitute a typical at-risk population: They have recently been identified as being alarmingly vulnerable to stress-related disorders[76,77], presumably due to ample exposure to significant stress and adversity during their medical internships[78,79]. We indexed levels of depression and anxiety at three time points: Prior to the internship (at the same time as fMRI and pupillometry) as well as 3 and 6 months later during the internship. This repeated clinical assessment protocol allowed us to account for

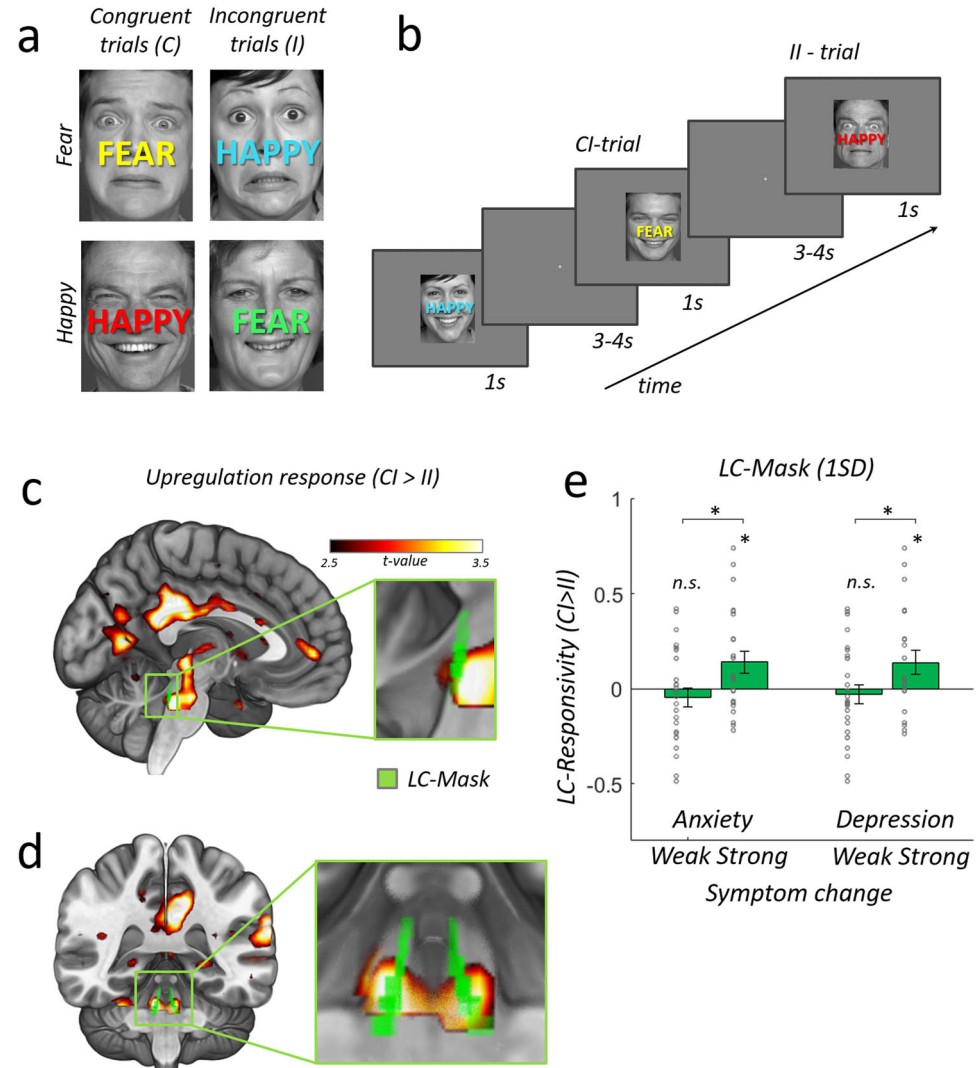

**Fig. 1 Experimental task and neural conflict-induced upregulation responses (CI > II). a** Example stimuli illustrating all four possible face/word combinations in the emotional-stroop task. Face stimuli used in our experiment were identical to the face stimuli used in Etkin et al. 2006. For illustrative purposes, we have replaced these images here with open access face stimuli (https://faces.mpdl.mpg.de/imeji/). Participants were instructed to react to the facial expression while ignoring the overlaid word and to answer as fast and accurately as possible. On each trial, the word color was randomly assigned in order to avoid adaptation effects. **b** Trial presentation schedule. A CI-trial is an incongruent trial preceded by a congruent trial. An II-trial is an incongruent trial preceded by an incongruent trial. Subtracting neural responses for II from CI trials reveals regions involved in the upregulation response (CI > II), while subtracting neural responses for CI from II trials reveals regions associated with implicit conflict adaptation (II > CI). See Supplemental Methods for details on stimulus presentation and counterbalancing of conditions. **c** Cortical and subcortical regions involved in generating an upregulation response to resolve conflict. Mid-saggital slice with activation clusters shows higher activity to incongruent trials preceded by a congruent trial (CI) as compared to incongruent trials preceded by an incongruent trial (II) (left superior temporal cortex (STC), posterior cingulate cortex (PCC), anterior visual cortex and a large subcortical cluster, FWE-cluster-correction at $p = 0.05$ with cluster-forming-threshold at $p = 0.001$, One-sample $t$ test, one-sided, the pseudo-color-map illustrating the one-sample $t$ statistic applies to all panels). Inset shows magnified lateral-view of subcortical cluster and an overlaid locus coeruleus mask in green (2SD-mask from Keren et al.[88]). **d** Coronal view of standard brain and magnified view of bilateral LC upregulation response (hot colors) overlaid with LC mask (green). **e** Participants ($N = 48$) with high subsequent anxiety/depression symptom changes show significantly stronger LC-NE responsivity (CI > II) than participants with lower symptom changes (median split), two-sample t test, two-sided, anxiety: $p = 0.019$; $T = 2.431$, depression: $p = 0.037$; $T = 2.154$. Bar plots show the LC-responsivity strength extracted from the CI > II contrast in the physiological noise controlled, unsmoothed data as weighted-average of LC-1SD mask voxels (see Supplemental Fig. S8 for detailed statistics and comparison with LC-2SD mask voxels). Single dots show individual data. Errorbars represent ±SEM. *$p < 0.05$. Source data are provided as a Source Data file.

initial individual baseline levels of distress prior to the real-world stressor and to fully capture expected variability in stress resilience amongst participants[4,80]. We quantified the utility of our biomarkers by comparing their predictive accuracy to that of subjective reports and ensured external validity by cross-validating our predictions[81].

**Results**

**Conflict-induced upregulation involves LC-NE.** We first established that our paradigm was indeed suitable by fully replicating previous behavioral and neural effects of conflict and trial sequence[36,82,83] (Supplemental Figures S1, see the Supplemental Information for comprehensive results, figures, and tables). Most

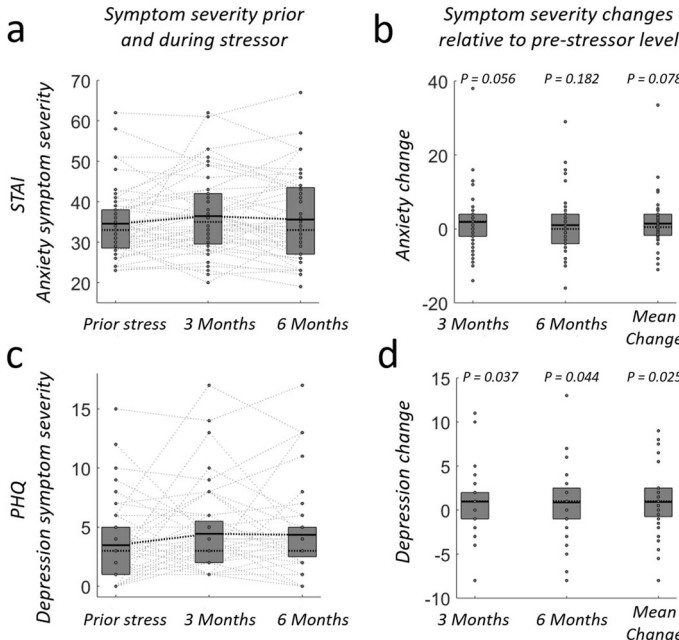

**Fig. 2 Symptom severity scores and severity changes across time. a** Anxiety symptom severity prior to and after 3 and 6 months of medical internship. **b** Anxiety symptom severity changes relative to initial anxiety baseline level prior to the internship (time of imaging experiment). One-sample t test, one-sided. **c** Depression-symptom severity prior and after 3 and 6 months of medical internship. **d** Depression-symptom severity changes relative to initial depression baseline level prior to the internship. Each dot (and their connections) represents data (and changes) for a single subject ($N = 48$). Top and bottom of boxes indicate 75th and 25th percentile of the underlying distribution respectively. Horizontal lines within boxes indicate the mean (black) and median (dotted). One-sample t test, one-sided. Source data are provided as a Source Data file.

importantly for our present purpose are the so-called trial-sequence effects: When a conflict trial is preceded by a no-conflict trial (CI), the brain needs to mobilize and upregulate resources on the current trial in order to meet these response conflict challenges. The effects of this mobilization are thought to persist and affect the next trial, leading to less conflict for subsequent conflict trials (II). Thus by contrasting neural activity (CI > II), we isolated the effects of individual LC-NE upregulation in the presence of identical stimuli and motor demands. In line with our predictions, the locus coeruleus was strongly recruited during conflict-induced upregulation. Contrasting CI > II trials (Fig. 1c–e) revealed robust activity in the midbrain/brainstem (cluster extent = 170, degrees of freedom (df) = 47, non-parametric $P$(FWE) = 0.039, X/Y/Z: 6/-27/-10, Fig. 1). A region-of-interest analysis in the locus coeruleus revealed significant CI > II responses, confirming a role of the subcortical noradrenergic arousal system in the conflict response (LC-Right: non-parametric $P$(SVC) = 0.003, X/Y/Z: 6/-37/-28, Fig. 1c–e and Supplemental Table S1). Other brain areas previously reported for this contrast (such as amygdala and DLPFC)[36,55,84] were also replicated here (Supplemental Fig. S2 and Supplemental Table S1; see also Supplemental Fig. S3 for replications of regions identified with the reverse contrast II > CI).

**Real-world stress exposure is associated with elevated anxiety and depression symptoms**. To predict individual resilience to real-wold-stress, we acquired our physiological laboratory measures as well as initial baseline anxiety and depression scores in the week prior to the onset of a stressful medical internship (timepoint $T_0$). To quantify the symptom-severity change from baseline level, we obtained additional anxiety and depression scores at 3 months ($T_1$) and 6 months ($T_2$), please see Methods for details. As expected, group-level symptom severity for both psychological test scores increased over time due to the stressful medical internship (the main effect of time for anxiety, $F = 4.01$,

$p = 0.022$ and depression, $F = 3.11$, $p = 0.049$; repeated-measures ANOVA, controlling for gender and age, degrees of freedom=2). Post-hoc t tests further identified symptom increases at 3 and 6 months relative to baseline level, albeit not always significantly (Fig. 2) (Depression: 3 months: $T_{(1,47)} = 1.83$, $p = 0.037$, 6 months: $T_{(1,47)} = 1.75$, $p = 0.0448$, mean-change: $T_{(1,47)} = 2.00$, $p = 0.025$, Anxiety: 3 months: $T_{(1,47)} = 1.62$, $p = 0.056$, 6 months: $T_{(1,47)} = 0.917$, $p = 0.182$, mean-change: $T_{(1,47)} = 1.44$, $p = 0.078$, one-sample t test, one-sided). The observed symptom changes are smaller than those reported in a previous study involving the American medical system[79], suggesting comparably milder levels of stress in the current Swiss cohort. Most importantly, however, we observed substantial interindividual variability in these symptom changes (see the individual symptom profiles in Fig. 2), confirming the individual differences in susceptibility to stress required for our predictive approach. Please note that we did not aim to predict whether or not a participant develops stress-related symptoms above a clinically relevant cut-off. However, see the Supplemental Information for quantification of effect sizes, demographic information, participant counts reaching clinically relevant cut-offs, and the relationship of symptom severity changes to non-prospective measures such as adverse events experienced during the internship (Supplemental Tables S2–S5 and supplemental section: adverse events during internship).

**Optimizing brainstem signals**. Optimal functional imaging of the brainstem, and in particular the LC, is notoriously difficult due to the small size of the nuclei involved, their proximity to the ventricles, and inherently low signal-to-noise ratio in the brainstem. In order to unequivocally identify LC-NE activity, non-standard techniques would be ideal for both data acquisition and analysis[85–87]. On the acquisition side this would, for instance, entail high-field imaging and partial-brain coverage, which allows particularly small (submillimeter) voxel resolution to avoid partial voluming and reduce pulsating artefacts from the adjacent 4th

ventricle (please see the limitations section for a discussion of fundamental methodological steps to improve brainstem imaging). However, the use of such a specialized imaging protocol would preclude whole-brain imaging and therefore inferences about influences of the LC on other brain systems (e.g., the amygdala and neocortical areas involved in conflict processing). Moreover, it would make it difficult for our approach to be replicated and extended in standard fMRI lab settings around the world. Thus, we opted for a standard 3T scanner and a routine fMRI-sequence with relatively low voxel resolution (2.5 mm isotropic) that nevertheless retains good signal-to-noise ratio in the brainstem (Supplemental Figs. S4–S6). Importantly, to ascertain the specificity of our results to the LC, we conducted multiple mutually corroborating analyses. These included weighted averaging for data extraction from brainstem regions of interest, control for additional brainstem nuclei, controlling for physiological nuisance variables based on principle component analysis of individually identified CSF probability tissue classes, as well as applying these nuisance variables in additional regression models to both smoothed and unsmoothed data. Please see the methods and supplementary methods section for detailed descriptions of these techniques, additional figures (S4–S7) and statistical results tables (Tables S4 and S6), as well as a formal temporal-signal-to-noise (tSNR) analysis of the whole brain and specifically the brainstem. Given all these methodical procedures, and the specificity of our results, the signals we extracted can be cautiously attributed to the LC despite our use of a more standard imaging protocol.

**LC-NE responsivity is associated with stress-related anxiety and depression symptom change.** To demonstrate that the LC is indeed reliably related to the CI > II contrast, while at the same time taking individual differences in LC conflict responsivity into account, we split the sample into participants who went on to develop stronger vs weaker mean anxiety/depression symptoms (median split). This allowed us to analyze LC responses (the weighted average LC-1SD extracted, physio-corrected, unsmoothed fMRI data) to our conflict task in people with high versus low susceptibility to develop psychopathology in response to stress. Given our hypothesis—derived from rodent studies—that hyper-responsivity of the LC-NE predisposes vulnerability to prolonged stress exposure, we expect participants with high symptom severity changes to also show high LC responsivity, while participants that exhibit less or no changes in symptom severity are expected to show low LC responsivity. Indeed, we found that participants with high symptom severity changes exhibited significant LC-NE responsivity (CI > II) that was significantly stronger than the corresponding effect in participants with low symptom severity changes (Fig. 1e). These effects were similarly present for both symptom types (Anxiety: high symptom changes group: df = 22, $T = 2.437$; $p = 0.023$; low symptom changes group: df = 24, $T = -0.895$; $p = 0.379$; high vs. low symptom changes groups: df = 46; $T = 2.431$; $p = 0.019$; Depression: high symptom changes group: df = 20; $T = 2.21$; $p = 0.039$; low symptom changes group: df = 26; $T = -0.611$; $p = 0.546$; high vs. low symptom changes groups: df = 46; $T = 2.154$; $p = 0.037$) and types of LC-NE mask choice (see Figure S8 for comparison between masks). Thus, the results of this analysis suggest that the LC is involved in response conflict adaptation, specifically for people who go on to develop stronger subsequent psychopathological symptoms. This validates our measure and suggests that it may be useful for predicting the development of stress-related psychopathology.

To formally establish this predictive validity of conflict-induced LC-NE responsivity for stress resilience, we correlated the participants' symptom severity changes at 3 and 6 months (Fig. 2)

with their individual fMRI-BOLD-amplitude during conflict-induced upregulation (CI > II) in the locus coeruleus (extracted from physiological noise corrected, unsmoothed data with weighted averaging across voxels in the LC-1SD-mask[88]). Individual LC-NE responsivity indeed correlated significantly with anxiety- and depression score changes measured three and six months into the internship as well as with the mean symptom changes across 3 and 6 months (df = 47, $t_1$, anxiety: Rho = 0.30, $p = 0.018$, depression: Rho = 0.38, $p = 0.004$, $t_2$, anxiety: Rho = 0.31, $p = 0.002$, depression: Rho = 0.26, $p = 0.034$; mean between $t_1$ and $t_2$ anxiety: Rho = 0.30, $p = 0.002$, depression: Rho = 0.36, $p = 0.006$, non-parametric Spearman's rank correlation coefficient and robust regression, Fig. 3a–f). That is, smaller conflict responses in the LC-NE system to the CI > II contrast were associated with less anxiety and depression symptom change, and thus more resilience, during the subsequent internship[89].

To ensure predictive relevance and local specificity for the locus coeruleus, we compared symptom change predictions for anxiety and depression between analyses employing two types of LC masks (1SD and 2SD), as well as for analyses based on activity extracted from several other brainstem nuclei in the vicinity of the LC, i.e.: medial raphe nucleus (MR), dorsal raphe nucleus (DR), and ventral tegmental area (VTA)[89]. We also compared the predictive power of LC signals with that of signals extracted from the substantia nigra (SN), and the amygdala (please see supplemental methods details and Supplemental Figure S6 for a visualization of these brainstem structures). In addition, we tested whether these predictions hold for LC-extracted weighted averages from different analysis pipelines with or without spatial smoothing and physiological noise correction, respectively.

The additional results reveal that stress-related anxiety and depression symptom changes were most strongly associated with the locus coeruleus, compared to all other brainstem nuclei. They also underline the robustness of our results in several ways. For example, the choice of LC mask did not bias the results: The LC was the only structure associated with symptom changes, for both available types of standardized LC masks (1SD & 2SD; the smaller and more robust mask (1SD) yields the stronger correlations). Physio-correction generally improved the statistics in nuclei closest to the 4th ventricle, such as the LC, medial raphe, and dorsal raphe (in fact, in the physio-corrected smoothed data, DR and MR correlate with symptom changes as well). However, the physio-corrected unsmoothed data (as reported in detail above and illustrated in Fig. 3a–f) showed that the LC was the only region associated with both anxiety and depression changes, irrespective of LC-mask choice, suggesting that DR and MR correlations in the smoothed data may stem from a smearing of LC activity into these neighboring regions. These additional analyses further strengthen our main conclusion that stress resilience is associated with responsivity of the LC. We summarize the results of all these analyses in the Supplemental Table S6 (also see Supplemental Figure S4–S8 illustrating our additional analyses).

We also conducted a formal analysis of the temporal signal-to-noise ratio (tSNR) across the whole brain, and in particular in the brainstem, making it easier to assess the signal quality of the extracted LC signals in comparison with other brainstem structures. The tSNR was computed by dividing the mean of each time series by its standard deviation for each voxel in the brain. The results confirmed that both the average and subject-specific tSNR in the LC was well above standard cut-offs (>30). We also found that the signal in the LC was in fact strongest amongst all brainstem nuclei, for both standard LC masks (1SD & 2SD, see Supplemental Figs. S4–S6).

Next, we formally tested the predictive validity of the individual LC-NE upregulation response for symptom changes in the

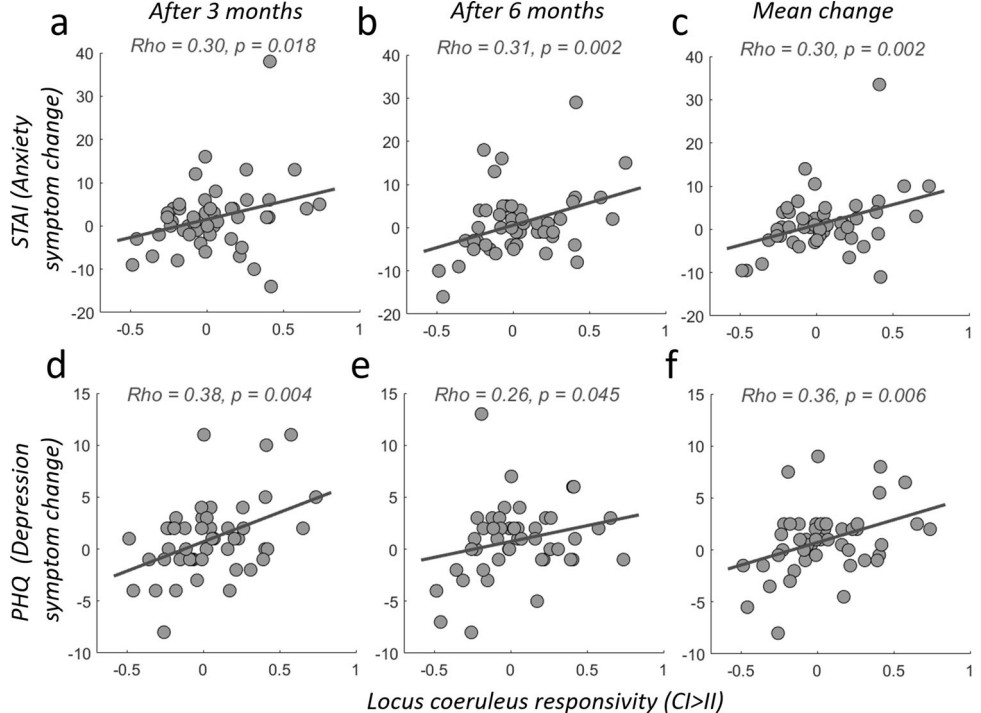

**Fig. 3 LC-NE responsivity (CI > II) relates to increases in anxiety and depression due to prolonged real-world stress exposure.** Each panel visualizes the correlation (robust linear regressions) between participants' symptom severity and individual CI > II responses, extracted from physiological-noise-corrected, unsmoothed data with weighted averaging across voxels in the LC-1SD-mask. Please note that the symptom severity used for these correlations is defined as the change from the individual symptoms baseline level at measurement time $t_0$. **a–c** Correlation between LC responsivity (CI > II) and severity of anxiety symptom changes (STAI) (top) measured after 3 months (**a**) and 6 months (**b**) of exposure to real-world chronic stress as well as the mean change between both measurement time points (**c**). **d–f** Same as (**a–c**), but for severity of depression symptoms change (PHQ). Source data are provided as a Source Data file.

population. We first compared the observed symptom severity change with the predicted change score in an out-of-sample fashion. To do so, we estimated a linear regression of psychological test score data on neural CI > II responses (weighted average LC-1SD extracted, physio-corrected, unsmoothed data) for the data of all participants excluding the current participant[90–92] and then used this fitted model to predict for the left-out participant the individual mean change in symptom severity. For simplicity, we focus on the mean symptom changes (mean across 3 and 6 months) in the remainder of the manuscript. A significant correlation in this out-of-sample procedure indicates that across the population, LC-NE responsivity can reliably predict individual stress resilience in the future[93]. We did observe such predictive validity: For both anxiety and depression, predicted symptom severity changes correlated with the observed symptom changes (df = 47, anxiety: Rho = 0.25, p = 0.01, depression: Rho = 0.28, p = 0.05, non-parametric Spearman's rank correlation coefficient and robust regression, Fig. 4a, c).

As a second step, we tested whether we can predict from the measure of LC-NE responsivity (weighted average LC-1SD extracted, physio-corrected, unsmoothed data) which out of two randomly chosen participants will be more resilient, i.e., incur a smaller symptom change after experiencing real-world stress. A leave-two-subjects out procedure (LTSO, see Methods) showed that the individual LC-NE upregulation response predicts above chance which subject developed higher anxiety symptom change (prediction accuracy 60.3%, p < 0.001, Fig. 4b). Similarly, LC-NE upregulation responses also predicted above chance which subjects developed higher depression symptom change due to real-life stress (prediction accuracy 59.4, p < 0.001, Fig. 4d). To compare between anxiety and depression predictions, and to

compare the predictive validity of different resilience predictors, we provide the receiver-operator characteristic curves (ROCs) and the associated area under the curve (AUC) plots (see below).

To establish the specificity of the LC-NE response for predicting symptom severity changes, we also tested other regions identified with the CI > II contrast such as the amygdala, dmPFC, vmPFC, and dlPFC. These regions (please see Supplemental Figure S9) either did not correlate at all with symptom changes (dmPFC, amygdala) or only for single time points or symptoms (3 months: vmPFC, anxiety: p = 0.048, depression: p = 0.023; depression at 3 months: dlPFC, p = 0.003). Moreover, to test the specificity of the upregulation response (CI > II) in predicting stress resilience, we explored whether conflict adaptation (II > CI) or mere conflict processing (I > C) responses in the LC revealed any significant correlations with symptom severity changes. Neither contrast revealed any significant cluster (all p > 0.05 FWE-corrected). Finally, no other regions of interest identified with these contrasts revealed any significantly correlation with symptom severity change (II > CI; left dlPFC or SMA, I > C; dmPFC, all p > 0.05, Supplemental Figure S10). This confirms the specificity of the LC-NE upregulation response as a biomarker for predicting future stress resilience in our sample (and comparable populations undergoing similar stressors).

**LC-NE amygdala functional coupling during upregulation is related to symptom changes.** Recent evidence from mouse models suggests that anxiety-like behavior is promoted by noradrenergic projections from the locus coeruleus to the amygdala[30,40]. We thus tested whether functional coupling between LC-NE and the amygdala during the upregulation response (CI > II) also relates to

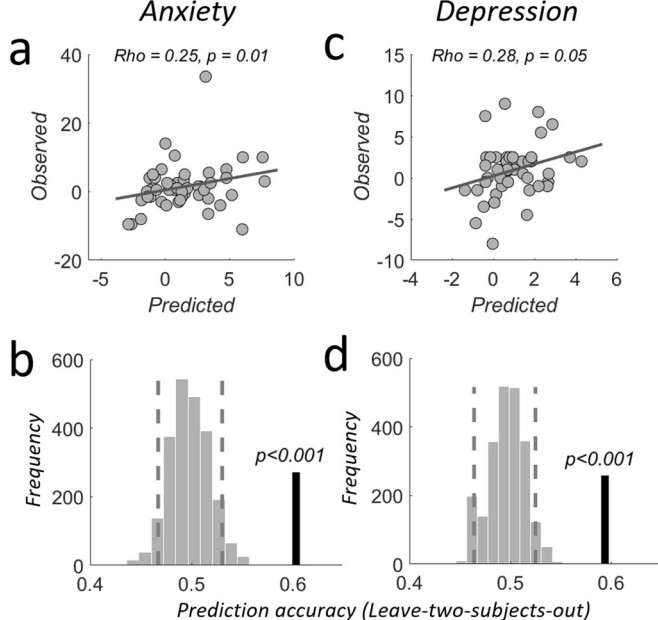

**Fig. 4 Out-of-sample prediction of mean symptom severity changes.**
**a** Correlation between out-of-sample predicted and observed mean anxiety symptom severity changes due to emergency room internship stress. Robust linear regressions. **b** LC-NE upregulation responses predict significantly above chance which of two subjects left out of the estimation will show stronger mean anxiety increases as a consequence of stress. Prediction analyses were based on a leave-two-subjects-out cross-validation procedure; their significance was tested using a permutation test with 1000 permutations for each possible left-out pair combination. Dashed lines indicate the 5th and 95th percentile of the randomized labels distribution, respectively. Thick black vertical line indicates the obtained prediction accuracy. (**c**, **d**) as in (**a**, **b**) but for depression symptom changes. Please see "Methods" section for details. Source data are provided as a Source Data file.

anxiety and even depression changes in humans. In a psychophysiological interaction analysis (PPI, Methods), an a priori region-of-interest in the amygdala showed functional coupling with the LC-NE during the upregulation response (CI > II), and the strength of this coupling correlated with the mean symptom severity changes for both anxiety and depression (df = 47, anxiety: P(SVC) < 0.001, X/Y/Z: -25/1/-23, T = 6.72, Z = 5.58; depression: P(SVC) < 0.001, X/Y/Z: 31/-2/-18, T = 3.45, Z = 3.24; Fig. 5). Note that correlations with symptom changes after either 3 or 6 months yields comparable results (please see Supplemental Figure S11, also for additional regions from exploratory whole-brain analysis), but we focus on the relationship with mean symptom changes for simplicity. This relationship was again specific to upregulation (CI > II), since the same analyses for conflict adaptation (II > CI) did not reveal a correlation with symptom severity change for LC-NE coupling with the amygdala (p > 0.05, small-volume-corrected) or other brain regions (p > 0.001, uncorrected). Likewise, no significant differences were observed when we analyzed coupling of LC-NE with the amygdala or other brain regions for differences between I > C trials or II > CI trials. Tests of predictive validity in the population (LTSO, see "Methods") again revealed that the individual LC-NE amygdala functional coupling during conflict response predicted above chance which of two new subjects developed stronger changes in anxiety (56.3%, p = 0.002) and depression (56.0%, p < 0.001) symptoms as a consequence of the real-life stressor (Fig. 5g, h).

Our results thus extend prior work in rodents showing that the functional coupling between LC-NE and amygdala is directly

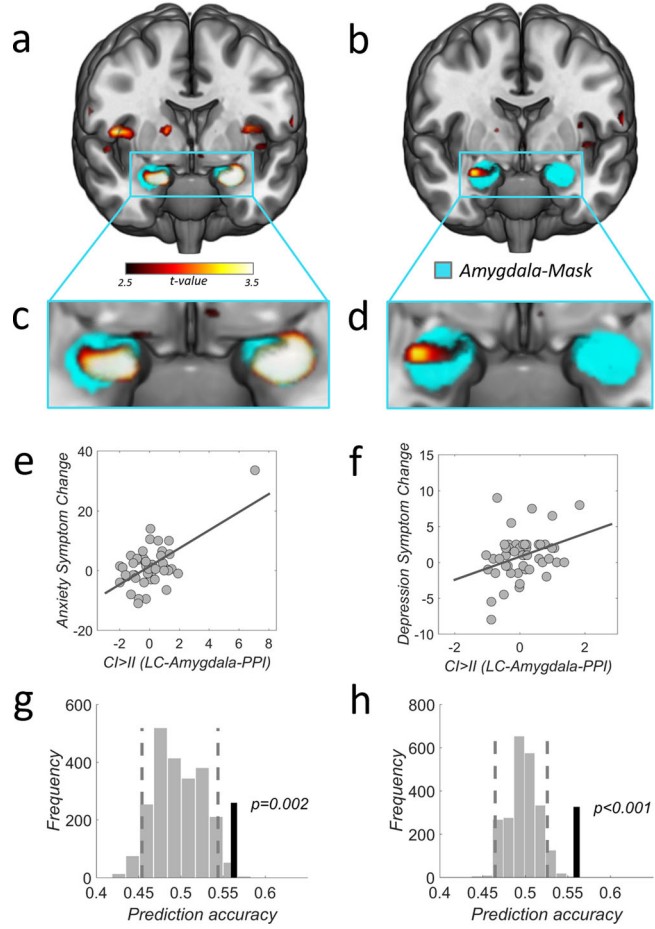

**Fig. 5 Functional coupling between LC-NE and amygdala during upregulation response relates to symptom changes.** Functional coupling between LC-NE and amygdala during the conflict response relates to individual changes in symptom severity for mean anxiety (**a**, **c**) and depression (**b**, **d**). **e** LC-NE-amygdala functional coupling (PPI: CI > II contrast betas extracted within bilateral amygdala mask using LOSO, see "Methods") relates to mean anxiety-symptom changes (R = 0.62, simple-regression: p < 0.001, robust regression, p = 0.052) and (**f**) mean depression-symptom changes (R = 0.34, simple-regression: p = 0.019, robust regression, p = 0.020). Mean symptom changes were defined as the mean between changes after 3 and 6 months. **g** LC-NE-amygdala functional coupling during upregulation response (extracted with LOSO) predicts mean anxiety symptom severity change above chance (p = 0.002, prediction accuracy 56.3%). **h** Similarly, for depression (p < 0.001, prediction accuracy 56.0%). Prediction analyses were based on a leave-two-subject-out cross-validation procedure; their significance was tested using a permutation test with 1000 permutations for each possible left out pair combination. Source data are provided as a Source Data file.

related to levels of fear and anxiety[30,40]. Given the well-known association of the amygdala with the perception of emotional intensity, fear, and threat in humans[36–39], our data may suggest that a pathological hyper-reactivity of noradrenergic LC-NE may enhance amygdala activity; this in turn may lead to elevated levels of fear and anxiety and eventually to stress-related psychpathology. Irrespective of this speculation, our data indicate that functional coupling of LC-NE with the amygdala during upregulation processes is a biomarker for predicting stress resilience, particularly with respect to anxiety symptoms To compare between anxiety and depression predictions, and to compare the predictive validity of different resilience predictors,

we provide ROC and the associated AUC plots (see below and Supplemental Figs. S12–S14).

**Conflict response and noradrenergic upregulation is reflected in pupil dilation.** Given the hypothesized link between pupil dilation and noradrenergic LC-NE firing[16,53,66], we investigated how pupil dilation related to emotional conflict responding and upregulation, and whether this index is useful for predicting stress resilience. An increasing number of studies have employed pupil dilation as an index of activity in the LC-NE system[94–98], and conflict-related pupil dilation has been observed in the classic Stroop task[99–102]. Here we extend these results to the emotional Stroop task: We found significantly enhanced pupil dilation for incongruent trials compared to congruent trials between 945 ms and 3668 ms post-stimulus onset ($p < 0.05$, one-sample t test, corrected for multiple comparisons using cluster-based permutation test[89], two-sided, df = 47, Fig. 6a, b, supplemental methods), demonstrating that the LC-NE contribution to conflict processing observed with fMRI is indeed also reflected in the pupil[101–103]. In keeping with our imaging analysis, we also observed trial sequence effects (CI > II) in the pupil signal: Pupil dilation for II trials yielded significantly enhanced pre-trial pupil dilation as compared to CI trials, between −3044 ms and −1222 ms prior trial onset ($p < 0.05$, one-sample t test, cluster-corrected, Fig. 6c). Interestingly, between 1530 and 4862 ms post-trial onset, we observed the opposite pattern of substantially lower pupil dilation for II compared to CI ($p < 0.05$, one-sample t test, cluster-corrected, Fig. 6d). These findings clearly indicate a (potentially noradrenergic) carry-over effect from previous- to current-conflict trials: Pupil signal was reduced on incongruent trials when these were preceded by an incongruent trial. These peripheral-physiological results further strengthen the evidence for the role of a putative noradrenergic mechanism in the conflict response and potentially in conflict adaptation.

Pupil dilation and constriction are typically observed in response to changes in ambient lighting. This well-documented light reflex response was reported to be driven by parasympathetic activity and opposes the influences of sympathetic arousal on the pupil[104–106]. We verified that conflict-related trial history effects on the pupil signal were not simply reflections of ambient light level differences between preceeding congruent and incongruent conditions. For this, we analyzed congruency effects on congruent trials without any conflict, during which any pupil differences would be attributed to differences in visual stimulation between preceeding trials (IC > CC, Fig. 6e,f). No significant pupil dilation differences were observed ($p > 0.05$, cluster-corrected), confirming that light reflex responses cannot explain the congruency-sequence effects in our pupil data.

**Pupil-related conflict response relates to LC responsivity and stress resilience.** Congruency-sequence effects in conflict tasks usually show that interference on a given trial is reduced if it is preceded by a conflict trial. Mechanistic interpretations of this effect suggest that conflict situations (such as incongruent stimuli) lead to arousal and noradrenalin release, facilitating conflict processing and resolution on the subsequent trial due to carry-over effects[68,73]. Such an account predicts that the strength of LC-NE activation on the preceeding incongruent trial should be inversely related to the LC-NE activity on the current incongruent trial, and reflected in pupil and in LC-BOLD responses. Both predictions were verified with our data: The increase in pre-trial pupil dilation for CI relative to II trials was negatively correlated with the subsequently-measured, stimulus-related CI > II pupil dilation difference (df = 47, $p = 0.00014$, $R = −0.52$, Fig. 7a) and the CI > II difference in LC-NE BOLD responses (df = 47,

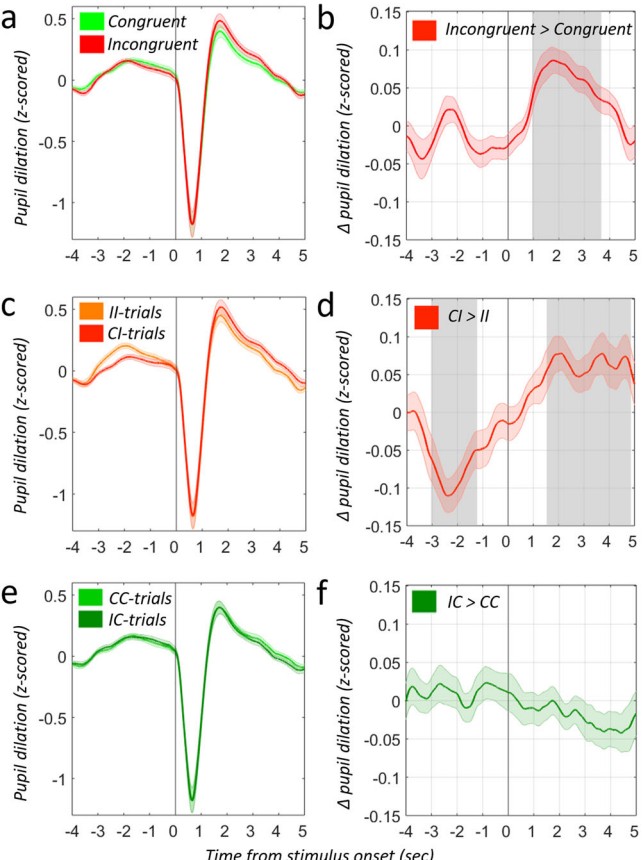

**Fig. 6 Pupil dilation during response conflict (I > C) and upregulation (CI > II). a** Mean Pupil dilation during congruent (green) and incongruent trials (red). **b** Pupil dilation during incongruent trials is significantly larger as compared to congruent trials. Gray shades area indicates sign. difference $p < 0.05$ (One-sample t test, two-sided, cluster-corrected). Vertical line indicates stimulus onset. **c** Mean Pupil dilation during CI trials (incongruent trials preceded by congruent trials = dark red) and II trials (incongruent trials preceded by incongruent trials = light orange). **d** II trials yield significantly enhanced pre-trial pupil dilation as compared to CI trials, indicating the need for noradrenergic upregulation on current CI trials. The gray shaded area indicates significant difference from zero at $p < 0.05$, (One-sample t test, two-sided, cluster-corrected). Vertical line indicates stimulus onset. **e** Mean Pupil dilation during CC trials (congruent trials preceded by congruent trials = light green) and IC trials (congruent trials preceded by incongruent trials = dark green). **f** Pupil dilation for CC trials does not significantly differ from IC trials before or during the current congruent trial, precluding potential light reflex differences (see main text). Error bands represent ±SEM at each timepoint. Source data are provided as a Source Data file.

$p = 0.038$, $R = −0.30$, Fig. 7b). To capture these trial-sequence effects in one pupil measure and relate it to individual symptom severity changes, we computed the pupil dilation distance (PDD) between current trial CI > II minus pre-trial CI > II and related it to mean anxiety and depression symptom severity changes. Even though we found a significant correlation between PDD and mean anxiety symptom changes ($p = 0.013$, $R = 0.36$), the out-of-sample predictions did not exceed chance level ($p = 0.09$, 52.6%). However, mean depression symptom changes were related to PDD ($p = 0.04$, $R = 0.30$) and also predicted out-of-sample above chance ($p < 0.001$, 55.3%). This suggests that pupil dilation measures alone can already be useful for predicting depression-related resilience in response to subsequent real-life stressors To compare between anxiety and depression predictions, and to

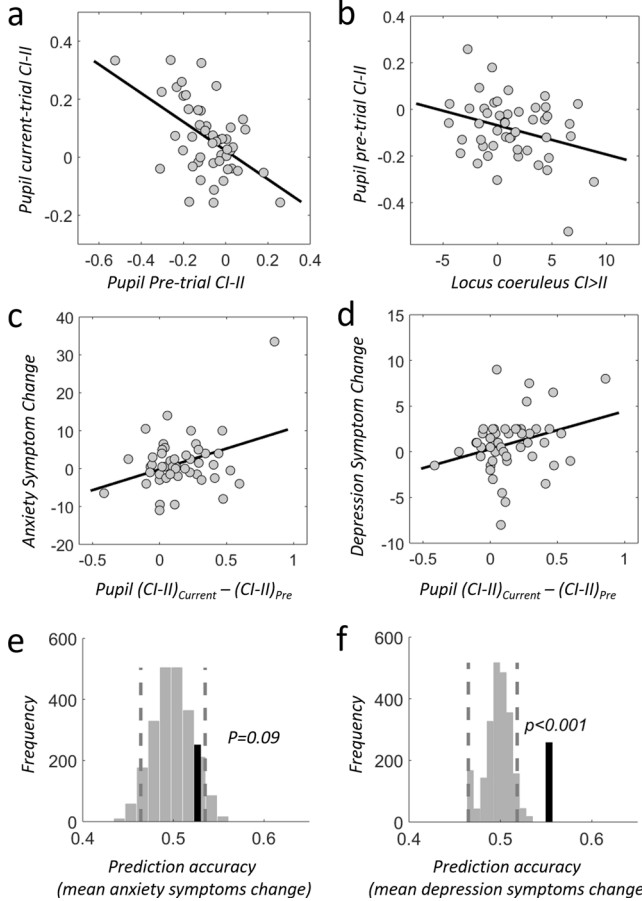

**Fig. 7 Conflict-related pupil response relates to LC responsivity and stress resilience. a** Evidence for carry-over effects of LC-NE responses as measured by pupil dilation: The stronger the pupil dilation before incongruent trials, the lower the pupil dilation elicited by this incongruent trial ($R = -0.52$, $p = 0.00014$); this effect is specific to incongruent trials. **b** The degree of pre-trial pupil-dilation difference (CI > II) correlates with LC-NE upregulation responses (CI-II), consistent with a noradrenergic mechanism that drives upregulation of resources to meet current trial conflict demands ($R = -0.30$, $p = 0.038$). **c, d** The larger the individual impact on pupil dilation from previous to current incongruent trials (pupil dilation distance between current CI-II and previous trial CI-II), the more increases are observed for (**c**) anxiety ($R = 0.36$, $p = 0.013$) and (**d**) depression ($R = 0.30$, $p = 0.04$) symptoms (for simplicity we averaged symptom changes between 3 and 6 months). **a–d** Pearson correlation. **e, f** The pupil dilation distance between current CI-II and previous trial CI-II predicts mean symptom changes due to real-world stress reliably out-of-sample for depression (**f**, $p < 0.001$), but only marginally for anxiety (**e**, $p = 0.09$). Prediction analyses were based on a leave-two-subject-out cross-validation procedure; their significance was tested using a permutation test with 1000 permutations for each possible left out pair combination. Source data are provided as a Source Data file.

compare the predictive validity of different resilience predictors, please see the ROCs and the associated AUC plots in Figs. 8, 9 as well as Supplemental Figs. S12–S14.

**Locus Coeruleus responsivity is a robust and reliable biomarker for stress resilience.** In a final analysis, we quantified and compared the usefulness of the identified biomarkers for predicting stress resilience by first comparing their predictive validity to that of a base-model (the current gold standard: self-report surveys of previous potentially traumatic experiences or current

symptoms) using a multiple GLM-approach. We also identified the most parsimonious parameter combinations for predicting individual anxiety or depression symptom change, by means of a stepwise-regression approach (Methods; for a comprehensive list of parameter test-statistics, goodness-of-fit measures, and model comparisons please see Supplemental Tables S7 and S8). Finally, we compared the out-of-sample prediction accuracy between the base model, full model (containing all parameters), and most parsimonious model using LTSO (Methods). Please note that the LC-specific regressor was extracted using the weighted average LC-1SD mask from the physio-corrected, unsmoothed fMRI data (Tables S7 and S8). For completeness, Supplemental Tables S9, S10 report the full list of statistics for data without these corrections.

These analyses showed that our identified biomarkers substantially improved predictions of anxiety symptom changes as compared to the gold-standard base-model. The adjusted explained variance was increased by 400 and 300%, respectively, when we added either LC ($p = 0.017$) or pupil ($p = 0.039$) to the regression. The classic behavioral congruency-sequence effect (CSE) was neither significant on its own ($p > 0.1$, model 2) nor in models containing either LC (model 3) or pupil (model 4). Having both LC and pupil regressors in one model explaining anxiety changes (model 5) further increased the explained adjusted variance (by about 20%); this model established LC ($p = 0.02$) and pupil ($p = 0.04$) as reliable predictors for anxiety changes. Importantly, adding the individual connectivity strength between LC and amygdala during the upregulation response (model 6) lead to another increase in adjusted explained variance (another 50%, resulting in approximately 12 times the variance explained by the base-model) and above-chance out-of-sample predictions ($p < 0.001$, 58.7%, Fig. 8b). These results thus establish both LC responsivity ($p = 0.038$) and LC-amygdala-connectivity ($p < 0.001$) during upregulation as important biological predictors for anxiety symptom changes and thus stress resilience. The usefulness of these variables was further underscored by the fact that the most parsimonious model contained LC-connectivity ($p < 0.001$), LC ($p = 0.025$), pupil ($p = 0.053$) and the behavioral CSE ($p < 0.031$). This model delivered the highest adjusted explained variance of 51.8% and predicted symptom severity change out-of sample ($p < 0.001$, 59.2%, Fig. 8c).

For depression symptom severity changes, the LC conflict response was also the most reliable predictor, even though the base model already explained 23.3% adjusted variance, primarily due to the PHQ-depression score at $T_0$ ($p = 0.0002$, model 1, see supplemental results for details). On top of this established measure, the individual LC upregulation response was the only biological marker that reliably related to depression symptom changes ($p = 0.046$), even when controlling for behavioral CSE ($p = 0.88$), pupil distance ($p = 0.14$), or LC-connectivity ($p = 0.74$). The LC upregulation regressor added 4% of the adjusted variance (27.1%, model 3) to that achieved by the base model; this was similar to the variance explained by the full model including all parameters (27.6%, model 6, with 64.4% out-of-sample accuracy Fig. 9b). LC ($p = 0.039$) and PHQ score at $T_0$ ($p = 0.0009$) were also the only two markers identified by the most parsimonious model, which explained 30.2% adjusted variance and significantly predicted mean symptom severity changes out-of-sample ($p < 0.001$, 67.7% accuracy, Fig. 9c). These results were also robust to non-prospective factors such as the number and severity of adverse events experienced during the internship (please see supplemental information for details and Supplemental Table S4 for comprehensive statistics).

Additional receiver operating characteristic (ROC) and AUC plots (Figs. 8, 9 and S12–S14) further facilitate the comparison between anxiety and depression predictions as well as between

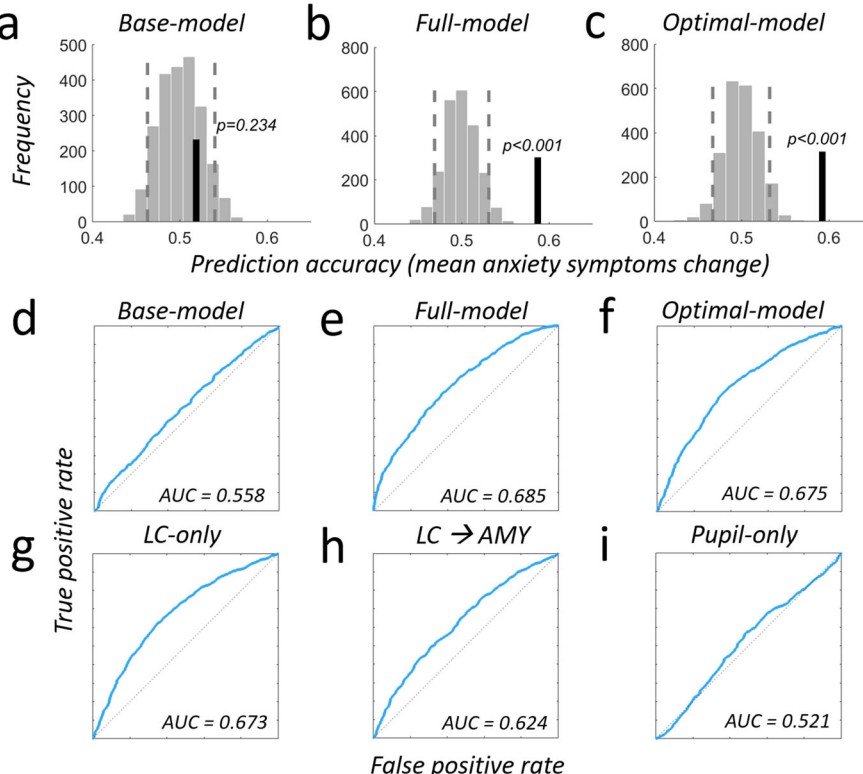

**Fig. 8 Comprehensive model comparison for predicting anxiety symptom change. a–c** Prediction analyses were based on a leave-two-subject-out cross-validation procedure and their significance was tested using a permutation test with 1000 permutations for each possible left-out pair combination. Light gray bars show the distribution of prediction accuracies that can be expected by chance (shuffled labels, see methods sections for details). Dashed vertical lines represent the 5th and 95th percentile of this distribution. Vertical black line indicates the obtained out-of sample accuracy. **a** A base-model containing scores from anxiety and pretrauma surveys does not predict the individual mean changes in anxiety symptom severity due to real-world stress above chance (out-of-sample accuracy = 51.86%, $p = 0.234$, $R^2 = 0.08$, adjusted $R^2 = 0.037$). **b** Using a full model that additionally contains behavioral-, neural- and pupil data predicts mean anxiety increases significantly above chance (out-of-sample accuracy = 58.7%, $p < 0.001$, $R^2 = 0.57$, adjusted $R^2 = 0.50$). Compared to the base-model, the full model increases the explained variance by 49% and the adjusted explained variance by 47%. Locus coeruleus contribution is significant ($p = 0.038$). **c** The optimal model, established using a stepwise-regression procedure (Methods), shows similar prediction improvements (out-of-sample accuracy = 59.2%, $p < 0.001$, $R^2 = 0.56$, adjusted $R^2 = 0.52$) but comprises only four parameters: locus coeruleus upregulation response ($p = 0.025$), behavioral congruency-sequence effect (CSE, $p = 0.031$), pupil ($p = 0.05$) and LC-NE-Amygdala coupling during the upregulation response ($p < 0.001$). Compared to the base-model, this sparse model predicts 49% more of the variance and also 48% more of the adjusted variance. **d–i** Receiver operating characteristic (ROC) plots and area under the curve (AUC) for different combinations of measures predicting anxiety: (**d**) Base-model, (**e**) Full-model, (**f**) Optimal model, (**g**) LC-only, (**h**) LC-Amygdala only, (**i**) pupil only. Please see Supplemental Table S7 for additional models, full details on single regressor contributions and model comparison. Source data are provided as a Source Data file.

different predictors. For instance, for prediction of both anxiety and depression, these plots show that the predictive power of LC-NE (Figs. 8g, 9g) exceeds that of pupil signals (Figs. 8i, 9i) as well as of LC-amygdala connectivity (Figs. 8h, 9h). While the predictive power of LC-NE for anxiety is fairly moderate (Fig. 8g), it clearly outperforms anxiety predictions from behavioral measures (Fig. 8d). Furthermore, LC-amygdala connectivity predictions for anxiety (Fig. 8h) are clearly stronger than for depression (Fig. 9h), while pupil dilation predictions for depression (Fig. 9i) outperform the ones for anxiety (Fig. 8i). Please see Supplemental Figs. S12–S14 for ROC plots and AUC quantification for all models tested. Taken together, both comprehensive regression analyses for anxiety and depression symptom change establish that noradrenergic LC responsivity constitutes a strong and reliable marker for prospectively predicting individual stress resilience in response to real-world stressors.

## Discussion

Stress resilience is conceptualized as adaptive, and presumably active, process rather than simply the absence of pathological

responses[26,107]. This key insight has recently led to a paradigm shift in resilience research away from disease-centered perspectives towards a health-focused agenda[50,108–110]. Despite growing interest in stress resilience in at-risk populations[6,7,111,112], it has so far been difficult to implement resilience trainings and monitor resilience indicators due to a lack of reliable indices and accurate assessment tools[113]. In particular, the precision of current assessments of stress resilience is compromised by the problems associated with meeting four methodological challenges.

First, individual assessments, particularly neurobiological assessments, of stress susceptibility have rarely been conducted before potentially traumatic stressors are experienced. It has thus been difficult to identify factors that genuinely predispose individuals to be resilient when they subsequently face adverse events. Identifying such predictors and potential resilience mechanisms is essential for the development of procedures to prevent the onset of stress-related psychopathology, such as anxiety and depression[52,113], in addition to informing potential treatments for when stress-related pathology has manifested. Second, potential resilience assessments have rarely been validated against the impact of real-world stressors. This is a crucial drawback, as

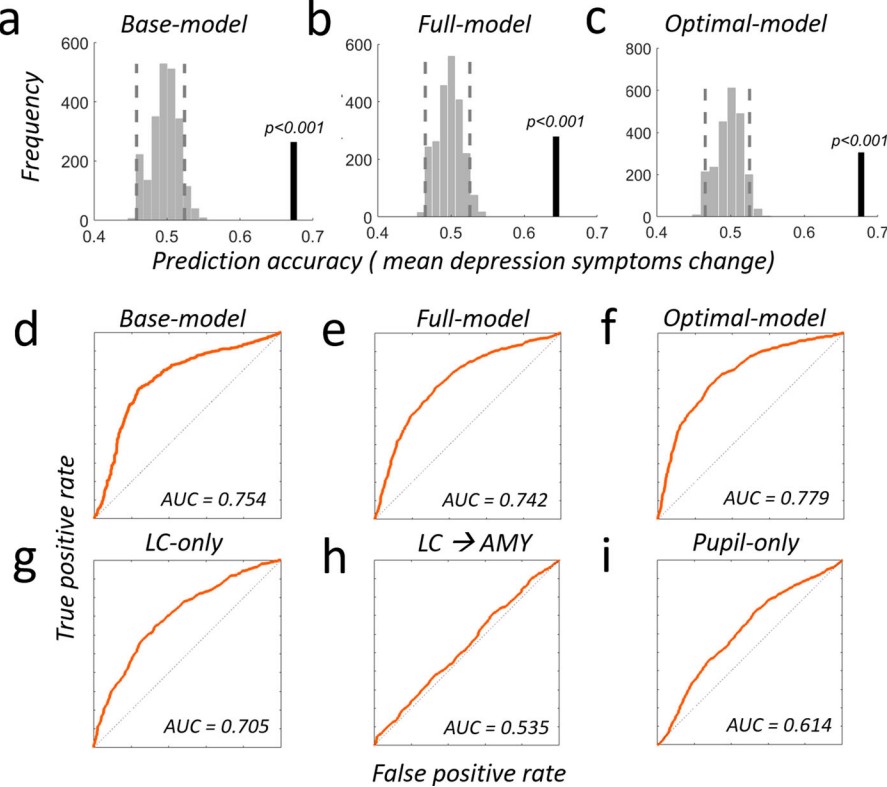

**Fig. 9 Comprehensive model comparison for predicting depression-symptom change. a–c** Prediction analyses were based on a leave-two-subject-out cross-validation procedure and their significance was tested using a permutation test with 1,000 permutations for each possible left-out pair combination. **a** The base-model predicts mean depression symptom increases significantly above chance (out-of-sample accuracy = 67.38%, $p < 0.001$, $R^2 = 0.27$ adjusted $R^2 = 0.23$). The PHQ-survey score is already a significant predictor for depression symptom severity changes ($p = 0.0002$). **b** The full model containing additional behavioral-, neural- and pupil data predicts mean depression increases significantly above chance (out-of-sample accuracy = 64.36%, $p < 0.001$, $R^2 = 0.37$, adjusted $R^2 = 0.28$) and increases the explained variance by 11% and the adjusted explained variance by 4.3%. **c** The optimal model has similar prediction improvements as the full model (out-of-sample accuracy = 67.7%, $p < 0.001$, $R^2 = 0.33$, adjusted $R^2 = 0.30$) but contains only two parameters: locus coeruleus upregulation response ($p = 0.039$) and the PHQ-depression survey ($p = 0.0009$). Compared to the base-model, this sparse model predicts 10% more of the variance and also 10% of the adjusted variance. **d–i** Receiver operating characteristic (ROC) plots and area under the curve (AUC) for different combinations of measures predicting depression: (**d**) Base-model, (**e**) Full-model, (**f**) Optimal model, (**g**) LC-only, (**h**) LC-Amygdala only, (**i**) pupil only. Please see Supplemental Table S8 for additional models, full details on single regressor contributions and model comparison. Source data are provided as a Source Data file.

previous stress experiments have primarily focused on the acute stress response without any consequences for real-life behavior[114,115]. Furthermore, the commonly employed laboratory stressors have multiple shortcomings—such as limited exposure time and low intensity—and thus lack real-world validity or predictability[56]. Third, resilience predictors have only rarely been cross-validated against independent data[81]. Fourth, only few predictors of resilience are grounded in experimentally-measured psychological and physiological mechanisms. This appears essential because assessments only based on self-report often have limited reliability and can be susceptible to self-reporting bias[113].

The current study met these challenges by identifying and cross-validating prospective biomarkers for resilience that fit these requirements. We show that greater activation of the LC-NE system during the upregulation response is associated with elevated symptoms of depression and anxiety after real-world stress, whereas lower conflict activation of the LC-NE system predicts resilience, i.e., the absence of future elevated symptoms. Furthermore, simultaneously-acquired measures of pupil-dilation, potentially associated with LC-NE firing[53], corroborate LC-responsivity as a reliable predictor for stress-related psychopathology and provide insights into the noradrenergic basis of conflict generation and adaptation.

Here we relate functional responses in the LC-NE system prospectively to stress resilience in humans. This link is not unexpected since the LC-NE system is a major component of the centrally-mediated fight-or-flight response and strongly activates as a result of various environmental stressors, including social and predator stress in rodents and non-human primates[22,24,80,116–120]. In addition to stress reactivity, the LC-NE system has been implicated in a large number of other physiological functions including arousal, memory, cognition, pain processing, and general behavioral flexibility, all of which may be mediated by its innervation of the entire neo-cortex through long-range noradrenergic projections[24,121–125]. Responsivity of LC-NE may thus play a prominent role in determining arousal state and environmental reactivity, making the LC-NE system an ideal neural hub for the facilitation of adaptive behavioral responses to stressors. Dysfunctions or hyperresponsiveness of the LC-NE system have indeed been implicated as a key factor for the development of a variety of pathophysiologic conditions, such as anorexia nervosa, obesity, PTSD, and related affective disorders in humans[25,46].

Medical professionals are exposed to stress in their jobs[76,77,79], requiring them to switch their own state to fit these contexts and to respond in adequate ways. In such high-stress environments, it is vital to keep arousal levels at bay to ensure patients' well-being and optimal outcomes[126]. The LC-NE has been associated with an

orienting and reorienting function that detects salient or behaviorally relevant stimuli and interrupts and resets ongoing activity[127–129]. A well-functioning LC-NE system that is not strongly taxed by this continuous challenge will have the capacity to optimally switch between behavioral relevant strategies in response to the changing demands. However, a maladaptive hyperactive LC-NE system may induce excessive arousal associated with unneccessary strategy switches or switches to task-irrelevant strategies; this in turn can violate own expectations as well as other peoples' and thereby cause further arousal and uncertainty that triggers anxiety and fear. This self-perpetuating cycle has been associated with increased LC-NE firing[65,130]. In fact, well-adapted behavioral and regulatory flexibility has recently been identified as a crucial psychological factor modulating anxiety susceptibility[131], which may in part be biologically predetermined by individual LC-NE responsity, as we show here.

Anatomical data in rats identified monosynaptic projections from the central nucleus of the amygdala to terminate in the LC[119]. More recent work in mice strongly implicated activity of the LC-NE system with stress-induced anxiety, by showing that selective inhibition of LC-NE neurons during stress prevents subsequent anxiety-like behavior[30]. In addition, it was demonstrated that photostimulation of LC projections to the basolateral amygdala releases NE. Moreover, this photostimulation evokes downstream modulation of neuronal activity in the amygdala, which resulted in anxiety-like and aversive behavioral responses in mice[40]. No such relationship has been demonstrated in humans. So far, human functional imaging studies associated the amygdala with emotional intensity as well as the perception of fear and threat[36–39]. Here we show that individual differences in functional coupling between LC-NE and amygdala prior to a real-world stressor is associated with the subsequent development of anxiety- but also depression-symptoms in humans, similar to documented mechanisms in mice[30,40,119]. This suggests a putative pathway by which LC-NE hyperreactivity may lead to the emergence of psychopathology: Perhaps excessive LC-NE influences on the amygdala may induce elevated levels of threat perception or emotional intensity, which in turn lead to enhanced anxiety and potentially related depression symptoms? Thus, our data imply not only LC-NE hyper-responsivity but also LC-NE coupling to the amygdala as a potential focus for intervention in preventing stress-related affective disorders.

Our results have potential implications for clinical and prevention science. Providing successful replication and extension to other samples, the LC-NE upregulation response, LC-connectivity, and potentially pupil dilation could be indexed to predict individual levels of resilience in stress-affected populations (e.g., policemen, soldiers, firemen, etc.). This may ensure that individuals at risk (i.e., high LC-NE responsivity predicting critical levels of symptom severity changes) are offered intervention or additional prevention approaches. It is conceivable that targeted change could be implemented by means of training functional response regulation or by neurofeedback in combination with real-time fMRI targeting the LC-NE[132]. Indeed, promising findings have recently indicated that emotion regulation can be trained via neurofeedback[133] and that connectivity measures are particularly suitable for this methodology[134,135]. Our finding that enhanced LC-amygdala functional coupling strongly relates to elevated anxiety symptom changes renders this connectivity measure an encouraging target for future prevention and intervention efforts.

Our study is not without limitations. Three major points should be taken into consideration. First, optimal functional imaging of the brainstem, and in particular the LC, is difficult due to the small size of the nuclei, their proximity to the ventricles, and inherently low signal-to-noise ratio in the brainstem. Identification of LC-NE

activity may thus benefit from more specialized data acquisition techniques than those applied here[85–87,136,137]. For instance, on the acquisition side, one may obtain high-field imaging for superior signal-to-noise ratio, high resolution T2-weighted anatomical imaging for accurate brainstem registration to standard space, neuromelanin-sensitive imaging for localization and dissociation of individual participants' LC from other brainstem nuclei. In addition, partial functional brain coverage could achieve particularly small (submillimeter) voxel resolution with high tSNR to avoid partial voluming effects and to counteract pulsating artifacts in the CSF and adjacent 4th ventricle. Pulsation could be quantified and mitigated with continuously recorded electrocardiogram. While future studies could use such specialized technology to focus on the LC with more certainty, in the present study, we applied more routine MR imaging protocols that can be replicated in numerous research settings worldwide. However, local specificity and physiological noise reduction can also be optimized via specialized analysis techniques, which we did apply here. For instance, we controlled for physiological noise by including principle components of the time-course in the CSF as nuisance regressors in the GLM analysis[138]. Moreover, to enhance the regional specificity of our results, we also analyzed activity from LC-adjacent brainstem nuclei and by repeating our initial analyses for unsmoothed data from a smaller LC mask (1SD vs. 2SD). Applying all these techniques to our data in fact enhanced the predictive accuracy of LC activity for symptom changes and showed that data from other brainstem nuclei do not allow this prediction. Thus, despite our use of more standard imaging protocols, our data provide evidence that specifically LC activation in response to emotional conflict predicts symptom changes in response to real-life stress (for detailed description of the applied methods and results please see supplemental methods, Tables S6–S8 and Figs. S4–S8).

Second, even though pupil dilation has primarily been associated with LC-firing and the noradrenergic system, recent evidence has identified also a cholinergic component[54,139]. Pupil dilation is thus hardly a specific noradrenergic marker but may reflect activity in multiple neuro-modulatory systems simultaneously. Dissociating the contributions of various neuro-modulatory systems to pupil dilation should be the focus of future human imaging studies. Nevertheless, because current eye-tracker systems can be portable, easy-to-use and inexpensive, our data suggest that pupil dilation can be very useful as a diagnostic-, monitoring- and treatment-tool not only for stress-related psychopathology[71,140–142] but also for studying the aging population and children.

Third, we investigated a modestly-sized medical student sample. While this sample is representative for the student population at Swiss medical schools, our results may not easily be generalizable to other populations across the world. For example, depression and anxiety levels were generally low in our sample, lower than in some other studies[79], hence indicating potential differences in selection or stress exposure. Working hours, in particular, tend to be lower in the Swiss medical internship system compared to the US[143], leading to potentially milder levels of stress exposure and symptom levels in the current cohort compared to medical residents in the US (see Supplemental Table S2–S3). In spite of such differences, there is clear evidence that Swiss medical internships are associated with occupational stress and vulnerability to psychopathology, as evidenced by a worsening of physical and psychological well-being and life satisfaction after the first year of internship/residency compared to before[144] and by the presence of relevant anxiety symptoms in 30% of residence physicians[145].

Furthermore, while maintenance of mental health and the absence or low levels of depression and anxiety can be conceptualized as resilient responding, resilience is potentially more

complex. Future studies may benefit from including further components of this concept, for example, variability over time and individual coping strategies, as well as investigating trajectories of change using latent growth mixture models[146]. Finally, it would be useful to repeatedly assess LC-NE responses to test whether those who are resilient (i.e., who remain healthy despite exposure to adversity) indeed show lower LC responsivity not just prior to but also during the real-life stressor.

In conclusion, our human prospective study of resilient responding identified laboratory measures of neurobiological mechanisms (LC-NE activation and connectivity during upregulation) that relate to future adaptive mental health outcome and resilience in response to real-life stressors. Our findings help to elucidate the neural basis of stress resilience in humans, inform prevention and intervention science and hold considerable potential for better diagnosis and treatment of stress-related psychopathology.

## Methods

**Participants**. We recruited medical students prior to their first medical internship. From the cohort of 200 students, 96 expressed an interest to participate. Following standard exclusion criteria (fMRI safety, psychopathology or attendance failure, see Supplement for details), the final sample consisted of 48 medical students ($n = 28$ women, mean age = 24 years, SD = 1.99). Participation was voluntary, and participants provided written informed consent.

**Study design and procedure**. The study was carried out over a period of 6 months. Just before commencing their internship ($t_0$), participants took part in an fMRI session that included the conflict adaptation task and several questionnaires. Three ($t_1$) and 6 months ($t_2$) after starting their internship, participants completed questionnaires assessing wellbeing, depression, anxiety and other variables (see below). After the study, participants were debriefed and compensated for their participation (US\$ 35 per hour). All procedures were approved by the Cantonal Ethics Committee of Zurich (KEK).

**Self-report questionnaires**. At baseline, participants completed questionnaires about demographic and clinical information (Supplemental Table S5) as well as an adapted version of the Trauma Checklist derived from the Posttraumatic Diagnostic Scale (PDS)[147], which was used to index the number of previously experienced potentially traumatic events (pretrauma-score) prior to the internship.

Anxiety symptoms[148] were assessed with the State-Trait Anxiety Inventory (STAI)[149] and depression symptoms with the Patient Health Questionnaire (PHQ-9)[150]. Both anxiety and depression levels were indexed at all three time points ($t_0$, $t_1$, $t_2$) to investigate potential changes in symptom-severity across time (see Supplement for details on questionnaires and internal consistency). Change scores were computed by subtracting baseline at $t_0$ from mid-internship questionnaire scores ($t_1$ and $t_2$), thus indexing change in anxiety and depression from baseline to mid-internship. These indices were then regressed on upregulation responses in the LC-NE arousal system (see below). As expected, we found considerable interindividual variation in depression and anxiety symptoms across participants, hence indicating different susceptibility to stress and resilience.

Non-prospective data were acquired to quantify the exposure to stressful experiences during the internship by asking subjects whether they had experienced stressful events at work by indicating exposure to patient death, invasive treatment, attending to grieving relatives, agitated relatives, grave treatment errors, or other events (detailed description in the Supplemental methods section).

## Data analysis

*Behavioral analyses*. To interrogate behavioral conflict responding, we used multiple linear (RT) and logistic (accuracy) regressions onto the following trial-wise predictor variables: current trial congruency, previous trial congruency, the interaction of current and previous trial congruency, and current-trial emotional valence. Each regression model was fitted independently for each subject; the resulting parameter estimates were standardized and their deviance from 0 was estimated with a two-sided t test. Statistical behavioral analyses were performed using the glmfit-, ttest-, and corr-functions implemented in the statistics toolbox in matlab version 2017a (see Supplement for further information).

*fMRI data analysis*. We used standard fMRI data-analysis procedures and estimated a general linear model (GLM) with SPM8 (Wellcome Trust Centre for Neuroimaging) to identify regions associated with upregulation processes, defined as the response difference between CI and II trials. This contrast was used to test whether upregultion responses in the LC-NE arousal system predict individual stress resilience. Eye movements and blinks were added as nuisance regressors[151]. The GLM regressed the blood oxygen level-dependent (BOLD) signal in each voxel

on these regressors and a set of hyperparameters modeling MR image auto-correlations with a first-order autoregressive model (see Supplement for full details).

*Psychophysiological analysis (PPI)*. We added to our GLM design matrix the BOLD time series extracted from a 5 mm sphere centered on the subject-specific LC peak in the CI > II contrast, determined by LOSO (see below). We also added two interaction terms corresponding to the interactions of the extracted BOLD time-course and the CI and II regressors. The difference in functional coupling during upregulation (CI > II) was then assessed in the amygdala and also utilized as stress-resilience predictor.

*Leave-one-subject-out (LOSO) procedure*. This method derives an unbiased prediction score, as each participant's data is extracted from a sphere (5 mm radius) around the group-peak coordinate in the data of all other participants, excluding the data from the current participant[90,91]. For each LOSO analysis, the peak was determined by the CI > II statistical contrast map within the region under study.

*Leave-two-subject-out (LTSO) procedure*. This method allows us to test how precisely a given model predicts which out of two randomly-drawn participants is more resilient, i.e., will develop lower symptom severity changes. We first generated all possible combinations of training- and test-sets: In each training set, we estimated a given model on the data of N-2 participants and predicted the symptom severity change for the two left-out participants. We then compared the predicted to the true change score for each of the 2256 possible left-out pairs and determined the prediction accuracy by calculating the percent correct predictions across all left-out pair combinations. In order to quantify how often this accuracy would occur by chance, we generated a null distribution of prediction accuracies. To this end, we repeated the model fit and prediction on shuffled symptom severity labels for each left-out pair 1000 times and calculated the obtained accuracy for each pair combination. The reported p-value represents the probability that the observed accuracy (based on unshuffled data) occurred by chance. Specifically, the 95th percentile of the null distribution is the lower accuracy bound above which all accuracies exceed a 5% probability of having occurred by chance.

*Eye-tracking and pupil dilation*. Subjects' fixation patterns and pupil size were recorded at 250 Hz with an MR-compatible infrared EyeLink 1000 eye-tracker system (SR Research Ltd.) and the Eyelink toolbox version 1.6 (https://github.com/uzh/edf-converter). Epochs of pupil signal loss were corrected via linear interpolation. Pupil time series were high-pass-filtered with a low cut-off of 0.05 Hz followed by a low-pass-filter with a high cut-off of 4 Hz. Each run-wise pupil time series was z-scored. Trial-wise pupil data were extracted ±5 s from stimulus onset and averaged per subject according to conditions of interest (congruent, incongruent, CI, II, CC, IC, Fig. 6a,c,e). Relevant contrasts (incongruent > congruent, CI > II, IC > CC) were computed for each participant. Time periods of significant difference from zero ($p < 0.05$) for pre-trial and current-trial pupil across all participants and contrasts were identified via one-sample t tests versus zero (Fig. 6b,d, f) and corrected for multiple comparisons using a cluster-based permutation test (Supplemental Methods). For each participant, the CI > II pupil dilation within the significant time intervals (gray shading Fig. 6f) was averaged to derive one score per subject for pre-trial (−3044 to −1222ms) and current-trial (1530 to 4862 ms) pupil dilation difference.

To assess the impact of pre-trial pupil dilation difference onto current-trial difference, and to estimate its relationship with LC-NE conflict upregulation, pre-trial pupil difference was correlated with current-trial pupil difference and LC-NE upregulation (CI > II) (Fig. 7a,b). To derive a subject-specific pupil dilation measure that quantifies the noradrenergic impact of pre-trial pupil difference onto current-trial pupil difference, a distance score was computed (Pupil dilation distance, PDD = $(CI-II)_{Current} - (CI-II)_{Pre}$, within above-defined time periods). This physiological marker used as predictor of mean symptom severity changes in multiple regressions and out-of-sample procedures (LTSO).

Quantifying physiological markers in stress resilience prediction. In order to quantify whether the identified physiological markers are useful for predicting prospective mean symptom severity change over and above the current gold standard for such predictions, we first assessed the predictive power of a base model (GLM, matlab function fitglm) containing only the scores from the traditionally used clinical resources, i.e.: respective symptom survey at $T_0$ and preTrauma-survey (model 1 in Supplemental Table S7 for anxiety and Table S8 for depression). We obtained several measures of model quality, such as variance explained, adjusted variance explained, Akaike Information Criterion (AIC), Bayesian Information Criterion (BIC). We also assessed whether adding our behavioral and physiological variables of interest (classic RT congruency-sequence effect (CSE) score, LC-responsivity, Pupil-dilation-distance (PDD), and LC-Amygdala functional coupling) to increasingly complex models significantly added to the model fit and/or improve goodness-of-fit measures (models 2-6 in Tables S7 and S8). Moreover, we quantified whether each model predicted mean symptom severity changes significantly above chance in an out-of-sample fashion using LTSO (see above). In addition, we quantified how much more variance and adjusted variance can be explained over and above the base model. Finally, we used a stepwise-regression approach to obtain the most parsimonious model that trades-

off model complexity with goodness-of-fit (model 7, implemented as stepwiseglm in matlab). The procedure starts by adding predictor variables to the constant model, as long as the value for the test statistic, deviance (the differences in the deviances between the models, tested via chi-squared test), is less than the default threshold value 0.05. If this criterion is not met the current term is removed. This procedure continues until all terms have been tested. Stepwise-regression balances the cost of poor fit due to too few model terms, with the chance of overfitting caused by too many model terms.

*Optimizing brainstem signal analysis.* Imaging the brainstem, and in particular the LC, is difficult due to the small size of the nuclei involved, their proximity to the ventricles, and inherently low signal-to-noise ratio in the brainstem. We conducted several analyses addressing physiological noise correction, weighted average data extraction, local specificity for the locus coeruleus, by comparing anxiety and depression symptom change predictions based on liberal and conservative LC masks (LC-1SD and LC-2SD) and several other brainstem nuclei.

We controlled for physiological noise with nuisance regressors that reflected the time-course within the cerebrospinal fluid (CSF)[138]. The CSF mask was generated for each individual by the non-linear unified segment procedure in SPM12 (see above). Time series were extracted for all voxels included in this mask and were submitted to principle component analysis using the matlab function pca.m included in the statistics toolbox (MATLAB, The MathWorks, Inc., Natick, Massachusetts, U.S., version 2017a). The first five principal components for each participant were used as nuisance regressors in the GLM analysis, alongside 6 motion regressors. Supplemental Fig. S7 provides a visualization for the first 2 participants and Supplemental Table S6 reports the improved statistical results for predicting symptom severity changes following the application of this technique.

We ensured predictive relevance and local specificity for the locus coeruleus by comparing anxiety and depression symptom change predictions based on both LC masks (1SD and 2SD) and several other brainstem nuclei. In addition, we employed a weighted-average data extraction that weighed every voxel's activity with the probability of membership in the ROI assigned to each voxel. These probabilistic maps included the main brainstem nuclei in the vicinity of the LC, i.e.: medial raphe nucleus (MR), dorsal raphe nucleus (DR), and ventral tegmental area (VTA) provided by the Harvard Ascending Arousal atlas available at https://www.martinos.org/resources/aan-atlas. We also compared LC prediction power with the substantia nigra (SN), available at https://www.nitrc.org/projects/atag/ and the amygdala (https://fsl.fmrib.ox.ac.uk/fsl/fslwiki/Atlases). In addition, we repeated all GLMs also for unsmoothed data, since the 6 mm smoothing kernel we applied may have smeared the activity between brainstem nuclei as well as with adjacent CSF. Please note that all LC-specific results reported in the main text are based on a weighted average of LC-1SD mask voxels extracted from the physiological-noise-corrected, unsmoothed fMRI data. Applying these optimized brainstem signal analysis techniques substantially enhanced the data quality and statistical significance of our results. Thus, even though our use of a more standard imaging protocol with normal rather than high spatial resolution requires the appropriate caution when assigning activations to the small LC, our extensive set of control analyses substantiated our conclusions that anxiety and depression symptom severity are best predicted by the responsivity of the human LC-NE arousal system.

**Data visualization.** MRIcroGL (https://www.mccauslandcenter.sc.edu/mricrogl/home/) was used for brain visualizations.

**Reporting summary.** Further information on research design is available in the Nature Research Reporting Summary linked to this article.

## Data availability
Data supporting the findings of this study are available at https://github.com/mgrues/LC_Stress_GitHubRepository (https://doi.org/10.5281/zenodo.4298505). A reporting summary for this Article is available as a Supplementary Information file. Source data are provided with this paper.

## Code availability
Matlab Code is available at https://github.com/mgrues/LC_Stress_GitHubRepository (https://doi.org/10.5281/zenodo.4298505).

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

## Acknowledgements

The authors would like to thank Karl Treiber for scanning assistance, Adrian Etter and Marc Biedermann for support with the eye-tracking setup and Gilles de Hollander for invaluable insight regarding brainstem imaging analyses. This research was funded by grants awarded by the Swiss National Science Foundation to B.K. (PZ00P1_126597, PZ00P1_150812) and C.C.R. (105314_152891). M.G. was funded by a grant awarded by the Richard-Büchner-Foundation (F-33153-02-01).

## Author contributions

M.G., H.T., C.C.R., and B.K., conceived of the project. M.G., H.T., C.C.R., and B.K. designed the study. U.E., J.B., M.B., A.E., R.B. provided resources. M.G., N.S., and H.T. collected the data. M.G. performed all analyses. M.G., C.C.R., and B.K. wrote the manuscript. All authors discussed the results and contributed to the final manuscript.

## Competing interests

The authors declare no competing interests.
