## [Peer Review File · Nature Communications]

Reviewers' comments:

Reviewer #1 (Remarks to the Author):

This article by Grueschow et al used functional MRI to link activity of the Locus Coeruleus (LC) activity to a prospective measure of stress resilience in a population of medical students. This was done in combination with behavioral measures and pupil dilation. Participants performed an emotional stroop task during MRI scanning, with facial expressions and congruent or incongruent words. Following this, participants filled in surveys for measures of depression and anxiety at three time points throughout their medical internship.

The authors looked at differences in LC activity in congruent-incongruent (CI) vs incongruent-incongruent (II) trial pairs, citing a more taxing upregulation process in the former contrast. Differences in these contrasts were linked to the temporal changes in depression and anxiety scores of participants throughout their medical internship. In addition to LC activity, measures of pupil dilation (a proxy measure of NE activity) was also compared to the neuroimaging contrast as well as the depression and anxiety scores.

LC activity in the CI>II contrast was significantly predictive of future depression and anxiety symptoms. LC activation correlated with pupil dilation; however, pupil dilation was only predictive of future depression scores, and did not predict not anxiety above chance level. Finally, the experimenters compared their LC models to available gold standard prediction models, and found them to improve symptom predictions. Overall, the paper is very well written, the methods are sound and clear, and the conclusions are well supported by the data. Below are (mainly minor) comments per section of the paper.

Introduction:

- The introduction has a smooth flow, introduces the topic and the rationale behind the study well.
- In line 54, the statement is a bit contentious. Who is to say to modern stressors outweigh “old” stressors like running away from predators?
- Line 116. The metaphor is a bit far-fetched.

Results:

- While the authors collected self-report data on depression and anxiety at t1 (3-month follow-up) and t2 (6-month follow-up), they do not report data regarding exposure to stressful experiences during the internship. It could be that some participants experienced more stress than others during the course of this study, and this may explain additional variance. The potential variability of stress exposure should be discussed.
- Was prior experience of stressful life events and/or trauma before the start of the study associated with the experimental measures (eg LC-NE function) and/or outcome measures?
- The LC is an extremely small structure. Given the spatial resolution of the EPI sequence used, and the limited precision of spatial normalization / LCmasks, the authors should be somewhat more cautious in attributing BOLD effects to the LC.
- Line 190-195. The authors use t-tests to assess changes in depression and anxiety from baseline to T1 and from baseline to T3. Why did the authors not use a repeated-measures ANOVA to assess changes across the three time points?

- Line 211. A specification as to what constituted the symptom severity change would be good to include here. Is it the slope? Or just mean change in depression score?
- The LTSO method was nicely used in this scenario, and improves the predictive validity of the model.
- Line 258. Typo.
- The analysis focusing on the predictive value of the LC measure is very interesting, and well rounded.

Discussion:

- The discussion appropriately covers the scope of the findings of the paper. Links to future directions are pointed out, and application/societal relevance of the findings are briefly explored.
- The limitations are mentioned, but could be extended with some points mentioned above (e.g., limitations in localizing the LC, lack of stressor exposure measures).

Methods:

- Preprocessing and nuisance correction for the neuroimaging data could be more thorough (e.g. physiological noise correction, motion correction).

Reviewer #2 (Remarks to the Author):

Recommendation: Accept with major revisions

Grueschow et al performed a timely task-based fMRI study of locus coeruleus in the prediction of depression outcomes in medical students starting internship. The strengths of the study are the use of a rigorous task to elicit surprise-related neural activity in the locus coeruleus and pupillary dilations, and the convincing correlations they observe between neural activity in LC (or LC-amygdala coupling) and subsequent depressive symptoms. These are important findings, building off what is known of LC-amygdala function in anxiety and depression from animal models and previous fMRI studies. The use of a prospective study design is another key addition to the literature on this subject and I believe this study will be well-cited and important.

However, the central framing of the study has a significant flaw which should be dealt with prior to acceptance. The paper claims to predict resilience to stress from LC fMRI, however it is not at all clear how much stress the participants have undergone (see discussion below and Figure 2). One can see from Figure 2 and from the subsequent correlations of change in PHQ-9 depression scores, that individuals are essentially equally likely to get more and less depressed. This disagrees with previous literature cited on the impact of medical internship on depression, and also calls into question this central claim of the paper. If people are equally likely to experience increases or decreases in depression, it may be more accurate to describe this as natural variability in depression rather than resilience to stress.

Major issues:

1. Is the study truly measuring resilience to stress, or just natural variability in the development of depressive symptoms? The paper claims to be measuring resilience to a stressor (medical internship), and cites a number of papers (refs 68-71) looking at the first year of postgraduate medical residency (internship) in the development of depressive symptoms. I am concerned about whether this is the

correct framing for this study since a) the group increase in depression scores is very questionable (see below) and b) the authors do not adequately justify how medical internship and associated stress may compare between the Swiss and American systems. In reality, it may be more accurate to describe this a. In the referenced American medical internship studies PHQ-9 scores typically increase about 4 points (Sen et al 2010) during internship. However, in the present study (Figure 2C-D), median PHQ-9 scores do not increase at all during internship (Figure 2C), and mean PHQ-9 scores increase very slightly (Figure 2D) driven by a few subjects. As well, there are methodological issues with the statistics in Figure 2D: they appear to be doing T tests uncorrected for multiple comparisons with a p value close to 0.05. They show in Figure 2D but do not report the effect sizes for these changes (e.g., absolute change in PHQ-9). They do not report the change in the percentage of participants developing depression (this goes from negligible to 20-30% in Sen et al 2010). One can also see in the correlation to fMRI measures figures (i.e., Figure 5E-F) that individuals are roughly equally likely to be depressed pre- and post-exposure to internship. If this is true, the claims and text should match that. There is a broader implication of this observation for this paper. It appears that people with high levels of anxiety and depression prior to internship are likely to have higher levels at each timepoint. That being the case, the relationships presented in figure 3 probably also pertain to the baseline timepoint. If this were to be true then the assumption that the relationship holds as a consequence of internship-related stress is undermined.

b. The specific level of stress associated with medical internship may be country-specific. The authors cite a number of studies of American medical students and interns. Have these studies (e.g., Sen, S. et al 2010) been validated for the training environment in Switzerland? There are multiple issues that may underlie national differences in the degree of stress during medical internship: typical work hours, social environment, medical system differences. Another is that the period described here as "medical internship" may or may not exactly match up with the period in the Sen study. This is a limitation which can be addressed with more information on the medical intership system in Switzerland in Methods and in discussion on why greater increases in depression were not seen in the study.

c. If it is not possible to adequately justify the "stress resilience" narrative, the authors may want to remove references to stress resilience which is questionable, and write the paper as "prediction of depressive and anxiety symptoms from LC/pupil biomarkers." It would be reasonable to describe the medical internship period as a period in which stress is expected but in which the degree of stress experience here is mild. So those few excess individuals who develop depression here may be "stress susceptible."

2. Discussion of the relationship between pupil and norepinephrine. The manuscript repeatedly describes the pupil as a simple readout of noradrenergic state (e.g., "a well-established external marker of noradrenergic LC-NE firing"), and cites Joshi et al 2016 as the primary source for this. However, the Joshi et al paper actually finds distributed correlations between pupil and a variety of brain areas, and other regions precede pupil-neuron coupling in LC (i.e., anterior cingulate). Furthermore, other studies find substantial (possibly greater) correlation between acetylcholine derived from the basal forebrain and the pupil than norepinephrine (Reimer et al 2016; Nelson and Mooney 2016). It is important that the paper describe the uncertainty in the literature as to the true neuromodulatory correlates of the pupil biomarker, and not attribute it simply to norepinephrine.

3. Is the study studying an appropriate sample to have clinical relevance? In order to have clinical relevance, there have to be sufficient numbers of individuals who reach clinically meaningful symptom

severity. On the PHQ-9, the cutoff for moderate depression is 10. However, in figure 2C, there are only 3-4 individuals with this level of depression at each timepoint and, as noted earlier, the number does not change with internship. This means that the investigators are not studying an appropriate sample to draw clinical inferences about depression at any timepoint. With respect to anxiety, moderate-severe anxiety would be any score above 37 (38-80). In figure 2A, it is evident that there about 10 subjects with moderate-severe anxiety at each timepoint. Thus, the study does not have an adequate sample to address clinically meaningful questions related to anxiety. Thus, this is a study about “resilience” in relation only to mild levels of anxiety or depression and it is not relevant to clinical populations. Thus, the thrust of the discussion is not in keeping with the actual nature of the data/results.

Minor issues:

- a. Paper moves back and forth between the terms norepinephrine and noradrenalin. Would recommend consistency on this.
- b. I would like to see ROC curves in addition to the leave two-out prediction accuracy metrics used in Figure 4,5,7 with reporting of Area-under-curve as an additional metric for prediction accuracy for the models reported.
- c. The term trauma is used multiple times in the manuscript to refer to the period of medical internship. This goes far beyond the limitations of the stress of medical internship concern I discuss above. Trauma is usually a life-threatening event, which this is not. Would remove this term from the manuscript.
- d. Can the pupil be a biomarker alone (or in concert with epidemiological factors) in prediction subsequent depression and anxiety? Given that it is much more easily collected than the LC fMRI biomarker, I would like to see model performance for the pupil biomarker alone at predicting subsequent symptoms (AUC, variance explained) in the main text. This would also be a good thing to highlight in the discussion, as this may have a broader practical application in clinical psychiatry but is somewhat under-emphasized in the paper.

Reviewer #3 (Remarks to the Author):

Grueschow and colleagues examined if resilience to stressful events can be predicted from fMRI responses in the brainstem noradrenergic nucleus locus coeruleus (LC), and from pupil diameter, in healthy medical students performing an emotional stroop task. The manuscript is well written, and contains many interesting results. I also see the relevance and novelty of prospective studies such as this one, which aim to examine the factors that make a person susceptible to the effects of stress, instead of examining the effects of stress after the fact.

However, I have a number of major concerns (both conceptual and methodological) that leave me unconvinced that the conclusions in the manuscript are justified. My primary concerns are that the task is not well grounded in electrophysiological studies that have charted the responsivity of LC neurons, and that the imaging protocol is unsuitable for functional imaging of the brainstem (specifics below). Unfortunately, these concerns preclude me from recommending that this manuscript is suitable for

publication in this journal.

Major:

1) Task: There is little validation from basic animal research as to what the spiking properties of the LC are during this particular task, and if the LC should be responsible for trial sequence-related behavioral effects during this task in the first place. This means that validation analyses, where a stimulus that is known to elicit a strong spiking response is used to validate the BOLD signal response in the LC, are not possible.

The literature that is cited to corroborate the involvement of the LC-NE system in this task is all pupil-related. The pupil is indeed a proxy of activity of the LC-NE system, but hardly a specific one, as the pupil is also correlated with activity in other neuromodulatory nuclei (see for example Reimer et al., 2016, and de Gee et al., 2017) and sensitive to abstract concepts such as cognitive effort that do not solely reflect activity in the LC and likely differ between task conditions.

While there is indeed a long-standing literature that implicates phasic noradrenergic hyperreactivity in the development of stress-related disorders, the part of this literature that involves direct LC recordings mainly centers on altered LC discharge properties in response to stressful / noxious stimuli, which are known to elicit a strong LC response. The predictions and interpretation of the results in the current experiment would have been more straightforward if stimuli of this kind were used (or another task that reliably engages the LC such as an oddball task), compared to complex trial-sequence effects of which the neurophysiological underpinnings are less established.

2) Data acquisition and analysis: Proper functional imaging of the brainstem, and in particular the LC, is notoriously difficult due to the small size of the nuclei involved, their proximity to the ventricles, and inherently low signal-to-noise ratio in the brainstem. As such, it requires non-standard techniques for both data acquisition and analysis (see Eckert et al., 2010 for a list of recommendations, also see Brooks et al., 2013, Forstmann et al., 2017, and Turker et al., 2019 for further considerations). Unfortunately, none of these techniques have been applied here.

- i) The voxel size of the functional sequence was rather large due to whole-brain field of view, whereas sequences that are optimized for brainstem imaging typically reduce the voxel size (and / or TR) at the expense of the field of view.
- ii) No high resolution T2-weighted anatomical image for accurate registration to standard space of the brainstem was acquired.
- iii) No neuromelanin sensitive sequence was used for localization of individual participants' LC.
- iv) The "new segment" tool that was used for registration to standard space performs a rigid body transform, not non-linear transformation or warping, which is essential for an accurate registration of the brainstem. Ideally, one would implement a separate registration procedure for the brainstem alone.
- v) Spatial smoothing (6 mm FWHM) was applied, smearing activity between the LC, surrounding neuromodulatory nuclei, and the 4th ventricle.
- vi) A sphere was placed on the center of mass of the LC template rather than using a weighted mean of

the template itself, further blurring the effective functional extent of the LC mask.

vii) No retrospective image correction (regression-based or data driven ICA-based) to suppress physiological noise was applied. This is really essential if one wants to separate the 4th ventricle and LC, because the former pulsates with every heartbeat and is directly adjacent to the LC.

viii) The 4th ventricle signal was not used as a nuisance regressor.

ix) No formal tSNR analysis of the brainstem was included, and thus it is difficult for the reader to assess the signal quality of the extracted LC signals.

For a future submission, I strongly advise the authors to address as many of these points as possible given the data that were acquired, and thoroughly discuss the limitations where addressing them is not possible.

3) Results and interpretation: The reported peak coordinates for the contrast $CI > II$ are over a centimeter away from the nearest voxel in the Keren LC atlas, which at 2 SD is already quite liberal in spatial extent. Moreover, the cluster of significant voxels also includes other neuromodulatory nuclei such as the dorsal and median raphe, substantia nigra, and ventral tegmental area. Thus, the conclusions are far too specific in terms of which neuromodulatory nuclei (i.e. LC-NE) drive the correlations with outcome measures.

Given that no control analyses using other neuromodulatory nuclei were performed, the authors should substantially tone down claims and rephrase the manuscript in terms that are agnostic about which specific neuromodulator is involved (e.g. “ascending arousal systems” or equivalent non-specific term) instead of making specific claims about LC-NE.

Minor:

How is it possible that CC and IC trials show no difference in pupil diameter around -2, -3 seconds, when the inter trial interval is 3-4 seconds, and C and I trials do show a difference at +1 second?

Typo on page 13, line 258: “develspd”

What was the median and range of trials numbers included for analysis?

References:

- Brooks JC, Faull OK, Pattinson KT, & Jenkinson M. (2013). Physiological noise in brainstem FMRI. *Front. Hum. Neurosci.* 7:623.
- Eckert MA, Keren NI, & Aston-Jones G. (2010). Looking forward with the locus coeruleus. *Science* (e-letter).
- Forstmann BU, Hollander GD, Maanen LV, Alkemade A, & Keuken MC. (2017). Towards a mechanistic understanding of the human subcortex. *Nat. Rev. Neurosci.* 18, 57–65.
- de Gee JW, Colizoli O, Kloosterman NA, Knäpen T, Nieuwenhuis S, & Donner T. (2017). Dynamic

modulation of decision biases by brainstem arousal systems. *eLife* 6:e23232.

- Reimer J. et al. (2016) Pupil fluctuations track rapid changes in adrenergic and cholinergic activity in cortex. *Nat. Commun.* 7, 13289.

- Turker HB, Riley E, Luh W-E, Colcombe SJ, & Swallow KM. (2019) Estimates of locus coeruleus function with functional magnetic resonance imaging are influenced by localization approaches and the use of multi-echo data. *bioRxiv*.

Reviewer #1 (Remarks to the Author):

This article by Grueschow et al used functional MRI to link activity of the Locus Coeruleus (LC) activity to a prospective measure of stress resilience in a population of medical students. This was done in combination with behavioral measures and pupil dilation. Participants performed an emotional stroop task during MRI scanning, with facial expressions and congruent or incongruent words. Following this, participants filled in surveys for measures of depression and anxiety at three time points throughout their medical internship. The authors looked at differences in LC activity in congruent-incongruent (CI) vs incongruent-incongruent (II) trial pairs, citing a more taxing upregulation process in the former contrast. Differences in these contrasts were linked to the temporal changes in depression and anxiety scores of participants throughout their medical internship. In addition to LC activity, measures of pupil dilation (a proxy measure of NE activity) was also compared to the neuroimaging contrast as well as the depression and anxiety scores. LC activity in the CI>II contrast was significantly predictive of future depression and anxiety symptoms. LC activation correlated with pupil dilation; however, pupil dilation was only predictive of future depression scores, and did not predict not anxiety above chance level. Finally, the experimenters compared their LC models to available gold standard prediction models, and found them to improve symptom predictions. Overall, the paper is very well written, the methods are sound and clear, and the conclusions are well supported by the data. Below are (mainly minor) comments per section of the paper.

We very much appreciate the reviewer stating that our work is very well written, contains sound and clear methods, and leads to conclusions that are well supported by the data. We are also grateful for the minor points raised which we address in a point-by-point fashion below. In summary, we followed the reviewer’s suggestions to (1) control for actual events experienced during the internship and baseline level of psychopathology, (2) confirm the specificity of our findings to the LC by adding many control analyses and preprocessing steps of the imaging data, (3) include a comprehensive analysis of the behavioral data that controls for multiple comparisons, and (4) discuss limitations of the present study. All of these changes substantiate our initial findings and therefore strengthen our paper substantially. Thanks again to the reviewer for pointing us to these issues and for suggesting how to address them.

Our responses to the reviewer’s comments are marked in ‘bold’ and new text added to the manuscript or supplemental material is marked in ‘green’, also in the respective documents.

Introduction:

- The introduction has a smooth flow, introduces the topic and the rationale behind the study well.

Thank you!

- In line 54, the statement is a bit contentious. Who is to say to modern stressors outweigh “old” stressors like running away from predators?

We thank the reviewer for pointing this out to us. We have now removed ‘modern’ from the statement.

- Line 116. The metaphor is a bit far-fetched.

We have also removed this statement from the manuscript.

Results:

- While the authors collected self-report data on depression and anxiety at t1 (3-month follow-up) and t2 (6-month follow-up), they do not report data regarding exposure to stressful experiences during the internship. It could be that some participants experienced more stress than others during the course of this study, and this may explain additional variance. The potential variability of stress exposure should be discussed.

We thank the reviewer for bringing this point back to our attention. Since our manuscript aimed primarily at truly prospective prediction, we initially included only predictors that were available prior to commencement of the internship. However, during the internship, we did in fact acquire data about exposure to stressful experiences, by quantifying subjects' experience of patient death, invasive treatment, attending to grieving relatives, agitated relatives, grave treatment errors, or other events. This is now reported in the results section (p. 11 and 25) and described in more detail in the supplemental methods section (p. 3):

‘Adverse events data during internship.

We conducted a control analysis, testing how individual symptom severity changes may relate to the exposure to potentially stress-inducing events during the internship. We did this by having participants indicate how often one of the following adverse types of incidents occurred: death of a patient, a particularly invasive treatment, grieving or agitated relatives, a treatment during which the participant or a colleague made a severe mistake, and similarly adverse events (13 items, adapted for the present study from Weiss et al., 2010). The experienced severity of these events was individually quantified on a scale ranging from 1 (not stressful) to 4 (extremely stressful). In order to derive an individual final score for adverse events, we multiplied their number and severity for each participant and correlated this measure with observed individual symptom changes.

This revealed no significant relationship between stressful event exposure during the internship and mean depression symptom changes (Pearson-correlation $p=0.346$; $R=-0.139$). However, we did find that mean anxiety symptoms increased significantly with experienced adverse events (Pearson-correlation $p=0.006$; $R=-0.393$, see supplemental methods above for details). We tested whether this explained any additional variance above and beyond our original prospective predictors, by including adverse events in our original full model. This showed that adverse events did not explain any substantial variance (R^2) or adjusted variance (adj R^2), while both LC and LC-Amygdala connectivity remained strong predictors of anxiety symptom changes (See supplemental table 6). These findings also did not depend on the choice of LC-mask (1SD or 2SD).’

- Was prior experience of stressful life events and/or trauma before the start of the study associated with the experimental measures (eg LC-NE function) and/or outcome measures?

We thank the reviewer for this suggestion and analyzed this relationship. There was no such relationship. In the supplemental material section, we now write on p. 3:

‘The level of prior stressful life events correlated neither with LC-responsivity ($p=0.2640$, $R=0.1644$) nor with LC-Amygdala connectivity ($p=0.9820$, $R=0.0033$). In addition, there was also no significant correlation between number of previously experienced traumatic events and mean anxiety or depression symptom changes ($p=0.7671$, $R=-0.0439$). While such relationships might potentially be expected in patients suffering from anxiety and depression well above clinically-relevant thresholds, they may not be present in our specific cohort of well-adapted medical students with relatively small numbers of prior traumatic events (mean number of pre-trauma events: 1.1, range: 0-4).’

- The LC is an extremely small structure. Given the spatial resolution of the EPI sequence used, and the limited precision of spatial normalization / LCmasks, the authors should be somewhat more cautious in attributing BOLD effects to the LC.

We agree with the reviewer and have now conducted an extensive list of control analyses that substantially increase the local specificity and statistical strength of our results and that strengthen the support for our conclusions. These analyses include weighted averaging for region-of-interest data extraction, control for additional brainstem-nuclei, controlling for physiological nuisance variables based on principle component analysis of CSF probability tissues classes, as well as applying these nuisance variables in new regression models for smoothed and unsmoothed data. Moreover, we have now included a formal temporal-signal-to-noise (tSNR) analysis of the whole brain and specifically the brainstem, making it easier for the reader to assess the signal quality of the extracted LC signals and to compare it with other brainstem nuclei. The new control analyses and their results are described in detail in the supplemental material. In addition, we illustrate our control analyses with several new supplemental figures (S7-S11) and summarize the statistical results in new supplemental tables (S5-S6). All these additions to the manuscript show that our results are robust and very likely originate from the LC.

- Line 190-195. The authors use t-tests to assess changes in depression and anxiety from baseline to T1 and from baseline to T3. Why did the authors not use a repeated-measures ANOVA to assess changes across the three time points?

We thank the reviewer for pointing this out and have conducted the requested repeated-measures ANOVAs. In post-hoc t-tests, we then further examined symptom changes at 3 and 6 months to also assess the variability of the outcome-measure and the stability of the predictions. The results section on page 10 now reads:

‘As expected, group-level symptom severity for both psychological test scores increased over time due to the stressful medical internship (main effect of time for anxiety, $F=4.01$, $p=0.022$ and depression, $F=3.11$, $p=0.049$; repeated-measures ANOVA, controlling for gender and age, degrees of freedom=2). Post-hoc t-tests further identified symptom increases at 3 and 6 months relative to baseline level, albeit not always significantly (**Figure 2**) (*Depression*: 3 months: $T_{(1,47)}=1.83$, $p=0.037$, 6 months: $T_{(1,47)}=1.75$, $p=0.0448$, mean-change: $T_{(1,47)}=2.00$, $p=0.025$, *Anxiety*: 3 months: $T_{(1,47)}=1.62$, $p=0.056$, 6 months: $T_{(1,47)}=0.917$, $p=0.182$, mean-change: $T_{(1,47)}=1.44$, $p=0.078$, one-sample t-test, one-sided).’

- Line 211. A specification as to what constituted the symptom severity change would be good to include here. Is it the slope? Or just mean change in depression score?

Throughout the manuscript, symptom severity changes are defined as mean changes in depression/anxiety scores from the baseline score before the internship to 3 and 6 months into the internship. We now state this more clearly in the revised text. For instance, in the results section on page 12 we now write:

‘To establish the predictive validity of conflict-induced LC-NE responsivity for stress resilience, we correlated the participants’ mean symptom severity changes at 3 and 6 months relative to the symptom severity level prior to the internship (Figure 2), with their individual fMRI-BOLD-amplitude during conflict-induced upregulation (CI>II) in the locus coeruleus (LC-mask(Keren et al., 2009), Figure 1C-D).’

- The LTSO method was nicely used in this scenario and improves the predictive validity of the model.

We thank the reviewer for this positive evaluation.

- Line 258. Typo.

We changed ‘developsd’ to ‘developed’. Thank you.

- The analysis focusing on the predictive value of the LC measure is very interesting, and well rounded.

We very much appreciate the reviewers’ compliment.

Discussion:

- The discussion appropriately covers the scope of the findings of the paper. Links to future directions are pointed out, and application/societal relevance of the findings are briefly explored.

Thank you very much.

- The limitations are mentioned, but could be extended with some points mentioned above (e.g., limitations in localizing the LC, lack of stressor exposure measures).

We agree with the reviewer and have now extended this text substantially to discuss both the specificity of our results to the LC and the noradrenergic system as well as the specifics of the sample under study. This new text starting on page 31 now reads:

‘Our study is not without limitations. Three major points should be taken into consideration. First, optimal functional imaging of the brainstem, and in particular the LC, is difficult due to the small size of the nuclei, their proximity to the ventricles, and inherently low signal-to-noise ratio in the brainstem. Identification of LC-NE activity may thus benefit from more specialized data acquisition techniques than those applied here (Brooks et al., 2013, de Gee et al., 2017, Forstmann et al., 2017, Schumann et al., 2018, Turker et al., 2019). For instance, on the acquisition side, one may obtain high-field imaging for superior signal-to-noise ratio, high resolution T2-weighted anatomical imaging for accurate brainstem registration to standard space, neuromelanin-sensitive imaging for localization and dissociation of individual participants’ LC from other brainstem nuclei. In addition, partial functional brain coverage

could achieve particularly small (submillimeter) voxel resolution with high tSNR to avoid partial voluming effects and to counteract pulsating artefacts in the CSF and adjacent 4th ventricle. Pulsation could be quantified and mitigated with continuously recorded electrocardiogram. While future studies could use such specialized technology to focus on the LC with more certainty, in the present study, we applied more routine MR imaging protocols that can be replicated in numerous research settings worldwide. However, local specificity and physiological noise reduction can also be optimized via specialized analysis techniques, which we did apply here. For instance, we controlled for physiological noise by including principle components of the time-course in the CSF as nuisance regressors in the GLM-analysis (Bazin et al., 2019). Moreover, to enhance regional specificity of our results, we also analyzed activity from LC-adjacent brain-stem nuclei and by repeating our initial analyses for unsmoothed data from a smaller LC-mask (1SD vs. 2SD). Applying all these techniques to our data in fact enhanced the predictive accuracy of LC activity for symptom changes and showed that data from other brainstem nuclei do not allow this prediction. Thus, despite our use of more standard imaging protocols, our data provide evidence that specifically LC activation in response to emotional conflict predicts symptom changes in response to real-life stress (for detailed description of the applied methods and results please see supplemental methods, tables S5-S6 and figures S7-S10).

Second, even though pupil dilation has primarily been associated with LC-firing and the noradrenergic system, recent evidence has identified also a cholinergic component (Nelson and Mooney, 2016, Reimer et al., 2016). Pupil dilation is thus hardly a specific noradrenergic marker but may reflect activity in multiple neuro-modulatory systems simultaneously. Dissociating the contributions of various neuro-modulatory systems to pupil dilation should be the focus of future human imaging studies. Nevertheless, because current eye-tracker systems can be portable, easy-to-use and inexpensive, our data suggest that pupil dilatation can be very useful as a diagnostic-, monitoring- and treatment-tool not only for stress-related psychopathology (Reimer et al., 2014, Vinck et al., 2015, Warren et al., 2016, van der Wel and van Steenbergen, 2018) but also for studying the aging population and children.

Third, we investigated a modestly-sized medical student sample. While this sample is representative for the student population at Swiss medical schools, our results may not easily be generalizable to other populations across the world. For example, depression and anxiety levels were generally low in our sample, lower than in some other studies (Sen et al., 2010), hence indicating potential cross-cultural differences in selection or stress exposure. Working hours, in particular, tend to be lower in the Swiss Medical internship system compared to the US (Temple, 2014), leading to potentially milder levels of stress exposure and symptom levels in the current cohort compared to medical residents in the US (see supplemental table S7-S8). In spite of such differences, there is clear evidence that Swiss medical internships are associated with occupational stress and vulnerability to psychopathology, as evidenced by a worsening of physical and psychological well-being and life satisfaction after the first year of internship/residency compared to before (Buddeberg-Fischer et al., 2005) and by the presence of relevant anxiety symptoms in 30% of residence physicians (Buddeberg-Fischer et al., 2009).'

Methods:

- Preprocessing and nuisance correction for the neuroimaging data could be more thorough (e.g. physiological noise correction, motion correction).

We address these points with additional analyses and additional figures and tables. Please see the responses to the point above and the detailed description of additional analyses steps in the supplemental material on p. 9-11:

‘fMRI control-analyses.

We controlled for physiological noise with nuisance regressors that reflected the time-course within the cerebrospinal fluid (CSF) (Bazin et al., 2019). The CSF mask was generated for each individual by the non-linear unified segment procedure in SPM12 (see above). Time series were extracted for all voxels included in this mask and were submitted to principle component analysis using the matlab function `pca.m` included in the statistics toolbox (MATLAB, The MathWorks, Inc., Natick, Massachusetts, U.S.). The first five principle components for each participant were used as nuisance regressors in the GLM analysis, alongside 6 motion regressors. Supplemental Figure S7 provides a visualization for the first 2 participants and supplemental Table S5 reports the improved statistical results for predicting symptom severity changes following application of this technique.

We ensured predictive relevance and local specificity for the locus coeruleus by comparing anxiety and depression symptom change predictions based on both LC-masks (1SD and 2SD) and several other brainstem nuclei. In addition, we employed a weighted-average data extraction that weighed every voxel’s activity with the probability of membership in the ROI assigned to each voxel. These probabilistic maps included the main brainstem nuclei in the vicinity of the LC, i.e.: medial raphe nucleus (MR), dorsal raphe nucleus (DR), and ventral tegmental area (VTA) provided by the Harvard Ascending Arousal atlas available at <https://www.martinos.org/resources/aan-atlas>. We also compared LC prediction power with the substantia nigra (SN), available at <https://www.nitrc.org/projects/atag/> and the amygdala (<https://fsl.fmrib.ox.ac.uk/fsl/fslwiki/Atlases>). In addition, we repeated all GLMs also for unsmoothed data, since the originally applied 6mm smoothing kernel may have smeared the activity between brain-stem nuclei as well as with adjacent CSF. We summarize the results of all these analyses in the supplemental table S5 (also see supplemental figure S7-S13 illustrating our control analyses).

The new results reveal that stress-related anxiety and depression symptom changes were best predicted by the locus coeruleus, compared to all other brainstem nuclei. They also underline the robustness of our results in several ways. For example, the choice of LC mask did not bias the results: The LC was the only structure to predict symptom changes, for both available types of standardized LC masks (1SD & 2SD; the smaller and more robust map (1SD) yields the stronger correlations). Physio-correction generally improved the statistics in nuclei closest to the 4th ventricle, such as the LC, medial raphe and dorsal raphe (in fact, in the physio-corrected smoothed data, DR and MR now correlate as well with symptom changes). However, the physio-corrected un-smoothed data showed once more that the LC was the only region predicting both anxiety and depression changes, irrespective of LC-mask choice, suggesting that DR and MR correlations in the smoothed data may stem from a smearing of LC activity into these neighboring regions. These additional analyses further strengthen our main conclusion that stress resilience is predicted by responsivity of the LC.

Finally, we conducted a formal analysis of the temporal signal-to-noise ratio (tSNR) across the whole brain, and in particular in the brainstem, making it easier to assess the signal quality of the extracted LC signals in comparison with other brainstem structures. The tSNR was computed by dividing the mean of each time series by its standard deviation for each voxel in the brain. The results confirmed that both the average and subject-specific tSNR in the LC was well above standard cut-offs (>30). We also found that the signal in the LC was in fact strongest amongst all brainstem nuclei, for both standard LC masks (1SD & 2SD, see supplemental figure S8-S11).’

Reviewer #2 (Remarks to the Author):

Recommendation: Accept with major revisions

Grueschow et al performed a timely task-based fMRI study of locus coeruleus in the prediction of depression outcomes in medical students starting internship. The strengths of the study are the use of a rigorous task to elicit surprise-related neural activity in the locus coeruleus and pupillary dilations, and the convincing correlations they observe between neural activity in LC (or LC-amygdala coupling) and subsequent depressive symptoms. These are important findings, building off what is known of LC-amygdala function in anxiety and depression from animal models and previous fMRI studies. The use of a prospective study design is another key addition to the literature on this subject and I believe this study will be well-cited and important.

However, the central framing of the study has a significant flaw which should be dealt with prior to acceptance. The paper claims to predict resilience to stress from LC fMRI, however it is not at all clear how much stress the participants have undergone (see discussion below and Figure 2). One can see from Figure 2 and from the subsequent correlations of change in PHQ-9 depression scores, that individuals are essentially equally likely to get more and less depressed. This disagrees with previous literature cited on the impact of medical internship on depression, and also calls into question this central claim of the paper. If people are equally likely to experience increases or decreases in depression, it may be more accurate to describe this as natural variability in depression rather than resilience to stress.

We thank the reviewer for judging our work timely, rigorous, and convincing, as well as predicting that this work will be well-cited and important. The reviewer did have some concerns which we address in a point-by-point fashion below. In summary, we followed the reviewer's suggestions to (1) discuss possible intercultural differences between medical systems in different countries and the resulting divergence between the present sample and that of other studies, (2) control for actual events experienced during the internship and baseline level of psychopathology, (3) clarify the rationale of the present study and differentiate it from studies focusing on groups with clinically-relevant symptoms, (4) confirm the specificity of our findings to the LC by adding many control analyses and preprocessing steps of the imaging data, (5) include a comprehensive analysis of the behavioral data that controls for multiple comparisons, and (6) discuss limitations of the present study. All of these changes substantiate our initial findings and therefore strengthen our paper substantially. Thanks again to the reviewer for pointing us to these issues and for suggesting how to address them.

Major issues:

1. Is the study truly measuring resilience to stress, or just natural variability in the development of depressive symptoms? The paper claims to be measuring resilience to a stressor (medical internship), and cites a number of papers (refs 68-71) looking at the first year of postgraduate medical residency (internship) in the development of depressive symptoms. I am concerned about whether this is the correct framing for this study since a) the group increase in depression scores is very questionable (see below) and b) the authors do not adequately justify how medical internship and associated stress may compare between the Swiss and American systems. In reality, it may be more accurate to describe this

We thank the reviewer for asking us to clarify the strength of symptom increases present in our sample, and to compare it with cohorts and studies from the US.

As for symptoms (point a), our analyses show statistically significant increases in both depression and anxiety over the three timepoints (anxiety, $F=4.01$, $p=0.022$ and depression, $F=3.11$, $p=0.049$; repeated-measures ANOVA). Thus, there is no question that across the group, symptoms do in fact increase over time. However, we want to emphasize that such *mean* group slope increases are not the focus of our study. Instead, we investigate *individual* variability in such changes and how this can be predicted from neurophysiological measures. We clarify this important point in the manuscript, for instance on page 7:

‘This repeated clinical assessment protocol allowed us to account for initial individual baseline levels of distress prior to the real-world stressor and to fully capture expected variability in stress resilience amongst participants (Bonanno et al., 2011, Curtis et al., 2012)’

With respect to possible differences in medical internships between the US and Switzerland (point b), we agree with the reviewer that these differences may be responsible for potentially lower levels of stress in our cohort. We therefore now describe this possibility in the results section and discuss some potential differences between European and American Medical internships, to clarify the differences as requested by the reviewer.

The results section now contains this statement:

‘The observed symptom changes are smaller as reported in a previous study involving the American medical system (Sen et al., 2010), indicating intercultural difference and suggesting comparably milder levels of stress in the current Swiss cohort.’

The limitations section in the discussion now contains this statement:

‘Third, we investigated a modestly-sized medical student sample. While this sample is representative for the student population at Swiss medical schools, our results may not easily be generalizable to other populations across the world. For example, depression and anxiety levels were generally low in our sample, lower than in some other studies (Sen et al., 2010), hence indicating potential cross-cultural differences in selection or stress exposure. Working hours, in particular, tend to be lower in the Swiss Medical internship system compared to the US (Temple, 2014), leading to potentially milder levels of stress exposure and symptom levels in the current cohort compared to medical residents in the US (see supplemental table S7-S8). In spite of such differences, there is clear evidence that Swiss medical internships are associated with occupational stress and vulnerability to psychopathology, as evidenced by a worsening of physical and psychological well-being and life satisfaction after the first year of internship/residency compared to before (Buddeberg-Fischer et al., 2005) and by the presence of relevant anxiety symptoms in 30% of residence physicians (Buddeberg-Fischer et al., 2009).’

a. In the referenced American medical internship studies, PHQ-9 scores typically increase about 4 points (Sen et al 2010) during internship. However, in the present study (Figure 2C-D), median PHQ-9 scores do not increase at all during internship (Figure 2C), and mean PHQ-9 scores increase very slightly (Figure 2D) driven by a few subjects.

We agree with the reviewer that the group mean symptom changes are smaller than in the Sen et al. 2010 study. This may be due to inherent differences between Swiss and US medical internships, as the reviewer rightfully suggested. We therefore now discuss this point in the limitation section of the main text (p. 31); moreover, we have added two supplementary tables with descriptive statistics about the absolute level of symptom change (Table S7) and the number/percentages of participants reaching symptom severity levels above clinically relevant thresholds (Table S8).

However, we want to emphasize that for our prediction, it is essential that individuals vary in their response to stress. It is much less important for this aim whether or not a majority of individuals displays symptoms of clinical relevance (please see the additional section below discussing clinical relevance). We make sure to clarify this important point in our rationale starting on page 10.

As well, there are methodological issues with the statistics in Figure 2D: they appear to be doing T tests uncorrected for multiple comparisons with a p value close to 0.05. They show in Figure 2D but do not report the effect sizes for these changes (e.g., absolute change in PHQ-9).

We now report the respective effect sizes in a new supplemental table (S7). In addition, the presented t-statistics are post-hoc tests following repeated measures ANOVAs (anxiety; $F=4.01$, $p=0.022$, depression; $F=3.11$, $p=0.049$), so our omnibus statistical analysis does correct for multiple comparisons before the subsequent post-hoc tests. These tests are necessary since reporting symptoms changes at 2 time-points helps to properly quantify individual variability and aids the reader to better judge the dependent variables.

They do not report the change in the percentage of participants developing depression (this goes from negligible to 20-30% in Sen et al 2010). One can also see in the correlation to fMRI measures figures (i.e., Figure 5E-F) that individuals are roughly equally like to be depressed pre- and post-exposure to internship. If this is true, the claims and text should match that.

We now provide an additional supplemental table (S8) that reports the number and percentage of participants developing clinically relevant symptom levels for all three time points. Out of $N=48$, 9, 14 and 16 participants showed above cut-off anxiety levels prior to, 3 months, or 6 months into the internship, respectively. For depression, 2, 3 and 5 subjects showed above cut-off symptom levels at these time-points (see table S8). Resilient responding (a decrease in symptoms across time) was evident in 6 out of 9 subjects that had above clinically relevant cut-off anxiety symptoms before the internship and in 4 out of 9 participants after 6 months. The opposite pattern of stress-related symptom increases from below-clinically relevant anxiety symptom levels was observed in 25 out of 38 participants after 3 months and 16 out of 38 participants after 6 months. A similar picture was evident for depression, where 25 out of 45 participants with below cut-off symptom level prior to the internship exhibited increased symptoms after 3- and 6-months internship stress. All these data further emphasize the between-participant variability necessary for our prospective prediction approach.

Regarding Figure 5 and all regression panels throughout the manuscript: The y-axis plots symptom changes with respect to the symptom level prior to the medical internship. Since this initial symptom level can be variable across participants (see Fig 2), it cannot be concluded from the figure that individuals are roughly equally likely to be depressed pre- and post-exposure to internship. We apologize if this has not been clear from the figure legend, which we have now updated to prevent such misunderstandings in the future. Please note also that in all our prediction analyses, we do account for the symptom baseline level as well as for the previous trauma scores. In fact, those two regressors constitute the base model against which we compare all prediction models (see previous tables on prediction stats and also the new ROC-AUC plots below). Thus, our prediction analyses are not confounded by prior depression or anxiety symptoms. Again, we have made this point very explicit in the manuscript (p. 26) to prevent any misunderstandings in the future.

There is a broader implication of this observation for this paper. It appears that people with high levels of anxiety and depression prior to internship are likely to have higher levels at each timepoint. That being the case, the relationships presented in figure 3 probably also pertain to the baseline timepoint. If this were to be true then the assumption that the relationship holds as a consequence of internship-related stress is undermined.

All our statistical analyses control for t0-scores (baseline) as well as for adverse events experienced prior to the internship (preTrauma in statistical tables). The t0 score does not significantly predict anxiety symptom changes, and it predicts depression symptom change with a *negative* direction effect (meaning that people with elevated depression levels significantly decrease their symptom over the time of stress). Even though we explicitly control for these effects, the LC-responsivity measure predicts symptom changes over and above these baseline measures with a positive effect, meaning that high LC-responsivity predicts increases in symptom severity characteristic of less efficient resilient responding. We now make it very explicit in the manuscript that these new results cannot reflect baseline symptoms and previous traumatic experiences (p. 26 and following) so that readers do not miss this crucial point.

b. The specific level of stress associated with medical internship may be country-specific. The authors cite a number of studies of American medical students and interns. Have these studies (e.g., Sen, S. et al 2010) been validated for the training environment in Switzerland? There are multiple issues that may underlie national differences in the degree of stress during medical internship: typical work hours, social environment, medical system differences. Another is that the period described here as "medical internship" may or may not exactly match up with the period in the Sen study. This is a limitation which can be addressed with more information on the medical internship system in Switzerland in Methods and in discussion on why greater increases in depression were not seen in the study.

We concur with the reviewer that these differences between medical training in the two countries may be responsible for potentially milder levels of stress in our cohort. We therefore now discuss some potential differences between European and American Medical internships in the limitation section in the main part of the manuscript on p. 32:

‘Third, we investigated a modestly-sized medical student sample. While this sample is representative for the student population at Swiss medical schools, our results may not easily be generalizable to other populations across the world. For example, depression and anxiety

levels were generally low in our sample, lower than in some other studies (Sen et al., 2010), hence indicating potential cross-cultural differences in selection or stress exposure. Working hours, in particular, tend to be lower in the Swiss Medical internship system compared to the US (Temple, 2014), leading to potentially milder levels of stress exposure and symptom levels in the current cohort compared to medical residents in the US (see supplemental table S7-S8). In spite of such differences, there is clear evidence that Swiss medical internships are associated with occupational stress and vulnerability to psychopathology, as evidenced by a worsening of physical and psychological well-being and life satisfaction after the first year of internship/residency compared to before (Buddeberg-Fischer et al., 2005) and by the presence of relevant anxiety symptoms in 30% of residence physicians (Buddeberg-Fischer et al., 2009).’

c. If it is not possible to adequately justify the "stress resilience" narrative, the authors may want to remove references to stress resilience which is questionable, and write the paper as "prediction of depressive and anxiety symptoms from LC/pupil biomarkers." It would be reasonable to describe the medical internship period as a period in which stress is expected but in which the degree of stress experience here is mild. So those few excess individuals who develop depression here may be "stress susceptible."

Given the unambiguous data on the experience of stress during Swiss medical internships and the emergence of related psychopathology (see point before and Buddeberg-Fischer et al., 2005, (Buddeberg-Fischer et al., 2009), we maintain that our study provides valid data on the predictive validity of LC function for resilience to real-life stress. However, we make it more explicit (see p. 10) that our work does not aim at predicting anxiety or depression symptoms beyond a clinical cut-off. The additional text reads:

‘Please note that we did not aim to predict whether or not a participant develops stress-related symptoms above a clinically relevant cut-off.’

Rather, we relate the variability in neurophysiological measures to the variability in individual symptom changes across the entire range possible in our cohort. In a larger sample, we may have used latent growth mixture modeling to identify the subgroup of individuals that is resilient, i.e. shows no increase in anxiety or depression levels over time. However, our sample size is too small for this analysis, so we now mention this possibility in the discussion section on p. 35:

‘Future studies may benefit from including further components of this concept, for example variability over time and individual coping strategies, as well as investigating trajectories of change using latent growth mixture models(Swartz et al., 2015).’

2. Discussion of the relationship between pupil and norepinephrine. The manuscript repeatedly describes the pupil as a simple readout of noradrenergic state (e.g., "a well-established external marker of noradrenergic LC-NE firing"), and cites Joshi et al 2016 as the primary source for this. However, the Joshi et al paper actually finds distributed correlations between pupil and a variety of brain areas, and other regions precede pupil-neuron coupling in LC (i.e., anterior cingulate). Furthermore, other studies find substantial (possibly greater) correlation between acetylcholine derived from the basal forebrain and the pupil than norepinephrine (Reimer et al 2016; Nelson and Mooney 2016). It is important that the paper describe the uncertainty in the literature as to the true neuromodulatory correlates of the pupil biomarker, and not attribute it simply to norepinephrine.

We thank the reviewer for this point and now emphasize the ambiguity in the literature about the neuro-modulatory sources underlying pupil dilation throughout the manuscript. We have removed and/or replaced every ‘well-established marker’ in the entire manuscript. For instance, in the abstract we have replaced ‘well-established marker’ with ‘potential marker’ of LC-NE.

In the introduction, p.5, we now write:

‘an external marker associated with noradrenergic LC-NE firing (Joshi et al., 2016) and cholinergic activity(Reimer et al., 2016)’.

Moreover, we now added the following statement to the discussion section (p. 32):

‘Second, even though pupil dilation has primarily been associated with LC-firing and the noradrenergic system, recent evidence has identified also a cholinergic component (Nelson and Mooney, 2016, Reimer et al., 2016). Pupil dilation is thus hardly a specific noradrenergic marker but may reflect activity in multiple neuro-modulatory systems simultaneously. Dissociating the contributions of various neuro-modulatory systems to pupil dilation should be the focus of future human imaging studies. Nevertheless, because current eye-tracker systems can be portable, easy-to-use and inexpensive, our data suggest that pupil dilatation can be very useful as a diagnostic-, monitoring- and treatment-tool not only for stress-related psychopathology (Reimer et al., 2014, Vinck et al., 2015, Warren et al., 2016, van der Wel and van Steenbergen, 2018) but also for studying the aging population and children.’

3. Is the study studying an appropriate sample to have clinical relevance? In order to have clinical relevance, there have to be sufficient numbers of individuals who reach clinically meaningful symptom severity. On the PHQ-9, the cutoff for moderate depression is 10. However, in figure 2C, there are only 3-4 individuals with this level of depression at each timepoint and, as noted earlier, the number does not change with internship. This means that the investigators are not studying an appropriate sample to draw clinical inferences about depression at any timepoint. With respect to anxiety, moderate-severe anxiety would be any score above 37 (38-80). In figure 2A, it is evident that there about 10 subjects with moderate-severe anxiety at each timepoint. Thus, the study does not have an adequate sample to address clinically meaningful questions related to anxiety. Thus, this is a study about “resilience” in relation only to mild levels of anxiety or depression and it is not relevant to clinical populations. Thus, the thrust of the discussion is not in keeping with the actual nature of the data/results.

We agree that our study is less related to clinical samples and is focused more on prediction of individual responses to real-life stress. To make this more explicit, we have added a limitations section to the discussion that critically evaluates the clinical relevance of our study (p. 32). It now reads:

‘Third, we investigated a modestly-sized medical student sample. While this sample is representative for the student population at Swiss medical schools, our results may not easily be generalizable to other populations across the world. For example, depression and anxiety levels were generally low in our sample, lower than in some other studies (Sen et al., 2010), hence indicating potential cross-cultural differences in selection or stress exposure. Working hours, in particular, tend to be lower in the Swiss Medical internship system compared to the

US (Temple, 2014), leading to potentially milder levels of stress exposure and symptom levels in the current cohort compared to medical residents in the US (see supplemental table S7-S8). In spite of such differences, there is clear evidence that Swiss medical internships are associated with occupational stress and vulnerability to psychopathology, as evidenced by a worsening of physical and psychological well-being and life satisfaction after the first year of internship/residency compared to before (Buddeberg-Fischer et al., 2005) and by the presence of relevant anxiety symptoms in 30% of residence physicians (Buddeberg-Fischer et al., 2009).’

Minor issues:

a. Paper moves back and forth between the terms norepinephrine and noradrenalin. Would recommend consistency on this.

We could not find a single use of norepinephrine in the current manuscript. If the reviewer refers to use of ‘LC-NE’ vs noradrenaline we opted for using LC-NE because it is the most-used abbreviation in the literature for the noradrenergic system.

b. I would like to see ROC curves in addition to the leave two-out prediction accuracy metrics used in Figure 4,5,7 with reporting of Area-under-curve as an additional metric for prediction accuracy for the models reported.

We have now added 3 additional figures (S11-S13) with ROC-AUC plots and scores for each of the three predictors (LC-responsivity, LC-Amygdala connectivity, and pupil dilation; S11) and for the combined prediction of the previously-reported models concerning anxiety (S12) and depression (S13).

c. The term trauma is used multiple times in the manuscript to refer to the period of medical internship. This goes far beyond the limitations of the stress of medical internship concern I discuss above. Trauma is usually a life-threatening event, which this is not. Would remove this term from the manuscript.

We agree with the reviewer and have now removed the term ‘trauma’ from the manuscript and supplemental material. We now refer to the internship as either ‘potentially traumatic events’, ‘adversity’, or ‘stress exposure’.

d. Can the pupil be a biomarker alone (or in concert with epidemiological factors) in prediction subsequent depression and anxiety? Given that it is much more easily collected than the LC fMRI biomarker, I would like to see model performance for the pupil biomarker alone at predicting subsequent symptoms (AUC, variance explained) in the main text.

This would also be a good thing to highlight in the discussion, as this may have a broader practical application in clinical psychiatry but is somewhat under-emphasized in the paper.

We agree with the reviewer that the pupil dilation may hold great potential as future biomarker in stress-related psychopathology. We have now added additional figures (S11-S13) reporting the ROC-AUC plots and score for the pupil alone (S11) and in concert with other factors (S12-S13). These figures show that pupil alone has some predictive validity, but also that the neural data add substantial explanatory power to the model. Based on these results, we wish to not over-interpret our pupil results. Nevertheless, ongoing studies in our and other laboratories are currently validating these

measures. To alert the reader to the potential of these measures, we did add a statement (p. 32) emphasizing the future relevance of pupil dilation as diagnostic-, monitoring- and treatment-tool in stress related psychopathology:

‘Nevertheless, because current eye-tracker systems can be portable, easy-to-use and inexpensive, our data suggest that pupil dilatation can be very useful as a diagnostic-, monitoring- and treatment-tool not only for stress-related psychopathology (Reimer et al., 2014, Vinck et al., 2015, Warren et al., 2016, van der Wel and van Steenbergen, 2018) but also for studying the aging population and children.’

Reviewer #3 (Remarks to the Author):

Grueschow and colleagues examined if resilience to stressful events can be predicted from fMRI responses in the brainstem noradrenergic nucleus locus coeruleus (LC), and from pupil diameter, in healthy medical students performing an emotional stroop task. The manuscript is well written, and contains many interesting results. I also see the relevance and novelty of prospective studies such as this one, which aim to examine the factors that make a person susceptible to the effects of stress, instead of examining the effects of stress after the fact.

We thank the reviewer for this positive assessment of our study, its novelty, and its relevance.

However, I have a number of major concerns (both conceptual and methodological) that leave me unconvinced that the conclusions in the manuscript are justified. My primary concerns are that the task is not well grounded in electrophysiological studies that have charted the responsivity of LC neurons, and that the imaging protocol is unsuitable for functional imaging of the brainstem (specifics below). Unfortunately, these concerns preclude me from recommending that this manuscript is suitable for publication in this journal.

We thank the reviewer for suggesting we give more theoretical background concerning the selection of our experimental task and for suggesting that we ascertain the specificity of our findings given the limitations of our brainstem imaging protocol. We have addressed all possible points raised by adding additional text and extensive control analyses that strengthen our conclusions substantially. We would like to thank the reviewer for the cogent suggestions that have allowed us to improve our paper. Please find our point-by-point response below.

Major:

1) Task: There is little validation from basic animal research as to what the spiking properties of the LC are during this particular task, and if the LC should be responsible for trial sequence-related behavioral effects during this task in the first place. This means that validation analyses, where a stimulus that is known to elicit a strong spiking response is used to validate the BOLD signal response in the LC, are not possible.

The literature that is cited to corroborate the involvement of the LC-NE system in this task is all pupil-related. The pupil is indeed a proxy of activity of the LC-NE system, but hardly a specific one, as the pupil is also correlated with activity in other neuromodulatory nuclei (see for example Reimer et al., 2016, and de Gee et al., 2017) and sensitive to abstract concepts such as cognitive effort that do not solely reflect activity in the LC and likely differ between task conditions.

While there is indeed a long-standing literature that implicates phasic noradrenergic hyperreactivity in the development of stress-related disorders, the part of this literature that involves direct LC recordings mainly centers on altered LC discharge properties in response to stressful / noxious stimuli, which are known to elicit a strong LC response. The predictions and interpretation of the results in the current experiment would have been more straightforward if stimuli of this kind were used (or another task that reliably engages the LC such as an oddball

task), compared to complex trial-sequence effects of which the neurophysiological underpinnings are less established.

We agree completely with the reviewer that pupil dilation is only a proxy of LC-NE system activity, and that it is hardly specific. We thank the reviewer for pointing this out and now emphasize throughout the manuscript (e.g. introduction, pp. 5) the ambiguity in the literature about the neuro-modulatory sources underlying pupil dilation. For instance, we have removed and/or replaced every reference to a ‘well-established marker’ in the entire manuscript. Most importantly, we have now added the statement given below to the discussion section (p. 32). All these changes make sure we do not overinterpret the selectivity of the link between pupil dilation and the LC-NE system.

‘Second, even though pupil dilation has primarily been associated with LC-firing and the noradrenergic system, recent evidence has identified also a cholinergic component (Nelson and Mooney, 2016, Reimer et al., 2016). Pupil dilation is thus hardly a specific noradrenergic marker but may reflect activity in multiple neuro-modulatory systems simultaneously. Dissociating the contributions of various neuro-modulatory systems to pupil dilation should be the focus of future human imaging studies. Nevertheless, because current eye-tracker systems can be portable, easy-to-use and inexpensive, our data suggest that pupil dilatation can be very useful as a diagnostic-, monitoring- and treatment-tool not only for stress-related psychopathology (Reimer et al., 2014, Vinck et al., 2015, Warren et al., 2016, van der Wel and van Steenbergen, 2018) but also for studying the aging population and children.’

We also agree with the reviewer that animal neurophysiology studies of the LC have primarily employed paradigms with very salient stimuli, such as odd-ball experiments. However, the previous literature in humans and characteristics of this task precluded us from employing such a paradigm for our purpose of stress-resilience prediction. That is, a recent human fMRI study reporting enhanced LC responses during auditory-odd-ball stimuli in PTSD patients (Naegeli, et al. 2017) found that the individual LC-response did not correlate with any additional physiological measure such as pupil-dilation, skin-conductance or heart-rate. Critically, the study also did not find any association between the odd-ball induced LC-response and the severity of PTSD symptoms (measured with an instrument that includes anxiety or depression sub-scales). Moreover, the LC-response induced via auditory odd-ball stimuli in the healthy controls in this study did not differ significantly from zero, suggesting that odd-ball stimuli do not generally drive LC responding in humans. Finally, since only about 20% of stimuli serve as odd-ball trials in such paradigms, the experimental time required to obtain sufficiently strong signals particularly in the LC is rather long and usually exceeds 20min (see below for technical difficulties associated with LC imaging). All these arguments prevented us from opting for an oddball-task for our specific purpose. To make this point explicit for future readers, we have now added the following point to the introduction (p. 6):

‘We chose the conflict task over another standard task reported to activate the arousal system, the odd-ball task (Murphy et al. 2014), because a previous report indicated no correlation between human LC-odd-ball-responses and any additional physiological measure such as pupil-dilation, skin-conductance or heart-rate (Naegeli, et al. 2017). Critically, this study also did not find any association between the odd-ball induced LC-response and the severity of anxiety or depression symptoms, which are the focus of the present work.’

Conversely, considerable evidence suggests that conflict-related signals are particularly capable in driving the arousal system in humans. First, several prior human functional imaging reports show reliable involvement of LC-NE system during conflict resolution involving stroop-tasks (Krebs et al., 2013, Kohler et al., 2016, Kohler et al., 2019) as well as during tasks requiring the resolution of unexpected uncertainty (Payzan-LeNestour et al., 2013). Second, a dedicated theoretical account suggests LC involvement in conflict resolution (Verguts and Notebaert, 2009, Laeng et al., 2011, van Steenbergen and Band, 2013, Fischer et al., 2018, van der Wel and van Steenbergen, 2018). Specifically, this account proposes that conflict is detected by the ACC, which triggers a phasic response of the LC and global release of NE, with the effect of modulating trial sequence effects and learning (reviewed in Berridge and Waterhouse, 2003; Bouret and Sara, 2005; Nieuwenhuis, 2011). Finally, there is considerable evidence for pupil signal involvement in our specific task (Laeng et al., 2011, van Steenbergen and Band, 2013, Fischer et al., 2018, van der Wel and van Steenbergen, 2018), and considerable evidence supports the notion that pupil dilation is indeed related to LC-firing (Joshi et al. Neuron 2016; Zerbi et al. Neuron 2019). We now describe this evidence for LC involvement in the task we selected in more detail in the manuscript (on p. 5-7) to make our reasoning explicit for future readers.

‘Considerable evidence suggests that such conflict-related signals engage the arousal system in humans. For instance, several prior human functional imaging reports show reliable involvement of LC-NE system during conflict resolution involving stroop-tasks (Krebs et al., 2013, Kohler et al., 2016, Kohler et al., 2019) as well as during tasks requiring the resolution of unexpected uncertainty (Payzan-LeNestour et al., 2013).’

...

‘The resolution of conflict incurs processing costs, including an upregulation of task-relevant information (Egner and Hirsch, 2005), which have been associated with increased arousal and noradrenalin release thought to involve the LC-NE (Verguts and Notebaert, 2009, Laeng et al., 2011, van Steenbergen and Band, 2013, Fischer et al., 2018, van der Wel and van Steenbergen, 2018). Behaviorally, the conflict is typically observed as higher reaction times (RT) for incongruent than congruent trials (Etkin et al., 2006, Egner, 2007, Egner et al., 2008) and as congruency-sequence effects (Egner, 2007, Mansouri et al., 2009) (Figure 1B): Responses in conflict-inducing incongruent trials are faster when the previous trial was also incongruent (II), compared to when the previous trial was congruent (CI), reflecting time-consuming noradrenergic upregulation processes necessary when conflict is encountered after no-conflict trials (Verguts and Notebaert, 2009, Fischer et al., 2018). These upregulation processes have lasting effects and therefore carry over to the subsequent incongruent stimulus on II trials (Botvinick et al., 2001, Etkin et al., 2006, Egner, 2007, Egner et al., 2008). We thus contrasted CI>II trials (which are identical in terms of presented stimuli and response requirements) to isolate neural processes involved in potentially noradrenergic (Laeng et al., 2011, van Steenbergen and Band, 2013, Fischer et al., 2018, van der Wel and van Steenbergen, 2018) upregulation of cognitive control. This contrast essentially provides us with a measure of how much an individual brain is taxed by upregulation to resolve emotional conflict. We indexed the effects of this contrast on basic LC-NE activation, the downstream consequence of functional coupling between LC-NE and amygdala, and the peripheral LC-NE-related pupil dilation (Laeng et al., 2011, van Steenbergen and Band, 2013, van der Wel and van Steenbergen, 2018). Using these measures, we could thus test whether higher responsivity of human LC-NE before the onset of a real-world stressor may predict the degree to which an individual will be affected by this stressor.’

Finally, several additional fMRI studies show that the LC-BOLD signal correlates with the phasic pupil response. Such activation has also been observed for several other brainstem regions, as the reviewer rightfully pointed out (Reimer et al., 2016, and de Gee et al., 2017). To corroborate the specificity of our findings, we now report multiple control analyses and additional figures (S7-S13) and tables (S5 & S6) that strongly emphasize the local specificity of our effects to the LC (see the separate point below for details).

In sum, the revised manuscript now states very clearly why oddball tasks are not suitable for our purpose and why the task we selected is indeed well suited, as also evidenced by our findings. We hope the reviewer agrees with us that these changes resolve any possible ambiguity on these points.

2) Data acquisition and analysis: Proper functional imaging of the brainstem, and in particular the LC, is notoriously difficult due to the small size of the nuclei involved, their proximity to the ventricles, and inherently low signal-to-noise ratio in the brainstem. As such, it requires non-standard techniques for both data acquisition and analysis (see Eckert et al., 2010 for a list of recommendations, also see Brooks et al., 2013, Forstmann et al., 2017, and Turker et al., 2019 for further considerations). Unfortunately, none of these techniques have been applied here.

We concur fully with the reviewer and now discuss these limitations and the reviewer’s suggestions (p. 31). In addition, we are grateful to the reviewer for providing us with crucial methodological advice on how to improve our analysis pipeline. We have followed this advice and have conducted substantial control analyses that have improved the quality and selectivity of our results. In the following, we summarize these analyses and all changes to the manuscript in a point-by-point response to every issue the reviewer raised. To sensitize the reader to all these issues, we have added a cautionary note to the results section (p. 20), which reads:

‘fMRI Control Analyses

Optimal functional imaging of the brainstem, and in particular the LC, is notoriously difficult due to the small size of the nuclei involved, their proximity to the ventricles, and inherently low signal-to-noise ratio in the brainstem. In order to unequivocally identify LC-NE activity, non-standard techniques would be ideal for both data acquisition and analysis (Brooks et al., 2013, Forstmann et al., 2017, Turker et al., 2019) On the acquisition side this would, for instance, entail high-field imaging and partial-brain coverage, which allows particularly small (submillimeter) voxel resolution to avoid partial voluming and reduce pulsating artefacts from the adjacent 4th ventricle (please see the limitations section for a discussion of fundamental methodological steps to improve brainstem imaging). However, use of such a specialized imaging protocol would preclude whole-brain imaging and therefore inferences about influences of the LC on other brain systems (e.g., the amygdala and neocortical areas involved in conflict processing). Moreover, it would make it difficult for our approach to be replicated and extended in standard fMRI lab settings around the world. Thus, we opted for a standard 3T scanner and a routine fMRI-sequence with relatively low voxel resolution (2.5 mm isotropic) that nevertheless retains good signal-to-noise ratio in the brain stem (supplemental figures S8-S10). Importantly, to ascertain the specificity of our results to the LC, we conducted multiple control analyses. These included weighted averaging for region-of-interest data extraction, control for additional brainstem-nuclei, controlling for physiological nuisance variables based on principle component analysis of individually identified CSF probability tissues classes, as

well as applying these nuisance variables in entirely new regression models to both smoothed and unsmoothed data. Applying these specialized analysis techniques, we found that (1) brainstem data quality was substantially enhanced, (2) the reported results and predictions not only replicate but (3) were statistically improved, (4) LC-specificity was strengthened and (5) the conclusions were substantiated. Please see the supplementary methods and results section for detailed descriptions of these techniques, additional Figures (S7-S10) and statistical results (Table S5 and S6), as well as a formal temporal-signal-to-noise (tSNR) analysis of the whole brain and specifically the brainstem.’

In addition, we have further added an extensive limitations section in the discussion which reads (starting p. 31):

‘Our study is not without limitations. Three major points should be taken into consideration. First, optimal functional imaging of the brainstem, and in particular the LC, is difficult due to the small size of the nuclei, their proximity to the ventricles, and inherently low signal-to-noise ratio in the brainstem. Identification of LC-NE activity may thus benefit from more specialized data acquisition techniques than those applied here (Brooks et al., 2013, de Gee et al., 2017, Forstmann et al., 2017, Schumann et al., 2018, Turker et al., 2019). For instance, on the acquisition side, one may obtain high-field imaging for superior signal-to-noise ratio, high resolution T2-weighted anatomical imaging for accurate brainstem registration to standard space, neuromelanin-sensitive imaging for localization and dissociation of individual participants’ LC from other brainstem nuclei. In addition, partial functional brain coverage could achieve particularly small (submillimeter) voxel resolution with high tSNR to avoid partial voluming effects and to counteract pulsating artefacts in the CSF and adjacent 4th ventricle. Pulsation could be quantified and mitigated with continuously recorded electrocardiogram. While future studies could use such specialized technology to focus on the LC with more certainty, in the present study, we applied more routine MR imaging protocols that can be replicated in numerous research settings worldwide. However, local specificity and physiological noise reduction can also be optimized via specialized analysis techniques, which we did apply here. For instance, we controlled for physiological noise by including principle components of the time-course in the CSF as nuisance regressors in the GLM-analysis (Bazin et al., 2019). Moreover, to enhance regional specificity of our results, we also analyzed activity from LC-adjacent brain-stem nuclei and by repeating our initial analyses for unsmoothed data from a smaller LC-mask (1SD vs. 2SD). Applying all these techniques to our data in fact enhanced the predictive accuracy of LC activity for symptom changes and showed that data from other brainstem nuclei do not allow this prediction. Thus, despite our use of more standard imaging protocols, our data provide evidence that specifically LC activation in response to emotional conflict predicts symptom changes in response to real-life stress (for detailed description of the applied methods and results please see supplemental methods, tables S5-S6 and figures S7-S10).

Second, even though pupil dilation has primarily been associated with LC-firing and the noradrenergic system, recent evidence has identified also a cholinergic component (Nelson and Mooney, 2016, Reimer et al., 2016). Pupil dilation is thus hardly a specific noradrenergic marker but may reflect activity in multiple neuro-modulatory systems simultaneously. Dissociating the contributions of various neuro-modulatory systems to pupil dilation should be the focus of future human imaging studies. Nevertheless, because current eye-tracker systems can be portable, easy-to-use and inexpensive, our data suggest that pupil dilatation can be very useful as a diagnostic-, monitoring- and treatment-tool not only for stress-related psychopathology (Reimer et al., 2014, Vinck et al., 2015, Warren et al., 2016, van der Wel and van Steenbergen, 2018) but also for studying the aging population and children.

Third, we investigated a modestly-sized medical student sample. While this sample is representative for the student population at Swiss medical schools, our results may not easily be generalizable to other populations across the world. For example, depression and anxiety levels were generally low in our sample, lower than in some other studies (Sen et al., 2010), hence indicating potential cross-cultural differences in selection or stress exposure. Working hours, in particular, tend to be lower in the Swiss Medical internship system compared to the US (Temple, 2014), leading to potentially milder levels of stress exposure and symptom levels in the current cohort compared to medical residents in the US (see supplemental table S7-S8). In spite of such differences, there is clear evidence that Swiss medical internships are associated with occupational stress and vulnerability to psychopathology, as evidenced by a worsening of physical and psychological well-being and life satisfaction after the first year of internship/residency compared to before (Buddeberg-Fischer et al., 2005) and by the presence of relevant anxiety symptoms in 30% of residence physicians (Buddeberg-Fischer et al., 2009).’

i) The voxel size of the functional sequence was rather large due to whole-brain field of view, whereas sequences that are optimized for brainstem imaging typically reduce the voxel size (and / or TR) at the expense of the field of view.

The 2.5mm isotropic voxel size we used was the absolute minimum we could achieve using our 3T-Phillips scanner while simultaneously covering the entire brain. We opted for a whole-brain imaging sequence for several reasons. It was for instance essential to cover the amygdala to assess symptom changes related to the functional coupling between LC and Amygdala, a hypothesis derived from animal neurophysiology as stated in the manuscript on p. 4. Moreover, we aimed to link the current work to previous human imaging results regarding conflict-control and emotional Stroop-tasks, which required imaging of the ACC and DLPFC. In addition, since we also aim for clinical application, usability and practicality, we specifically aimed to show that it is possible to acquire data using standardized sequences rather than very advanced techniques that are potentially unavailable to many laboratories or clinics. We stress this reasoning on p. 20 to make it transparent to the reader.

ii) No high resolution T2-weighted anatomical image for accurate registration to standard space of the brainstem was acquired.

We very much agree with this comment but unfortunately did not acquire such an image, precluding us from specific brainstem registration. Reviewing the suggested literature made it clear that this procedure would further enhance the reliability of the signal and we now emphasize this point in the new text (see above and p. 31).

iii) No neuromelanin sensitive sequence was used for localization of individual participants’ LC.

Unfortunately, we did not acquire a neuromelanin sensitive sequence, which we now also list as a limitation in the discussion section for LC signal reliability (see above and p. 31).

iv) The “new segment” tool that was used for registration to standard space performs a rigid body transform, not non-linear transformation or warping, which is essential for an accurate registration of the brainstem. Ideally, one would implement a separate registration procedure for the brainstem alone.

We apologize for an inaccuracy in reporting the employed normalization procedure. In the Methods section we wrongfully reported the ‘new segment’ procedure while in fact we used and cited the ‘unified segment’ framework as implemented in SPM8 (Ashburner and Friston, 2005). We have now added an additional section to the supplemental methods (p. 6) to make this point clearer:

‘The procedure incorporates spatial normalization and tissue class segmentation within the same model so that an optimal solution is found for both within the same framework. The procedure uses 6 tissue probability classes for MW, GM, CSF, skull, soft tissue and other (i.e.: eyes). These standardized probability maps for different tissue classes were constructed from a large number of brains that are registered into a common space. In the ‘unified segment’ Bayesian framework, these maps represent the prior probability of any voxel belonging to a particular tissue class (priors). The procedure warps the standard tissue probability maps to match the current subjects’ maps by maximizing their mutual information. The inverse transform can then be used to normalize the functional images to standard MNI space. A recent report stated that when taking prior tissue class information into account, the ‘unified segment’ approach as implemented in SPM outperforms several other methods in both precision of registration as well as tissue classification (Valverde et al., 2015).’

v) Spatial smoothing (6 mm FWHM) was applied, smearing activity between the LC, surrounding neuromodulatory nuclei, and the 4th ventricle.

Spatially smoothing is necessary to apply random-field theory and mass-univariate statistical inference across the whole brain. We were not only interested in the LC but also in replicating and connecting with previous work on conflict-adaptation and conflict resolution, which enhances data reliability and addresses a large number of scientific interests outside of the psychopathology community. In addition, slightly spatially smoothing the data is also the basis for statistical inference when using psychophysiological interactions, which is essential to noninvasively test in humans the predictive power of LC-Amygdala connectivity for anxiety vulnerability, a hypothesis we derived from recent rodent literature (McCall et al., 2015, McCall et al., 2017). We therefore maintain that the main analyses reported in the text are valid.

However, in terms of inferences regarding specifically the LC, the reviewer rightfully points out that smoothing may smear activity with surrounding neuro-modulatory nuclei and CSF. We have now addressed this concern with numerous control analyses suggested by the reviewer, which involve weighted averaging for data extraction, control for additional brainstem-nuclei, including nuisance variables based on principle component analysis of CSF probability tissues classes identified for each individual, and repeating our predictions for both smoothed and unsmoothed data. Moreover, as suggested by the reviewer, we have now included a formal tSNR analysis of the whole brain and specifically the brainstem, making it easier for the reader to assess the signal quality of the extracted LC signals and other brainstem nuclei. When possible, we illustrated our control analyses with additional figures (S7-S10) and summarized the statistical results in new tables (S5-S6).

We shortly list and describe the additional control analyses in the Methods section of the main paper (see text cited above and on p. 42 in the manuscript). In the supplemental material, we give a detailed explanation of the additional analysis steps, we list and

describe the results, and we have added new tables and figures. The supplemental section now reads (see p. 9-11):

‘fMRI control-analyses.

We controlled for physiological noise with nuisance regressors that reflected the time-course within the cerebrospinal fluid (CSF) (Bazin et al., 2019). The CSF mask was generated for each individual by the non-linear unified segment procedure in SPM12 (see above). Time series were extracted for all voxels included in this mask and were submitted to principle component analysis using the matlab function `pca.m` included in the statistics toolbox (MATLAB, The MathWorks, Inc., Natick, Massachusetts, U.S.). The first five principle components for each participant were used as nuisance regressors in the GLM analysis, alongside 6 motion regressors. Supplemental Figure S7 provides a visualization for the first 2 participants and supplemental Table S5 reports the improved statistical results for predicting symptom severity changes following application of this technique.

We ensured predictive relevance and local specificity for the locus coeruleus by comparing anxiety and depression symptom change predictions based on both LC-masks (1SD and 2SD) and several other brainstem nuclei. In addition, we employed a weighted-average data extraction that weighed every voxel’s activity with the probability of membership in the ROI assigned to each voxel. These probabilistic maps included the main brainstem nuclei in the vicinity of the LC, i.e.: medial raphe nucleus (MR), dorsal raphe nucleus (DR), and ventral tegmental area (VTA) provided by the Harvard Ascending Arousal atlas available at <https://www.martinos.org/resources/aan-atlas>. We also compared LC prediction power with the substantia nigra (SN), available at <https://www.nitrc.org/projects/atag/> and the amygdala (<https://fsl.fmrib.ox.ac.uk/fsl/fslwiki/Atlases>). In addition, we repeated all GLMs also for unsmoothed data, since the originally applied 6mm smoothing kernel may have smeared the activity between brain-stem nuclei as well as with adjacent CSF. We summarize the results of all these analyses in the supplemental table S5 (also see supplemental figure S7-S13 illustrating our control analyses).

The new results reveal that stress-related anxiety and depression symptom changes were best predicted by the locus coeruleus, compared to all other brainstem nuclei. They also underline the robustness of our results in several ways. For example, the choice of LC mask did not bias the results: The LC was the only structure to predict symptom changes, for both available types of standardized LC masks (1SD & 2SD; the smaller and more robust map (1SD) yields the stronger correlations). Physio-correction generally improved the statistics in nuclei closest to the 4th ventricle, such as the LC, medial raphe and dorsal raphe (in fact, in the physio-corrected smoothed data, DR and MR now correlate as well with symptom changes). However, the physio-corrected un-smoothed data showed once more that the LC was the only region predicting both anxiety and depression changes, irrespective of LC-mask choice, suggesting that DR and MR correlations in the smoothed data may stem from a smearing of LC activity into these neighboring regions. These additional analyses further strengthen our main conclusion that stress resilience is predicted by responsivity of the LC.

Finally, we conducted a formal analysis of the temporal signal-to-noise ratio (tSNR) across the whole brain, and in particular in the brainstem, making it easier to assess the signal quality of the extracted LC signals in comparison with other brainstem structures. The tSNR was computed by dividing the mean of each time series by its standard deviation for each voxel in the brain. The results confirmed that both the average and subject-specific tSNR in the LC was well above standard cut-offs (>30). We also found that the signal in the LC was in fact strongest amongst all brainstem nuclei, for both standard LC masks (1SD & 2SD, see supplemental figure S8-S11).’

vi) A sphere was placed on the center of mass of the LC template rather than using a weighted mean of the template itself, further blurring the effective functional extent of the LC mask.

We have now included the weighted average extraction (based on region membership probability) for both standard LC-masks and all other brainstem nuclei suggested by the reviewer (Harvard Ascending Arousal atlas available at <https://www.martinos.org/resources/aan-atlas>, and substantia nigra available at <https://www.nitrc.org/projects/atag/>). We additionally also added the amygdala to test for its predictive power as this region is known to be involved in emotion regulation (<https://fsl.fmrib.ox.ac.uk/fsl/fslwiki/Atlases>). We summarize the results for all regions extracted using a weighted average in the new table below for the original unsmoothed data, physio-corrected smoothed data (see methodological specifics below), as well as for physio-corrected un-smoothed data. The new results confirm that stress-related anxiety and depression symptom changes correlated overall best with the LC (significant statistics marked in BOLD). Several additional interesting observations can be made based on physio-correction as well as physio-corrected un-smoothed data.

- 1.) The weighted average extraction from the original smoothed data reveals that only the LC significantly predicts anxiety and depression symptom changes, irrespective of which LC mask is used (1SD or 2SD).
- 2.) As suspected by the reviewer, the physio-correction generally improves the statistics in nuclei closest to the 4th ventricle, such as the LC, medial raphe and dorsal raphe.
- 3.) In the physio-corrected but still smoothed data, DR and MR now correlate as well with symptom changes.
- 4.) However, the physio-corrected un-smoothed data reveal once more that only the LC signal predicts both anxiety and depression changes irrespective of LC-mask choice. This suggests that DR and MR correlations in the smoothed data may stem from a smearing of LC activity into these regions, since these nuclei are located directly adjacent to the LC. The reviewer had rightfully cautioned us to check for this possibility, and our analyses now control for it.

Weighted average of smoothed data							
ROIs	voxels	CI>II		Anxiety		Depression	
		T	p	R	p	R	p
LC_2SD	135	3.075	0.004	0.37	0.01	0.301	0.038
LC_1SD	84	2.948	0.005	0.357	0.013	0.301	0.038
MR	77	4.138	0.001	0.383	0.007	0.275	0.058
DR	200	1.977	0.054	0.182	0.216	0.164	0.266
AMY	2966	1.612	0.114	0.069	0.642	0.123	0.403
SN	1144	1.564	0.124	0.24	0.1	0.1	0.5
VTA	697	0.514	0.61	0.193	0.19	0.138	0.351
Weighted average of physio-corrected smoothed data							
	voxels	CI>II		Anxiety		Depression	

ROIs		T	p	R	p	R	p
LC_2SD	135	2.234	0.03	0.407	0.004	0.36	0.012
LC_1SD	84	2.018	0.049	0.397	0.005	0.364	0.011
MR	77	3.35	0.002	0.35	0.015	0.271	0.062
DR	200	1.503	0.139	0.444	0.002	0.31	0.032
AMY	2966	1.186	0.242	0.246	0.091	0.249	0.088
SN	1144	2.055	0.045	0.288	0.048	0.254	0.082
VTA	697	1.403	0.167	0.175	0.234	0.19	0.195
Weighted average of physio-corrected unsmoothed data							
	voxels	CI>II		Anxiety		Depression	
ROIs		T	p	R	p	R	p
LC_2SD	135	1.574	0.122	0.388	0.006	0.335	0.02
LC_1SD	84	1.089	0.282	0.375	0.009	0.382	0.007
MR	77	2.608	0.012	0.173	0.239	0.077	0.605
DR	200	0.748	0.458	0.373	0.009	0.071	0.63
AMY	2966	0.946	0.349	0.194	0.186	0.211	0.15
SN	1144	1.607	0.115	0.298	0.04	0.199	0.175
VTA	697	1.343	0.186	0.201	0.171	0.209	0.153

vii) No retrospective image correction (regression-based or data driven ICA-based) to suppress physiological noise was applied. This is really essential if one wants to separate the 4th ventricle and LC, because the former pulsates with every heartbeat and is directly adjacent to the LC.

viii) The 4th ventricle signal was not used as a nuisance regressor.

We are grateful to the reviewer for these propositions because applying these suggested physiological control measures has substantially enhanced data quality. Introducing the suggested nuisance variables to control for physiological noise increased the statistical significance of the LC-responsivity predictions, for both anxiety and depression symptom changes, and simultaneously enhanced the local specificity of our conclusion. Again, these improvements were obtained irrespective of the choice of LC-mask and in both smoothed and unsmoothed data.

Physiological noise regressors were derived applying principle component analysis (PCA, Basin et al. 2019) to the time-series in voxels corresponding to each participants' individually segmented CSF probabilistic map as obtained during the 'unified segment' procedure (see above). For each subject's GLM (smoothed and unsmoothed data), the first five principle components were added as nuisance regressors along with the six motion regressors obtained during the realignment procedure. A new supplemental figure illustrates this approach with the participant-specific CSF tissue map (dark blue) overlaid on a standard brain for the first two subjects. In addition, the motion regressors (translation = magenta, rotation = cyan) and the first five components for each of the two participants (dark blue) are plotted. We now include this information and this figure in the supplemental methods section.

ix) No formal tSNR analysis of the brainstem was included, and thus it is difficult for the reader to assess the signal quality of the extracted LC signals.

We thank the reviewer for this suggestion and have now included a formal tSNR analysis of the un-smoothed whole brain data as well as the brainstem data, making it easier for the reader to assess the signal quality of the extracted LC signal. We now include three new supplementary figures illustrating whole brain tSNR as well as focusing on the

brainstem. We also compared LC signal quality with the adjacent brainstem nuclei and find that not only is the signal quality for LC sufficiently high for both LC-masks (tSNR: $1SD = \sim 68$, $2SD = \sim 66$, (Murphy et al., 2007)), but it is also highest amongst all adjacent brainstem nuclei (see three additional supplementary figures below).

Temporal signal to noise ratio (tSNR) averaged across all 48 participants. (A) Sagittal slices were chosen here to emphasize the difference between signal quality between cortex and brainstem. Color-scale ranges from 0-100. The red rectangle indicates the cutout region displayed in B focusing on the brainstem. (B) tSNR values in sagittal slices of the brainstem. Please note that color-scale limits range 0-80 in order to make tSNR contrasts within the brainstem more easily apparent. See next figure for coronal LC-mask comparison and overlay.

Mean temporal signal-to-noise ratio (N= 48) in 3 coronal views slicing through the LC. For easy comparison, each different slice-view (indicated by the Y-coordinate above) displays a zoomed view of the same slice containing the probabilistic LC-2SD map on the left (in shades of green).

Location of brainstem nuclei and their extracted mean and individual tSNR-values. (A) Location of brain stem nuclei in reference to the whole brain. (B) Zoomed-in version of the brainstem nuclei and their color-code used in C. SN = Substantia nigra, DR = Dorsal raphe, VTA = Ventral tegmental area, MR = Medial raphe, LC = Locus coeruleus. (C) Each panel contains bar-plots depicting individual tSNR value for each brainstem nucleus and participant (N=48). Each panels title reports the mean tSNR per region.

For a future submission, I strongly advise the authors to address as many of these points as possible given the data that were acquired, and thoroughly discuss the limitations where addressing them is not possible.

We have followed the methodological propositions suggested by this reviewer, which have enhanced the data quality and substantiated our conclusions. Nevertheless, some suggestions were impossible to conduct because the necessary data were simply not acquired. We value the reviewer's input very much and agree that future studies would benefit from acquiring additional data with advanced sequences, enabling the researchers for a more precise localization of the LC and even stronger statistics following the control measures above. We now explain the missing measures as limitations in a dedicated discussion section and emphasize their importance for future studies. The section in the discussion on p. 31 now reads:

‘Our study is not without limitations. Three major points should be taken into consideration. First, optimal functional imaging of the brainstem, and in particular the LC, is difficult due to the small size of the nuclei, their proximity to the ventricles, and inherently low signal-to-noise ratio in the brainstem. Identification of LC-NE activity may thus benefit from more specialized data acquisition techniques than those applied here (Brooks et al., 2013, de Gee et al., 2017, Forstmann et al., 2017, Schumann et al., 2018, Turker et al., 2019). For instance, on the acquisition side, one may obtain high-field imaging for superior signal-to-noise ratio, high resolution T2-weighted anatomical imaging for accurate brainstem registration to standard space, neuromelanin-sensitive imaging for localization and dissociation of individual participants' LC from other brainstem nuclei. In addition, partial functional brain coverage could achieve particularly small (submillimeter) voxel resolution with high tSNR to avoid partial voluming effects and to counteract pulsating artefacts in the CSF and adjacent 4th ventricle. Pulsation could be quantified and mitigated with continuously recorded electrocardiogram. While future studies could use such specialized technology to focus on the LC with more certainty, in the present study, we applied more routine MR imaging protocols that can be replicated in numerous research settings worldwide. However, local specificity and physiological noise reduction can also be optimized via specialized analysis techniques, which we did apply here. For instance, we controlled for physiological noise by including principle components of the time-course in the CSF as nuisance regressors in the GLM-analysis (Bazin et al., 2019). Moreover, to enhance regional specificity of our results, we also analyzed activity from LC-adjacent brain-stem nuclei and by repeating our initial analyses for unsmoothed data from a smaller LC-mask (1SD vs. 2SD). Applying all these techniques to our data in fact enhanced the predictive accuracy of LC activity for symptom changes and showed that data from other brainstem nuclei do not allow this prediction. Thus, despite our use of more standard imaging protocols, our data provide evidence that specifically LC activation in response to emotional conflict predicts symptom changes in response to real-life stress (for detailed description of the applied methods and results please see supplemental methods, tables S5-S6 and figures S7-S10).

Second, even though pupil dilation has primarily been associated with LC-firing and the noradrenergic system, recent evidence has identified also a cholinergic component (Nelson and Mooney, 2016, Reimer et al., 2016). Pupil dilation is thus hardly a specific noradrenergic marker but may reflect activity in multiple neuro-modulatory systems simultaneously. Dissociating the contributions of various neuro-modulatory systems to pupil dilation should be the focus of future human imaging studies. Nevertheless, because current eye-tracker systems can be portable, easy-to-use and inexpensive, our data suggest that pupil dilatation can be very useful as a diagnostic-, monitoring- and treatment-tool not only for stress-related

psychopathology (Reimer et al., 2014, Vinck et al., 2015, Warren et al., 2016, van der Wel and van Steenbergen, 2018) but also for studying the aging population and children.

Third, we investigated a modestly-sized medical student sample. While this sample is representative for the student population at Swiss medical schools, our results may not easily be generalizable to other populations across the world. For example, depression and anxiety levels were generally low in our sample, lower than in some other studies (Sen et al., 2010), hence indicating potential cross-cultural differences in selection or stress exposure. Working hours, in particular, tend to be lower in the Swiss Medical internship system compared to the US (Temple, 2014), leading to potentially milder levels of stress exposure and symptom levels in the current cohort compared to medical residents in the US (see supplemental table S7-S8). In spite of such differences, there is clear evidence that Swiss medical internships are associated with occupational stress and vulnerability to psychopathology, as evidenced by a worsening of physical and psychological well-being and life satisfaction after the first year of internship/residency compared to before (Buddeberg-Fischer et al., 2005) and by the presence of relevant anxiety symptoms in 30% of residence physicians (Buddeberg-Fischer et al., 2009).'

3) Results and interpretation: The reported peak coordinates for the contrast CI > II are over a centimeter away from the nearest voxel in the Keren LC atlas, which at 2 SD is already quite liberal in spatial extent.

It is indeed the case that other brainstem-nuclei also respond to the CI>II contrast, but to support our hypotheses, the specific CI>II peak activity does not need to be in LC. Important is that anxiety and depression changes can be predicted from the specific signal elicited in LC. We have now validated this with multiple control analyses (see above), including weighted-averaging data extraction, controlling for other surrounding brainstem nuclei, and with physio-corrected and un-smoothed data.

Moreover, the cluster of significant voxels also includes other neuromodulatory nuclei such as the dorsal and median raphe, substantia nigra, and ventral tegmental area. Thus, the conclusions are far too specific in terms of which neuromodulatory nuclei (i.e. LC-NE) drive the correlations with outcome measures.

We have now conducted the relevant control analyses in which we extracted activity using a weighted average from all brainstem control regions suggested by the reviewer (please see figures above). These analyses show that our conclusions about the link between LC responses and stress-induced changes in anxiety and depression symptoms hold and are specific. We are explicit about this in the manuscript on p. 20 and thank the reviewer for these important suggestions that have strengthened our manuscript.

Given that no control analyses using other neuromodulatory nuclei were performed, the authors should substantially tone down claims and rephrase the manuscript in terms that are agnostic about which specific neuromodulator is involved (e.g. “ascending arousal systems” or equivalent non-specific term) instead of making specific claims about LC-NE.

We have conducted multiple control analyses that demonstrated the specificity of our results (see above and p. 20 & 42). However, the reviewer’s point about possible links between arousal, LC function, and other brainstem nuclei is well taken. We now indicate in the introduction and discussion sections (see p. 5 and 32) that arousal may indeed be

linked with activity in LC as well as other neuro-modulatory systems, all of which may differently impact pupil dilation (Reimer et al., 2016).

We now write in the discussion section on p. 32:

‘Second, even though pupil dilation has primarily been associated with LC-firing and the noradrenergic system, recent evidence has identified also a cholinergic component (Nelson and Mooney, 2016, Reimer et al., 2016). Pupil dilation is thus hardly a specific noradrenergic marker but may reflect activity in multiple neuro-modulatory systems simultaneously. Dissociating the contributions of various neuro-modulatory systems to pupil dilation should be the focus of future human imaging studies. Nevertheless, because current eye-tracker systems can be portable, easy-to-use and inexpensive, our data suggest that pupil dilatation can be very useful as a diagnostic-, monitoring- and treatment-tool not only for stress-related psychopathology (Reimer et al., 2014, Vinck et al., 2015, Warren et al., 2016, van der Wel and van Steenbergen, 2018) but also for studying the aging population and children.’

Minor:

How is it possible that CC and IC trials show no difference in pupil diameter around -2, -3 seconds, when the inter trial interval is 3-4 seconds, and C and I trials do show a difference at +1 second?

This can potentially be linked to the different number of trials that went into each condition. As CC and IC curves are only composed of half the number of trials that went into the C and I average, the data may be too noisy and mask the suspected difference. In fact, scrutinizing the average difference curve between -3 and -2 seconds in figure 6F, a small increase can be made out, but this does not reach above the variability inherent in the data.

Typo on page 13, line 258: “develspd”

Thank you, this has been changed.

What was the median and range of trials numbers included for analysis?

‘45 out of 48 Participants viewed 200 trials (2 runs, 100 trials per run). Some subjects missed a few trials and 3 subjects did only 1 run (100 trials). Since we removed trials with an RT larger than 2 standard deviations above the mean, the median trial number was 190 trials (minimum: 94, maximum: 196 trials).’

We have now added this information to the supplemental material on p. 5.

In closing, we would like to thank all reviewers for their careful reading of the manuscript and all suggestions for how we can improve our work. We are confident that incorporating the suggested changes has indeed allowed us to strengthen the conclusiveness of our manuscript and look forward to your reply.

References

- Ashburner J, Friston KJ (2005) Unified segmentation. *Neuroimage* 26:839-851.
- Bazin PL, Alkemade A, van der Zwaag W, Caan M, Mulder M, Forstmann BU (2019) Denoising High-Field Multi-Dimensional MRI With Local Complex PCA. *Front Neurosci-Switz* 13.
- Bonanno GA, Westphal M, Mancini AD (2011) Resilience to Loss and Potential Trauma. *Annu Rev Clin Psycho* 7:511-535.
- Botvinick MM, Braver TS, Barch DM, Carter CS, Cohen JD (2001) Conflict monitoring and cognitive control. *Psychological Review* 108:624-652.
- Brooks JCW, Faull OK, Pattinson KTS, Jenkinson M (2013) Physiological noise in brainstem fMRI. *Frontiers in Human Neuroscience* 7.
- Buddeberg-Fischer B, Klaghofer FC, Buddeberg C (2005) Stress at work and well-being in junior residents. *Z Psychosom Med Psyc* 51:163-178.
- Buddeberg-Fischer B, Stamm M, Buddeberg C, Klaghofer R (2009) Anxiety and depression in residents - Results of a Swiss longitudinal study. *Z Psychosom Med Psyc* 55:37-50.
- Curtis AL, Leiser SC, Snyder K, Valentino RJ (2012) Predator stress engages corticotropin-releasing factor and opioid systems to alter the operating mode of locus coeruleus norepinephrine neurons. *Neuropharmacology* 62:1737-1745.
- de Gee JW, Colizoli O, Kloosterman NA, Knapen T, Nieuwenhuis S, Donner TH (2017) Dynamic modulation of decision biases by brainstem arousal systems. *Elife* 6.
- Egner T (2007) Congruency sequence effects and cognitive control. *Cogn Affect Behav Ne* 7:380-390.
- Egner T, Etkin A, Gale S, Hirsch J (2008) Dissociable neural systems resolve conflict from emotional versus nonemotional distracters. *Cereb Cortex* 18:1475-1484.
- Egner T, Hirsch J (2005) Cognitive control mechanisms resolve conflict through cortical amplification of task-relevant information. *Nature Neuroscience* 8:1784-1790.
- Etkin A, Egner T, Peraza DM, Kandel ER, Hirsch J (2006) Resolving emotional conflict: A role for the rostral anterior cingulate cortex in modulating activity in the amygdala. *Neuron* 51:871-882.
- Fischer R, Ventura-Bort C, Hamm A, Weymar M (2018) Transcutaneous vagus nerve stimulation (tvNS) enhances conflict-triggered adjustment of cognitive control. *Cogn Affect Behav Ne* 18:680-693.
- Forstmann BU, de Hollander G, van Maanen L, Alkemade A, Keuken MC (2017) Towards a mechanistic understanding of the human subcortex. *Nat Rev Neurosci* 18:57-65.
- Joshi S, Li Y, Kalwani RM, Gold JI (2016) Relationships between Pupil Diameter and Neuronal Activity in the Locus Coeruleus, Colliculi, and Cingulate Cortex. *Neuron* 89:221-234.
- Keren NI, Lozar CT, Harris KC, Morgan PS, Eckert MA (2009) In vivo mapping of the human locus coeruleus. *Neuroimage* 47:1261-1267.
- Kohler S, Bar KJ, Wagner G (2016) Differential Involvement of Brainstem Noradrenergic and Midbrain Dopaminergic Nuclei in Cognitive Control. *Human Brain Mapping* 37:2305-2318.
- Kohler S, Wagner G, Bar KJ (2019) Activation of brainstem and midbrain nuclei during cognitive control in medicated patients with schizophrenia. *Hum Brain Mapp* 40:202-213.
- Krebs RM, Fias W, Achten E, Boehler CN (2013) Picture novelty attenuates semantic interference and modulates concomitant neural activity in the anterior cingulate cortex and the locus coeruleus. *Neuroimage* 74:179-187.
- Laeng B, Orbo M, Holmlund T, Miozzo M (2011) Pupillary Stroop effects. *Cognitive Processing* 12:13-21.
- Mansouri FA, Tanaka K, Buckley MJ (2009) Conflict-induced behavioural adjustment: a clue to the executive functions of the prefrontal cortex. *Nat Rev Neurosci* 10:141-152.
- McCall JG, Al-Hasani R, Siuda ER, Hong DY, Norris AJ, Ford CP, Bruchas MR (2015) CRH Engagement of the Locus Coeruleus Noradrenergic System Mediates Stress-Induced Anxiety. *Neuron* 87:605-620.

- McCall JG, Siuda ER, Bhatti DL, Lawson LA, McElligott ZA, Stuber GD, Bruchas MR (2017) Locus coeruleus to basolateral amygdala noradrenergic projections promote anxiety-like behavior. *Elife* 6.
- Murphy K, Bodurka J, Bandettini PA (2007) How long to scan? The relationship between fMRI temporal signal to noise ratio and necessary scan duration. *Neuroimage* 34:565-574.
- Nelson A, Mooney R (2016) The Basal Forebrain and Motor Cortex Provide Convergent yet Distinct Movement-Related Inputs to the Auditory Cortex. *Neuron* 90:635-648.
- Payzan-LeNestour E, Dunne S, Bossaerts P, O'Doherty JP (2013) The neural representation of unexpected uncertainty during value-based decision making. *Neuron* 79:191-201.
- Reimer J, Froudarakis E, Cadwell CR, Yatsenko D, Denfield GH, Tolias AS (2014) Pupil fluctuations track fast switching of cortical states during quiet wakefulness. *Neuron* 84:355-362.
- Reimer J, McGinley MJ, Liu Y, Rodenkirch C, Wang Q, McCormick DA, Tolias AS (2016) Pupil fluctuations track rapid changes in adrenergic and cholinergic activity in cortex. *Nat Commun* 7:13289.
- Schumann A, Kohler S, de la Cruz F, Gullmar D, Reichenbach JR, Wagner G, Bar KJ (2018) The Use of Physiological Signals in Brainstem/Midbrain fMRI. *Front Neurosci-Switz* 12.
- Sen S, Kranzler HR, Krystal JH, Speller H, Chan G, Gelernter J, Guille C (2010) A Prospective Cohort Study Investigating Factors Associated With Depression During Medical Internship. *Arch Gen Psychiat* 67:557-565.
- Swartz JR, Knodt AR, Radtke SR, Hariri AR (2015) A Neural Biomarker of Psychological Vulnerability to Future Life Stress. *Neuron* 85:505-511.
- Temple J (2014) Resident duty hours around the globe: where are we now? *Bmc Med Educ* 14.
- Turker H, Riley E, Luh W-E, Colcombe S, Swallow K (2019) Estimates of locus coeruleus function with functional magnetic resonance imaging are influenced by localization approaches and the use of multi-echo data. *bioRxiv*.
- Valverde S, Oliver A, Cabezas M, Roura E, Llado X (2015) Comparison of 10 Brain Tissue Segmentation Methods Using Revisited IBSR Annotations. *J Magn Reson Imaging* 41:93-101.
- van der Wel P, van Steenbergen H (2018) Pupil dilation as an index of effort in cognitive control tasks: A review. *Psychon B Rev* 25:2005-2015.
- van Steenbergen H, Band GPH (2013) Pupil dilation in the Simon task as a marker of conflict processing. *Frontiers in Human Neuroscience* 7.
- Verguts T, Notebaert W (2009) Adaptation by binding: a learning account of cognitive control. *Trends in Cognitive Sciences* 13:252-257.
- Vinck M, Batista-Brito R, Knoblich U, Cardin JA (2015) Arousal and locomotion make distinct contributions to cortical activity patterns and visual encoding. *Neuron* 86:740-754.
- Warren CM, Eldar E, van den Brink RL, Tona KD, van der Wee NJ, Giltay EJ, van Noorden MS, Bosch JA, Wilson RC, Cohen JD, Nieuwenhuis S (2016) Catecholamine-Mediated Increases in Gain Enhance the Precision of Cortical Representations. *Journal of Neuroscience* 36:5699-5708.

REVIEWER COMMENTS

Reviewer #1 (Remarks to the Author):

The authors have responded elaborately to my previous remarks, for which I thank them. I have no reservations in endorsing publication of this article in its current form.

Erno Hermans (assisted by Rayyan Tutunji)

Reviewer #2 (Remarks to the Author):

In our first review (Reviewer 2), we were primarily concerned about 1) the level of depression and anxiety experienced, 2) analysis of predictive value of LC/pupil, and 3) accuracy in the description of pupil relation to LC activity. The authors have improved the paper substantially and now more accurately describe the level of stress and LC fMRI approach. We commend the authors on adding the additional analyses of predictive value of behavior/LC/pupil/LC-amygdala connectivity (Supplemental Figures 11-13), which we believe substantially improve the article. We believe the article as it stands now substantially advances our understanding of noradrenergic function in relation to the prediction of mild stressor response, and request only some improvements in the presentation of the model comparison Figures 7-8.

Key issue:

In rebuttal, the authors comment that "We have now added additional figures (S11-S13) reporting the ROC-AUC plots and score for the pupil alone (S11) and in concert with other factors (S12-S13). These figures show that pupil alone has some predictive validity, but also that the neural data add substantial explanatory power to the model. Based on these results, we wish to not over-interpret our pupil results."

The added plots are very useful to understanding the predictive value of LC and are complementary to Figure 7EF and 8. Unlike those figures, they make a number of key points clear: a) LC>pupil/LC-amygdala for prediction, 2) predictive value of LC is moderate but clearly improved over behavior and these other measures, 3) depression > anxiety prediction. Since the title of the article is "Predicting real-world stress resilience from the responsivity of the human locus coeruleus", I believe these points are key for the reader to judge the accuracy of this claim. Otherwise the reader may be confused about the quality of the prediction or may think LC-amygdala coupling is more impactful to the predictions than it is. Therefore, I request that the authors add the key AUC curves from Supplemental 11-13 analyses to the main text, either as an additional Figure or as part of Figures 7-8. I think S11 in particular could be simply added to Figure 8 as side panels and that would be sufficient to convey the moderate effect size and likely source (LC) of the prediction value.

Minor issues:

(line 406) "light-reflexe" should be "light reflex"

(line 397) mechanism is misspelled

Reviewer #3 (Remarks to the Author):

Grueschow and colleagues examined if resilience to stressful events (medical internship) can be predicted from fMRI responses in the brainstem noradrenergic nucleus locus coeruleus (LC), and from pupil diameter, in healthy medical students performing an emotional Stroop task. I appreciate the effort that went into the extensive control analyses that were added. However, the results in the main text were left unaltered despite known and preventable measurement issues, and the supplementary control analyses confirm an issue that I had brought up in my initial review of the manuscript. As a consequence, I remain unconvinced that the conclusions about selective involvement of the LC are justified.

1) In my previous review of this manuscript I had brought up methodological issues associated with imaging of the brainstem, and suggested ways in which some of these issues could be mitigated (a number of issues could not be mitigated because the data required to do so were not collected). While the authors have implemented the majority of these suggestions, they have done so only in control analyses that are reported in the supplement, and left the results in the main text practically unaltered. All figures in the main text are identical compared to the previous version. As a result, the main findings in the manuscript still reflect findings that are contaminated by (preventable) measurement issues that are known to occur with brainstem imaging.

2) The results of a control analysis are reported (in the supplement) where the authors have taken physiological noise into account, not applied spatial smoothing, and used a weighted average of the LC mask to obtain LC signals. In doing so, critical sources of noise are counteracted and this procedure more closely follows best practice guidelines in the field, as should be done throughout the manuscript. However, from Table S5 it is also apparent that when doing so, the LC no longer shows (significant) responses in the CI > II contrast that is used as the critical measure of response conflict. This complicates the interpretation of any correlation between LC responses and other variables. I do understand that the authors make use of inter-individual variability. But in the absence of a reliable LC response, the question is: inter-individual variability in what?

3) The lack of a significant LC response in the CI > II contrast also highlights another concern I had initially brought up: is the LC relevant for trial sequence-related behavioral effects during this task in the first place? In addition to theoretical accounts and pupil studies, the authors now cite three human neuroimaging studies (Krebs et al., 2013; Köhler et al., 2016, 2018). However, these articles suffer from the exact same methodological issues that under discussion here. Thus, while I admit that the authors have now included some evidence from the literature for the involvement of the LC in the current task, this evidence is not unequivocal. The most direct assessment of the involvement of the LC,

electrophysiological recordings (in animals), remains absent. Most importantly, the findings in the current manuscript (when noise is appropriately corrected for), actually speak against involvement of the LC.

Point-by-point response:

Reviewer #1 (Remarks to the Author):

The authors have responded elaborately to my previous remarks, for which I thank them. I have no reservations in endorsing publication of this article in its current form.

Erno Hermans (assisted by Rayyan Tutunji)

We thank the reviewer team for their important suggestions, which have substantially improved our manuscript.

Reviewer #2 (Remarks to the Author):

In our first review (Reviewer 2), we were primarily concerned about 1) the level of depression and anxiety experienced, 2) analysis of predictive value of LC/pupil, and 3) accuracy in the description of pupil relation to LC activity. The authors have improved the paper substantially and now more accurately describe the level of stress and LC fMRI approach. We commend the authors on adding the additional analyses of predictive value of behavior/LC/pupil/LC-amygdala connectivity (Supplemental Figures 11-13), which we believe substantially improve the article. We believe the article as it stands now substantially advances our understanding of noradrenergic function in relation to the prediction of mild stressor response, and request only some improvements in the presentation of the model comparison Figures 7-8.

We thank the reviewer for endorsing our work and for judging that our manuscript substantially advances the field. We appreciated the reviewers' very useful suggestions and have now included the requested ROC-curves in Figure 8 of the main manuscript. Moreover, we have added a paragraph in the main text that elaborates on the differences in predictive value of the various measures, as derived from the ROC-curve comparisons (see below for detailed response).

Key issue:

In rebuttal, the authors comment that "We have now added additional figures (S11-S13) reporting the ROC-AUC plots and score for the pupil alone (S11) and in concert with other factors (S12-S13). These figures show that pupil alone has some predictive validity, but also that the neural data add substantial explanatory power to the model. Based on these results, we wish to not over-interpret our pupil results."

The added plots are very useful to understanding the predictive value of LC and are complementary to Figure 7EF and 8. Unlike those figures, they make a number of key points clear: a) LC>pupil/LC-amygdala for prediction, 2) predictive value of LC is moderate but clearly improved over behavior and these other measures, 3) depression > anxiety prediction. Since the title of the article is "Predicting real-world stress resilience from the responsivity of the human locus coeruleus", I believe these points are key for the reader to judge the accuracy of this claim. Otherwise the reader may be confused about the quality of the prediction or may think LC-amygdala coupling is more impactful to the predictions than it is. Therefore, I request that the authors add the key AUC curves from Supplemental 11-13 analyses to the main text, either as an additional Figure or as part of Figures 7-8. I think S11 in particular could be simply

added to Figure 8 as side panels and that would be sufficient to convey the moderate effect size and likely source (LC) of the prediction value.

We agree with the reviewer that these plots are very useful for understanding the predictive value of LC activity. We have followed the reviewers' suggestion and have included them in Figure 8 of the main text (see below). In addition, we have updated the caption of figure 8 and refer to this figure in the respective parts of the main text. Finally, as suggested by the reviewer, we discuss the implications of the ROC-curves and AUC values for the predictive validity of our measures regarding anxiety and depression changes in a new paragraph in the main text. This paragraph reads:

Additional receiver operating characteristic (ROC) and area under the curve (AUC) plots (**Figure 8** and **S11-S13**) further facilitate the comparison between anxiety and depression predictions as well as between different predictors. For instance, for prediction of both anxiety and depression, these plots show that the predictive power of LC-NE (**Figure 8g, 8p**) exceeds that of pupil signals (**Figure 8i, 8r**) as well as of LC-amygdala connectivity (**Figure 8h, 8q**). While the predictive power of LC-NE for anxiety is fairly moderate (**Figure 8g**), it clearly outperforms anxiety predictions from behavioral measures (**Figure 8d**). Furthermore, LC-amygdala connectivity predictions for anxiety (**Figure 8h**) are clearly stronger than for depression (**Figure 8q**), while pupil dilation predictions for depression (**Figure 8r**) outperform the ones for anxiety (**Figure 8i**). Please see supplemental **Figures S11-S13** for ROC plots and AUC quantification for all models tested.

Updated figure 8 in main text:

Updated caption for Figure 8:

Figure 8. Comprehensive Model-Comparison for Predicting Anxiety and Depression Symptom Change

(a-c & j-l) Light grey bars show the distribution of prediction accuracies that can be expected by chance (shuffled labels, see methods sections for details). Dashed vertical lines represent the 5th and 95th percentile of this distribution. Vertical black line indicates the obtained out-of sample accuracy. **(a)** A *base-model* containing scores from anxiety and pretrauma surveys does not predict the individual mean changes in anxiety symptom severity due to real-world stress above chance (out-of-sample accuracy = 51.86%, $p = 0.234$, $R^2 = 0.08$, adjusted $R^2 = 0.037$). **(b)** Using a *full model* that additionally contains behavioral-, neural- and pupil data predicts mean anxiety increases significantly above chance (out-of-sample accuracy = 58.7%, $p < 0.001$, $R^2 = 0.57$, adjusted $R^2 = 0.50$). Compared to the base-model, the full model increases the explained variance by 49% and the adjusted explained variance by 47%. Locus coeruleus contribution is significant ($p=0.038$) **(c)** The *optimal model*, established using a stepwise regression procedure (Methods), shows similar prediction improvements (out-of-sample accuracy = 59.2%, $p < 0.001$, $R^2 = 0.56$, adjusted $R^2 = 0.52$) but comprises only four parameters: locus coeruleus upregulation response ($p=0.025$), behavioral congruency sequence effect (CSE, $p=0.031$), pupil ($p=0.05$) and LC-NE-Amygdala coupling during the upregulation response ($p<0.001$). Compared to the base-model, this sparse model predicts 49% more of the variance and also 48% more of the adjusted variance. Please see supplemental table S2 for additional models, full details on single regressor contributions, and model comparison. **(d-i)** Receiver operating characteristic (ROC) plots and area under the curve (AUC) for different combinations of measures predicting anxiety: **(d)** Base-model, **(e)** Full-model, **(f)** Optimal model, **(g)** LC-only, **(h)** LC-Amygdala only, **(i)** pupil only. **(j)** The *base-model* predicts mean depression symptom increases significantly above chance (out-of-sample accuracy = 67.38%, $p < 0.001$, $R^2 = 0.27$ adjusted $R^2 = 0.23$). The PHQ-survey score is already a significant predictor for depression symptom severity changes ($p=0.0002$). **(k)** The *full model* containing additional behavioral-, neural- and pupil-data predicts mean depression increases significantly above chance (out-of-sample accuracy = 64.36%, $p < 0.001$, $R^2 = 0.37$, adjusted $R^2 = 0.28$) and increases the explained variance by 11% and the adjusted explained variance by 4.3%. **(l)** The *optimal model* has similar prediction improvements as the full model (out-of-sample accuracy = 67.7%, $p < 0.001$, $R^2 = 0.33$, adjusted $R^2 = 0.30$) but contains only two parameters: locus coeruleus upregulation response ($p=0.039$) and the PHQ-depression survey ($p=0.0009$). Compared to the base-model, this sparse model predicts 10% more of the variance and also 10% of the adjusted variance. **(m-r)** Receiver operating characteristic (ROC) plots and area under the curve (AUC) for different combinations of measures predicting depression: **(m)** Base-model, **(n)** Full-model, **(o)** Optimal model, **(p)** LC-only, **(q)** LC-Amygdala only, **(r)** pupil only. Please see supplemental table S3 for additional models, full details on single regressor contributions and model comparison.

To point the reader to the additional ROC-AUC data as well as to the new figure panels, we now provide these statements in the respective results sections:

In the LC-NE results section:

To compare between anxiety and depression predictions, and to compare the predictive validity of different resilience predictors, please see the receiver-operator characteristic curves (ROC) and the associated area under the curve (AUC) plots in Figure 8g and 8p as well as supplemental Figure S11-S13.

In the LC-NE-amygdala connectivity results section:

To compare between anxiety and depression predictions, and to compare the predictive validity of different resilience predictors, please see the receiver-operator characteristic curves (ROC) and the associated area under the curve (AUC) plots in Figure 8h and 8q as well as supplemental Figure S11-S13.

In the pupil results section:

To compare between anxiety and depression predictions, and to compare the predictive validity of different resilience predictors, please see the receiver-operator characteristic curves (ROC) and the associated area under the curve (AUC) plots in Figure 8i and 8r as well as supplemental Figure S11-S13.

Minor issues:

(line 406) "light-reflexe" should be "light reflex"

(line 397) mechanism is misspelled

Thank you very much indeed. These typos have been fixed.

Reviewer #3 (Remarks to the Author):

Grueschow and colleagues examined if resilience to stressful events (medical internship) can be predicted from fMRI responses in the brainstem noradrenergic nucleus locus coeruleus (LC), and from pupil diameter, in healthy medical students performing an emotional Stroop task. I appreciate the effort that went into the extensive control analyses that were added.

We are glad that the reviewer appreciates the considerable effort that went into the control analyses that support our main conclusions.

However, the results in the main text were left unaltered despite known and preventable measurement issue, and the supplementary control analyses confirm an issue that I had brought up in my initial review of the manuscript. As a consequence, I remain unconvinced that the conclusions about selective involvement of the LC are justified.

We understood the reviewers remarks as a request for control analyses, which are usually added to the supplement to bolster a manuscript's conclusions. However, the reviewer's

reaction to our revision makes it clear that we should rather embed these analyses in the main results in the paper. This is no problem and we have now extended the LC specific results in the main text as requested. More specifically:

(1) We have updated several main text figures (Figure 1, Figure 3, Figure 4 and Figure 8) and have added a supplementary figure 14 and the associated statistics in a new supplementary table. For instance, we have completely revised figure 3, which illustrates the relationship between LC and symptom severity changes due to prolonged stress exposure, but we now display the data, as requested, from physiological-noise-corrected, unsmoothed data using probability weighting in the LC-1SD-Mask.

(2) We also have followed the request to add to the main text the local specificity analysis involving additional brainstem clusters and physiological-noise-corrected, unsmoothed data.

(3) We have now also followed the request to report the results from the temporal signal-to-noise analysis in the main text.

(4) We have updated Figure 1 with an additional analysis addressing the reviewer's concern about the LC involvement in trial sequence effects, showing that there is indeed higher activity for the (CI>II) comparisons in high risk individuals, also as compared to low risk individuals. These analyses employ weighted averaging in LC-1SD mask voxels from unsmoothed physio-corrected data, as requested. This confirms that the LC is indeed involved in conflict adaptation, and specifically so for people who are at risk of developing psychopathology.

(5) We have added an additional supplementary figure 14 and a statistics table that show that this relationship using unsmoothed data is also independent of the specific LC-Mask employed.

(6) We have updated the comprehensive model comparison section in the main text and in the supplementary materials. We have now updated figure 8, its caption, and associated results sections in the main text as well as the related supplementary tables S2 and S3. These now only contain LC specific results obtained from the weighted average of LC-1SD mask voxels from the physio-corrected, unsmoothed fMRI data. All conclusions regarding our out-of-sample predictions remain unaltered.

Together, these results corroborate the importance of LC in conflict adaptation and show that individual differences in LC responsivity, as identified using our conflict control task, predict stress resilience. We hope the reviewer appreciates that we implemented all of her/his suggestions and now concurs with the other reviewers that our manuscript is conclusive and of interest to the field. In the following, we respond to each of the points raised by the reviewer in more detail.

1) In my previous review of this manuscript I had brought up methodological issues associated with imaging of the brainstem, and suggested ways in which some of these issues could be mitigated (a number of issues could not be mitigated because the data required to do so were not collected). While the authors have implemented the majority of these suggestions, they have done so only in control analyses that are reported in the supplement, and left the results in the main text practically unaltered. All figures in the main text are identical compared to the

previous version. As a result, the main findings in the manuscript still reflect findings that are contaminated by (preventable) measurement issues that are known to occur with brainstem imaging.

We had understood the reviewer’s initial comments as a request for control analyses. Because these were very extensive, and in line with common practice, we added them as supplementary results. We now understand that the reviewer wants us to add these results to the main text and, with the benefit of hindsight, we agree with this recommendation. We have therefore substantially updated the main text results section. First, to deemphasize the notion that data analyses using physiological noise-correction and unsmoothed data via weighted averaging are mere control analyses, we have now replaced the term ‘control analyses’ with ‘optimizing brainstem signals**’ or ‘**additional analyses**’, in both the main text and supplementary materials.**

We have updated the LC results section substantially, with particular focus on optimization of brain stem imaging as well as emphasizing the results extracted from physiological noise-correction and unsmoothed data via weighted averaging in the text and in the related figure and figure caption (see below). We have completely reworked figure 3, figure 4 and figure 8, which now illustrate physiological-noise-corrected unsmoothed data extracted and probability weighted from the LC-1SD-Mask.

The results section in the main text now reads:

Optimizing brainstem signals

Optimal functional imaging of the brainstem, and in particular the LC, is notoriously difficult due to the small size of the nuclei involved, their proximity to the ventricles, and inherently low signal-to-noise ratio in the brainstem. In order to unequivocally identify LC-NE activity, non-standard techniques would be ideal for both data acquisition and analysis (Brooks et al., 2013, Forstmann et al., 2017, Turker et al., 2019) On the acquisition side this would, for instance, entail high-field imaging and partial-brain coverage, which allows particularly small (submillimeter) voxel resolution to avoid partial voluming and reduce pulsating artefacts from the adjacent 4th ventricle (please see the limitations section for a discussion of fundamental methodological steps to improve brainstem imaging). However, use of such a specialized imaging protocol would preclude whole-brain imaging and therefore inferences about influences of the LC on other brain systems (e.g., the amygdala and neocortical areas involved in conflict processing). Moreover, it would make it difficult for our approach to be replicated and extended in standard fMRI lab settings around the world. Thus, we opted for a standard 3T scanner and a routine fMRI-sequence with relatively low voxel resolution (2.5 mm isotropic) that nevertheless retains good signal-to-noise ratio in the brain stem (supplemental figures S8-S10). Importantly, to ascertain the specificity of our results to the LC, we conducted multiple mutually corroborating analyses. These included weighted averaging for data extraction from brainstem regions-of-interest, control for additional brainstem-nuclei, controlling for physiological nuisance variables based on principle component analysis of individually identified CSF probability tissue classes, as well as applying these nuisance variables in additional regression models to both smoothed and unsmoothed data. Please see the methods and supplementary methods section for detailed descriptions of these techniques, additional Figures (S7-S10) and statistical results tables (Table S5 and S6), as well as a formal temporal-signal-to-noise (tSNR) analysis of the whole brain and specifically the brainstem.

LC-NE responsivity predicts stress-related anxiety and depression symptom change

To demonstrate that the LC is indeed reliably involved in the CI>II contrast, while at the same time taking individual differences in LC conflict responsivity into account, we split the sample into participants who went on to develop stronger vs weaker mean anxiety/depression symptoms (median-split). This allowed us to analyze LC responses (the weighted average LC-1SD extracted, physio-corrected, unsmoothed fMRI data) to our conflict task in people with high versus low susceptibility to develop psychopathology in response to stress. Given our hypothesis - derived from rodent studies - that hyper-responsivity of the LC-NE predisposes vulnerability to prolonged stress exposure, we expect participants with high symptom severity changes to also show high LC responsivity, while participants that exhibit less or no changes in symptom severity changes are expected to show low LC responsivity. Indeed, we found that participants with high symptom severity changes exhibited significant LC-NE responsivity (CI>II) that was significantly stronger than the corresponding effect in participants with low symptom severity changes (Figure 1e). These effects were similarly present for both symptom types (Anxiety: high symptom changes group: $df=22$, $T=2.437$; $p=0.023$; low symptom changes group: $df=24$, $T=-0.895$; $p=0.379$; high vs. low symptom changes groups: $df=46$; $T=2.431$; $p=0.019$; Depression: high symptom changes group: $df=20$; $T=2.21$; $p=0.039$; low symptom changes group: $df=26$; $T=-0.611$; $p=0.546$; high vs. low symptom changes groups: $df=46$; $T=2.154$; $p=0.037$) and types of LC-NE mask choice (see Figure S14 for comparison between masks). Thus, the results of this analysis confirm that the LC is indeed involved in response conflict adaptation, specifically for people who go on to develop stronger subsequent psychopathological symptoms. This validates our measure and already suggests that it may be useful for predicting the development of stress-related psychopathology.

To formally establish this predictive validity of conflict-induced LC-NE responsivity for stress resilience, we correlated the participants' symptom severity changes at 3 and 6 months (**Figure 2**) with their individual fMRI-BOLD-amplitude during conflict-induced upregulation (CI>II) in the locus coeruleus (extracted from physiological noise corrected, unsmoothed data with weighted averaging across voxels in the LC-1SD-mask (Keren et al., 2009)). Individual LC-NE responsivity indeed correlated significantly with anxiety- and depression score changes measured three and six months into the internship as well as with the mean symptom changes across 3 and 6 months ($df = 47$, t_1 , anxiety: $Rho = 0.30$, $p = 0.018$, depression: $Rho = 0.38$, $p = 0.004$, t_2 , anxiety: $Rho = 0.31$, $p = 0.002$, depression: $Rho = 0.26$, $p = 0.034$; mean between t_1 and t_2 , anxiety: $Rho = 0.30$, $p = 0.002$, depression: $Rho = 0.36$, $p = 0.006$, non-parametric Spearman's rank correlation coefficient and robust regression, **Figure 3a-f**). That is, smaller conflict responses in the LC-NE system to the CI>II contrast were associated with less anxiety and depression symptom change, and thus more resilience, during the subsequent internship.

To ensure predictive relevance and local specificity for the locus coeruleus, we compared symptom change predictions for anxiety and depression between analyses employing two types of LC-masks (1SD and 2SD), as well as for analyses based on activity extracted from several other brainstem nuclei in the vicinity of the LC, i.e.: medial raphe nucleus (MR), dorsal raphe nucleus (DR), and ventral tegmental area (VTA). We also compared the predictive power of LC signals with that of signals extracted from the substantia nigra (SN), and the amygdala (please see supplemental methods details and Figure S10 for a visualization of these brainstem structures). In addition, we tested whether these predictions hold for LC-extracted weighted averages from different analysis pipelines with or without spatial smoothing and physiological noise correction, respectively.

The additional results reveal that stress-related anxiety and depression symptom changes were best predicted by the locus coeruleus, compared to all other brainstem nuclei. They also underline the robustness of our results in several ways. For example, the choice of LC mask did not bias the results: The LC was the only structure to predict symptom changes, for both

available types of standardized LC masks (1SD & 2SD; the smaller and more robust map (1SD) yields the stronger correlations). Physio-correction generally improved the statistics in nuclei closest to the 4th ventricle, such as the LC, medial raphe and dorsal raphe (in fact, in the physio-corrected smoothed data, DR and MR correlate with symptom changes as well). However, the physio-corrected un-smoothed data (as reported in detail above and illustrated in Figure 3a-f) showed that the LC was the only region predicting both anxiety and depression changes, irrespective of LC-mask choice, suggesting that DR and MR correlations in the smoothed data may stem from a smearing of LC activity into these neighboring regions. These additional analyses further strengthen our main conclusion that stress resilience is predicted by responsiveness of the LC. We summarize the results of all these analyses in the supplemental table S5 (also see supplemental figure S7-S13 illustrating our additional analyses).

We also conducted a formal analysis of the temporal signal-to-noise ratio (tSNR) across the whole brain, and in particular in the brainstem, making it easier to assess the signal quality of the extracted LC signals in comparison with other brainstem structures. The tSNR was computed by dividing the mean of each time series by its standard deviation for each voxel in the brain. The results confirmed that both the average and subject-specific tSNR in the LC was well above standard cut-offs (>30). We also found that the signal in the LC was in fact strongest amongst all brainstem nuclei, for both standard LC masks (1SD & 2SD, see supplemental figure S8-S11).

Updated figure 3 and caption:

Figure 3. LC-NE responsivity (CI>II) relates to increases in anxiety and depression due to prolonged real-world stress exposure

Each panel visualizes the correlation between participants' symptom severity and individual CI>II responses, extracted from physiological-noise-corrected, unsmoothed data with weighted averaging across voxels in the LC-1SD-mask. Please note that the symptom severity used for these correlations is defined as the change from the individual symptoms baseline level at measurement time t_0 . **(A-C)** Correlation between LC responsivity (CI>II) and severity of anxiety symptom changes (STAI) (top) measured after 3 months **(a)** and 6 months **(b)** of exposure to real-world chronic stress as well as the mean change between both measurement time points **(c)**. **(d-f)** Same as **(a to c)**, but for severity of depression symptoms change (PHQ).

Next, we formally tested the predictive validity of the individual LC-NE upregulation response for symptom changes in the population. We first compared the observed symptom severity change with the predicted change score in an out-of-sample fashion. To do so, we estimated a linear regression of psychological test score data on neural CI>II responses (**weighted average LC-1SD extracted, physio-corrected, unsmoothed data**) for the data of all participants excluding the current participant (Kriegeskorte et al., 2009, Poldrack and Mumford, 2009, Esterman et al., 2010) and then used this fitted model to predict for the left-out participant the individual mean change in symptom severity. **For simplicity, we focus on the mean symptom changes (mean across 3 and 6 months) in the remainder of the manuscript.** A significant correlation in this out-of-sample procedure indicates that across the population, LC-NE responsivity can reliably predict individual stress-resilience in the future (Pineiro et al., 2008). We did observe such predictive validity: **For both anxiety and depression, predicted symptom severity changes correlated with the observed symptom changes ($df = 47$, anxiety: $Rho = 0.25$, $p = 0.01$, depression: $Rho = 0.28$, $p = 0.05$, non-parametric Spearman's rank correlation coefficient and robust regression, **Figure 4a and 4c**).**

As a second step, we tested whether we can predict from the measure of LC-NE responsivity (**weighted average LC-1SD extracted, physio-corrected, unsmoothed data**) which out of two randomly chosen participants will be more resilient, i.e., incur a smaller symptom change after experiencing real-world stress. A leave-two-subjects out procedure (LTSO, see **Methods**) showed that the individual LC-NE upregulation response predicts above chance which subject developed higher anxiety symptom change (**prediction accuracy 60.3%, $p < 0.001$, **Figure 4c****). Similarly, LC-NE upregulation responses also predicted above chance which subject developed higher depression symptom change due to real-life stress (**prediction accuracy 59.4, $p < 0.001$, **Figure 4d****). To compare between anxiety and depression predictions, and to compare the predictive validity of different resilience predictors, please see the receiver-operator characteristic curves (ROC) and the associated area under the curve (AUC) plots in **Figure 8G and 8P** as well as supplemental **Figure S11-S13**.

Updated figure 4 and caption:

Figure 4. Out-of-sample prediction of mean symptom severity changes.

(a) Correlation between out-of-sample predicted and observed mean anxiety symptom severity changes due to emergency room internship stress. (b) LC-NE upregulation responses predict significantly above chance which of two subjects left out of the estimation will show stronger mean anxiety increases as a consequence of stress. Dashed lines indicate the 5th and 95th percentile of the randomized labels distribution, respectively. Thick black vertical line indicates the obtained prediction accuracy. (c-d) as in (a-b) but for depression symptom changes. Please see **Methods** section for details.

Moreover, related to the comprehensive model comparisons we conducted, we have now updated figure 8, its caption, and associated results sections in the main text as well as the related supplementary tables S2 and S3. These now only contain LC-specific results obtained from the weighted average of LC-1SD mask voxels from the physio-corrected, unsmoothed fMRI data.

The results section now reads:

Locus Coeruleus responsivity is a robust and reliable bio-marker for stress resilience

In a final analysis, we quantified and compared the usefulness of the identified bio-markers for predicting stress resilience by first comparing their predictive validity to that of a base-model

(the current gold standard: self-report surveys of previous potentially traumatic experiences or current symptoms) using a multiple GLM-approach. We also identified the most parsimonious parameter combinations for predicting individual anxiety or depression symptom change, by means of a stepwise-regression approach (**Methods**; for a comprehensive list of parameter test-statistics, goodness-of-fit measures, and model comparisons please see **Supplemental Tables S2 and S3**). Finally, we compared the out-of-sample prediction accuracy between the base model, full model (containing all parameters), and most parsimonious model using LTSO (**Methods**). **Please note that the LC specific regressor was extracted using the weighted average LC-1SD mask from the physio-corrected, unsmoothed fMRI data (Tables S2 and S3). For completeness, supplemental Tables S9 and S10 report the full list of statistics for data without these corrections.**

These analyses showed that our identified bio-markers substantially improved predictions of anxiety symptom changes as compared to the gold-standard base-model. The adjusted explained variance was increased by 400% and 300%, respectively, when we added either LC ($p=0.017$) or pupil ($p=0.039$) to the regression. The classic behavioral congruency-sequence effect (CSE) was neither significant on its own ($p>0.1$, model 2) nor in models containing either LC (model 3) or pupil (model 4). Having both LC and pupil regressors in one model explaining anxiety changes (model 5) further increased the explained adjusted variance (by about 20%); **this model established LC ($p=0.02$) and pupil ($p=0.04$) as reliable predictors for anxiety changes.** Importantly, adding the individual connectivity strength between LC and amygdala during the upregulation response (model 6) lead to another increase in adjusted explained variance (another 50%, resulting in approximately 12 times the variance explained by the base-model) and above-chance out-of-sample predictions ($p<0.001$, **58.7%**, **Figure 8b**). These results thus establish both LC responsivity ($p=0.038$) and LC-amygdala-connectivity ($p<0.001$) during upregulation as important biological predictors for anxiety symptom changes and thus stress resilience. The usefulness of these variables was further underscored by the fact that the most parsimonious model contained LC-connectivity ($p<0.001$), LC ($p=0.025$), pupil ($p=0.053$) and the behavioural CSE ($p<0.031$). This model delivered the highest adjusted explained variance of **51.8%** and predicted symptom severity change out-of sample ($p<0.001$, **59.2%**, **Figure 8c**).

For depression symptom severity changes, the LC conflict response was also the most reliable predictor, even though the base model already explained 23.3% adjusted variance, primarily due to the PHQ-depression score at T_0 ($p=0.0002$, model 1, see **supplemental results** for details). On top of this established measure, the individual LC upregulation response was the only biological marker that reliably related to depression symptom changes ($p = 0.046$), even when controlling for behavioral CSE ($p=0.88$), pupil distance ($p=0.14$), or LC-connectivity ($p=0.74$). The LC upregulation regressor added 4% of adjusted variance (27.1%, model 3) to that achieved by the base model; this was similar to the variance explained by the full model including all parameters (27.6%, model 6, with **64.4%** out-of-sample accuracy **Figure 8k**). LC ($p=0.039$) and PHQ score at T_0 ($p=0.0009$) were also the the only two markers identified by the most parsimonious model, which explained **30.2%** adjusted variance and significantly predicted mean symptom severity changes out-of-sample ($p<0.001$, **67.7%** accuracy, **Figure 8l**). These results were also robust to non-prospective factors such as the number and severity of adverse events experienced during the internship (please see supplemental information for details and supplemental table S6 for comprehensive statistics).

Updated figure 8 and caption:

Figure 8. Comprehensive Model-Comparison for Predicting Anxiety and Depression Symptom Change

(a-c & j-l) Light grey bars show the distribution of prediction accuracies that can be expected by chance (shuffled labels, see methods sections for details). Dashed vertical lines represent the 5th and 95th percentile of this distribution. Vertical black line indicates the obtained out-of sample accuracy. (a) A *base-model* containing scores from anxiety and pretrauma surveys does not predict the individual mean changes in anxiety symptom severity due to real-world stress above chance (out-of-sample accuracy = 51.86%, $p = 0.234$, $R^2 = 0.08$, adjusted $R^2 = 0.037$). (b) Using a *full model* that additionally contains behavioral-, neural- and pupil data predicts mean anxiety increases significantly above chance (out-of-sample accuracy = 58.7%, $p < 0.001$, $R^2 = 0.57$, adjusted $R^2 = 0.50$). Compared to the base-model, the full model increases the explained variance by 49% and the adjusted explained variance by 47%. Locus coeruleus contribution is significant ($p=0.038$) (c) The *optimal model*, established using a stepwise regression procedure (Methods), shows similar prediction improvements (out-of-sample accuracy = 59.2%, $p < 0.001$, $R^2 = 0.56$, adjusted $R^2 = 0.52$) but comprises only four parameters: locus coeruleus upregulation response ($p=0.025$), behavioral congruency sequence effect (CSE, $p=0.031$), pupil ($p=0.05$) and LC-NE-Amygdala coupling during the upregulation response ($p<0.001$). Compared to the base-model, this sparse model predicts 49% more of the variance and also 48% more of the adjusted variance. Please see supplemental table S2 for additional models, full details on single regressor contributions, and model comparison. (d-i) Receiver operating characteristic (ROC) plots and area under the curve (AUC) for different combinations of measures predicting anxiety: (d) Base-model, (e) Full-model, (f) Optimal model, (g) LC-only, (h) LC-Amygdala only, (i) pupil only. (j) The *base-model* predicts mean depression symptom increases significantly above chance (out-of-sample accuracy = 67.38%, $p < 0.001$, $R^2 = 0.27$ adjusted $R^2 = 0.23$). The PHQ-survey score is already a significant predictor for depression symptom severity changes ($p=0.0002$). (k) The *full model* containing additional behavioral-, neural- and pupil-data predicts mean depression increases significantly above chance (out-of-sample accuracy = 64.36%, $p < 0.001$, $R^2 = 0.37$, adjusted $R^2 = 0.28$) and increases the explained variance by 11% and the adjusted explained variance by 4.3%. (l) The *optimal model* has similar prediction improvements as the full model (out-of-sample accuracy = 67.7%, $p < 0.001$, $R^2 = 0.33$, adjusted $R^2 = 0.30$) but contains only two parameters: locus coeruleus upregulation response ($p=0.039$) and the PHQ-depression survey ($p=0.0009$). Compared to the base-model, this sparse model predicts 10% more of the variance and also 10% of the adjusted variance. (m-r) Receiver operating characteristic (ROC) plots and area under the curve (AUC) for different combinations of measures predicting depression: (m) Base-model, (n) Full-model, (o) Optimal model, (p) LC-only, (q) LC-Amygdala only, (r) pupil only. Please see supplemental table S3 for additional models, full details on single regressor contributions and model comparison.

Finally, an updated Methods section now features all additional analyses conducted to optimize brainstem signals in the main text.

The added methods section now reads:

We controlled for physiological noise with nuisance regressors that reflected the time-course within the cerebrospinal fluid (CSF) (Bazin et al., 2019). The CSF mask was generated for each individual by the non-linear unified segment procedure in SPM12 (see above). Time series were extracted for all voxels included in this mask and were submitted to principle component analysis using the matlab function `pca.m` included in the statistics toolbox (MATLAB, The MathWorks, Inc., Natick, Massachusetts, U.S.). The first five principle components for each participant were used as nuisance regressors in the GLM analysis, alongside 6 motion regressors. Supplemental Figure S7 provides a visualization for the first 2 participants and supplemental Table S5 reports the improved statistical results for predicting symptom severity changes following application of this technique.

We ensured predictive relevance and local specificity for the locus coeruleus by comparing anxiety and depression symptom change predictions based on both LC-masks (1SD and 2SD) and several other brainstem nuclei. In addition, we employed a weighted-average data extraction that weighed every voxel's activity with the probability of membership in the ROI assigned to each voxel. These probabilistic maps included the main brainstem nuclei in the vicinity of the LC, i.e.: medial raphe nucleus (MR), dorsal raphe nucleus (DR), and ventral tegmental area (VTA) provided by the Harvard Ascending Arousal atlas available at <https://www.martinos.org/resources/aan-atlas>. We also compared LC prediction power with the substantia nigra (SN), available at <https://www.nitrc.org/projects/ataq/> and the amygdala (<https://fsl.fmrib.ox.ac.uk/fsl/fslwiki/Atlases>). In addition, we repeated all GLMs also for unsmoothed data, since the 6mm smoothing kernel we applied may have smeared the activity between brain-stem nuclei as well as with adjacent CSF. Please note that all LC-specific results reported in the main text are based on a weighted average of LC-1SD mask voxels extracted from the physiological-noise-corrected, unsmoothed fMRI data. Applying these optimized brainstem signal analysis techniques substantially enhanced data quality and statistical significance, and substantiated our conclusions that the responsiveness of the human LC-NE arousal system predicts anxiety and depression symptom severity changes induced by relatively mild levels of real-world occupational stress.

2) The results of a control analysis are reported (in the supplement) where the authors have taken physiological noise into account, not applied spatial smoothing, and used a weighted average of the LC mask to obtain LC signals. In doing so, critical sources of noise are counteracted and this procedure more closely follows best practice guidelines in the field, as should be done throughout the manuscript. However, from Table S5 it is also apparent that when doing so, the LC no longer shows (significant) responses in the CI > II contrast that is used as the critical measure of response conflict. This complicates the interpretation of any correlation between LC responses and other variables. I do understand that the authors make use of inter-individual variability. But in the absence of a reliable LC response, the question is: inter-individual variability in what?

We agree that the reader would benefit from a demonstration that the LC is indeed reliably involved in the CI>II contrast, while at the same time taking into account that people differ substantially in this responsiveness. We have therefore now followed the suggestion by another reviewer and provide this demonstration in a new analysis. In this analysis, we median split the sample into participants developing stronger vs weaker anxiety/depression, and analyze the CI>II contrast in the weighted average LC-1SD extracted, physio-corrected unsmoothed data. Given our hypothesis derived from rodent studies, the participants developing stronger symptoms are expected to show high LC responsiveness and participants developing weak symptoms are expected to show low LC responsiveness. The results of this analysis confirm these predictions and establish the LC involvement in response conflict: People with stronger subsequent symptoms show significant LC activation for the CI > II contrast, in line with their hypothesized increased LC responsiveness to response conflict, and this signal is significantly stronger than that in people who go on to develop weaker symptoms, already suggesting the predictive validity of this signal for subsequent psychopathology. We now report this analysis in the main text and illustrate it in figure 1e. Supplementary figure 14 and the associated table provide evidence that these results are also robust to the choice of LC-mask. Together, these new results therefore show that the LC signal in our study relates to response conflict and that increased conflict signals predict psychopathology.

The main text results section now reads:

To demonstrate that the LC is indeed reliably involved in the CI>II contrast, while at the same time taking individual differences in LC conflict responsivity into account, we split the sample into participants who went on to develop stronger vs weaker mean anxiety/depression symptoms (median-split). This allowed us to analyze LC responses (the weighted average LC-1SD extracted, physio-corrected, unsmoothed fMRI data) to our conflict task in people with high versus low susceptibility to develop psychopathology in response to stress. Given our hypothesis - derived from rodent studies - that hyper-responsivity of the LC-NE predisposes vulnerability to prolonged stress exposure, we expect participants with high symptom severity changes to also show high LC responsivity, while participants that exhibit less or no changes in symptom severity changes are expected to show low LC responsivity. Indeed, we found that participants with high symptom severity changes exhibited significant LC-NE responsivity (CI>II) that was significantly stronger than the corresponding effect in participants with low symptom severity changes (Figure 1e). These effects were similarly present for both symptom types (Anxiety: high symptom changes group: $df=22$, $T=2.437$; $p=0.023$; low symptom changes group: $df=24$, $T=-0.895$; $p=0.379$; high vs. low symptom changes groups: $df=46$; $T=2.431$; $p=0.019$; Depression: high symptom changes group: $df=20$; $T=2.21$; $p=0.039$; low symptom changes group: $df=26$; $T=-0.611$; $p=0.546$; high vs. low symptom changes groups: $df=46$; $T=2.154$; $p=0.037$) and types of LC-NE mask choice (see Figure S14 for comparison between masks). Thus, the results of this analysis confirm that the LC is indeed involved in response conflict adaptation, specifically for people who go on to develop stronger subsequent psychopathological symptoms. This validates our measure and already suggests that it may be useful for predicting the development of stress-related psychopathology.

Updated figure 1 and replaced caption for panel 1e:

Figure 1. Experimental task and neural conflict-induced upregulation responses (CI>II).

(a): Example stimuli illustrating all four possible face/word combinations in the emotional-stroop task. Face stimuli used in our experiment were identical to the face stimuli used in Etkin et al. 2006. For illustrative purposes, we have replaced these images here with open access face stimuli (<https://faces.mpdl.mpg.de/imeji/>). Participants were instructed to react to the facial expression while ignoring the overlaid word and to answer as fast and accurately as possible. On each trial, the word color was randomly assigned in order to avoid adaptation effects. (b): Trial presentation schedule. A CI-trial is an incongruent trial preceded by a congruent trial. An II-trial is an incongruent trial preceded by an incongruent trial. Subtracting neural responses for II from CI trials reveals regions involved in the upregulation response (CI>II), while subtracting neural responses for CI from II trials reveals regions associated with implicit conflict adaptation (II>CI). See **Supplemental Methods** for details on stimulus presentation and counterbalancing of conditions. (c) Cortical and subcortical regions involved in generating an upregulation response to resolve conflict. Mid-sagittal slice with activation clusters shows higher activity to incongruent trials preceded by a congruent trial (CI) as compared to incongruent trials preceded by an incongruent trial (II) (left superior temporal cortex (STC), posterior

cingulate cortex (PCC), anterior visual cortex and a large subcortical cluster, FWE-cluster-correction at $p=0.05$ with cluster-forming-threshold at $p=0.001$, the pseudo-color-map illustrating the one-sample- t -statistic applies to all panels). Inset shows magnified lateral-view of subcortical cluster and an overlaid locus coeruleus mask in green (2SD-mask from Keren et al., 2009 (Keren et al., 2009)). **(d)** Coronal view of standard brain and magnified view of bilateral LC upregulation response (hot colors) overlaid with LC-mask (green). **(e)** Participants with high subsequent anxiety/depression symptom changes show significantly stronger LC-NE responsivity ($CI>II$) than participants with lower symptom changes (median split). Bar plots show the LC-responsivity strength extracted from the $CI>II$ contrast in the physiological noise controlled, unsmoothed data as weighted-average of LC-1SD mask-voxels (see supplemental figure S14 for detailed statistics and comparison with LC-2SD mask voxels).

Supplemental Figure S14:

LC_2SD (CI>II)				
	Anxiety		Depression	
	T	p	T	p
High Risk	2.756	0.012	2.404	0.026
Low Risk	0.518	0.609	0.104	0.918
High > Low Risk	2.435	0.019	1.989	0.050
LC_1SD (CI>II)				
	Anxiety		Depression	
	T	p	T	p
High Risk	2.437	0.023	2.21	0.039
Low Risk	0.895	0.379	0.611	0.546
High > Low Risk	2.431	0.019	2.154	0.037

Strength of LC-responsivity based on subsequent symptom severity change

Strong changes in anxiety/depression symptoms are accompanied by significantly stronger LC-NE responsivity (CI>II) irrespective of LC-mask. Bar plots show the strength of LC-responsivity (CI>II contrast), extracted from the physiological-noise-controlled, unsmoothed data, as weighted-average of LC-2SD (A) and LC-1SD (B) mask-voxels, respectively.

3) The lack of a significant LC response in the CI > II contrast also highlights another concern I had initially brought up: is the LC relevant for trial sequence-related behavioral effects during this task in the first place? In addition to theoretical accounts and pupil studies, the authors now cite three human neuroimaging studies (Krebs et al., 2013; Köhler et al., 2016, 2018). However, these articles suffer from the exact same methodological issues that under discussion here. Thus, while I admit that the authors have now included some evidence from the literature for the involvement of the LC in the current task, this evidence is not unequivocal. The most direct assessment of the involvement of the LC, electrophysiological recordings (in animals), remains absent. Most importantly, the findings in the current manuscript (when noise is appropriately corrected for), actually speak against involvement of the LC.

The new analysis added to the manuscript - derived using data that appropriately corrects for noise as suggested by the reviewer - now clearly demonstrates the involvement of the LC in trial-sequence conflict adaptation effects, specifically for people who go on to develop psychopathological symptoms. Thus, the data in the current manuscript provide clear support that the LC is involved in conflict sequence effects, in a manner that predicts vulnerability to stress-related psychopathology.

With respect to congruence of our findings with the previous literature, we are glad to see that the reviewer acknowledges that our new conclusions are in line with what could be expected based on the results of previous fMRI studies. However, we were somewhat surprised to read the reviewer's comments about the lack of findings from animal electrophysiology, given that it is often difficult to conduct completely parallel tasks in animals and humans, and that the existence of very similar findings in animals would be seen by many readers as compromising the novelty of our results. Nevertheless, we have inspected the literature even more closely and are happy to say that we have found further studies that used conflict tasks in monkeys to investigate pupil dilation along with intracranial recordings, albeit in the dACC, a region strongly connected to LC-NE (Ebitz et al., 2014, Ebitz and Platt, 2015). These studies found that conflict-related signals in the dACC predicted subsequent reduced distractor interference and changes in pupil size, interpreted as a peripheral index of arousal linked to noradrenergic tone. Thus, this study replicates in the monkey what many human neuroimaging studies as well as human pupil studies have reported. These authors state that their findings provide neurophysiological endorsement of the hypothesis that conflict is regulated, in part, via modulation of pupil-linked processes such as arousal. To provide a closer link of our findings to this existing evidence in animals, we therefore now write in the introduction:

Considerable evidence suggests that such conflict-related signals engage the arousal system in monkeys and humans. For instance, conflict signals in the macaque brain, induced via task-congruent and -incongruent stimuli of monkey faces, predicted subsequent changes in pupil size and reduced behavioral distractor interference (Ebitz et al., 2014, Ebitz and Platt, 2015). These data are consistent with the hypothesis that pupil-linked arousal mechanisms regulate conflict adjustments in non-human primates (Ebitz et al., 2014, Ebitz and Platt, 2015).

In closing, we would like to thank all reviewers for the many insightful comments and suggestions that have helped us to improve our manuscript. We hope all reviewers now find our study conclusive, interesting, and ready for publication.

References

- Bazin PL, Alkemade A, van der Zwaag W, Caan M, Mulder M, Forstmann BU (2019) Denoising High-Field Multi-Dimensional MRI With Local Complex PCA. *Front Neurosci-Switz* 13.
- Brooks JCW, Faull OK, Pattinson KTS, Jenkinson M (2013) Physiological noise in brainstem fMRI. *Frontiers in Human Neuroscience* 7.
- Ebitz RB, Pearson JM, Platt ML (2014) Pupil size and social vigilance in rhesus macaques. *Front Neurosci* 8:100.
- Ebitz RB, Platt ML (2015) Neuronal activity in primate dorsal anterior cingulate cortex signals task conflict and predicts adjustments in pupil-linked arousal. *Neuron* 85:628-640.
- Esterman M, Tamber-Rosenau BJ, Chiu YC, Yantis S (2010) Avoiding non-independence in fMRI data analysis: Leave one subject out. *Neuroimage* 50:572-576.
- Forstmann BU, de Hollander G, van Maanen L, Alkemade A, Keuken MC (2017) Towards a mechanistic understanding of the human subcortex. *Nat Rev Neurosci* 18:57-65.
- Keren NI, Lozar CT, Harris KC, Morgan PS, Eckert MA (2009) In vivo mapping of the human locus coeruleus. *Neuroimage* 47:1261-1267.
- Kriegeskorte N, Simmons WK, Bellgowan PSF, Baker CI (2009) Circular analysis in systems neuroscience: the dangers of double dipping. *Nature Neuroscience* 12:535-540.
- Pineiro G, Perelman S, Guerschman JP, Paruelo JM (2008) How to evaluate models: Observed vs. predicted or predicted vs. observed? *Ecol Model* 216:316-322.
- Poldrack RA, Mumford JA (2009) Independence in ROI analysis: where is the voodoo? *Soc Cogn Affect Neur* 4:208-213.
- Turker H, Riley E, Luh W-E, Colcombe S, Swallow K (2019) Estimates of locus coeruleus function with functional magnetic resonance imaging are influenced by localization approaches and the use of multi-echo data. *bioRxiv*.

REVIEWERS' COMMENTS

Reviewer #2 (Remarks to the Author):

The authors have responded reasonably to issues that we raised in review.

Reviewer #3 (Remarks to the Author):

Grueschow and colleagues examined if resilience to stressful events (medical internship) can be predicted from fMRI responses in the brainstem noradrenergic nucleus locus coeruleus (LC), and from pupil diameter, in healthy medical students performing an emotional Stroop task. As per my request the authors have replaced most findings in the main text with those that result from analyses that more appropriately correct for measurement issues.

Although I feel that the lack of an overall effect of the CI>II contrast (without median split and in the noise-corrected data) on LC responses remains problematic, I am now satisfied that the manuscript contains enough material for the reader to make up their own mind. I therefore support publication of this manuscript in its current form.